# Diversity Augmentation of Dynamic User Preference Data for Boosting Personalized Text Summarizers

**Parthiv Chatterjee**                                        *202421011@dau.ac.in*
*Knowledge Discovery & Management Lab*
*Dhirubhai Ambani University*

**Shivam Sonawane**                                          *202321014@dau.ac.in*
*Knowledge Discovery & Management Lab*
*Dhirubhai Ambani University*

**Amey Hengle**                                              *ameyhengle22@gmail.com*
*Indian Institute of Technology Delhi*

**Aditya Tanna**                                             *202103023@dau.ac.in*
*Knowledge Discovery & Management Lab*
*Dhirubhai Ambani University*

**Sourish Dasgupta**                                         *sourish_dasgupta@dau.ac.in*
*Knowledge Discovery & Management Lab*
*Dhirubhai Ambani University*

**Tanmoy Chakraborty**                                       *tanchak@iitd.ac.in*
*Indian Institute of Technology Delhi*

**Reviewed on OpenReview:** https://openreview.net/forum?id=JVx7Qi8tz3

## Abstract

Document summarization facilitates efficient identification and assimilation of user-relevant content, a process inherently influenced by individual subjectivity. Discerning *subjective* salient information within a document, particularly when it has multiple facets, poses significant challenges. This complexity underscores the necessity for *personalized summarization*. However, training models for personalized summarization has so far been challenging, particularly because diverse training data containing both user preference history (i.e., *click-skip* trajectory) and expected (gold-reference) summaries are scarce. The MS/CAS PENS dataset is a rare resource in this direction. However, the training data only contains preference history *without any target summaries, thereby blocking end-to-end supervised learning*. Also, the diversity in terms of topic transitions along the trajectory is relatively low, thereby leaving scope for better generalization. To address this, we propose PerAugy, a novel cross-trajectory shuffling and summary-content perturbation-based data augmentation technique that significantly boosts the accuracy of four state-of-the-art (SOTA) baseline user-encoders commonly used in personalized summarization frameworks (best result: 0.132↑ w.r.t AUC). We select two such SOTA summarizer frameworks as baselines and observe that when augmented with their corresponding improved user-encoders, they consistently show an increase in personalization (avg. boost: 61.2% ↑ w.r.t. PSE-SU4 metric). As a post-hoc analysis of the role of induced diversity in the augmented dataset by PerAugy, we introduce three dataset diversity metrics – TP, RTC, and DegreeD to quantify the induced diversity. We find that TP and DegreeD have a strong correlation with the user-encoder performance when trained on the PerAugy-generated dataset across all accuracy metrics, indicating that the increase in dataset diversity plays a major role in performance gain.

# 1 Introduction

The rapid increase in information requires efficient summarizers for fast comprehension and prioritization (Ter Hoeve et al., 2022). However, identifying "salient" information is subjective, particularly in multi-aspect documents, which makes *personalized summarization* critical. Training such models requires datasets with diverse user histories and subjective summaries. However, such datasets are scarce due to privacy concerns (Kirk et al., 2024; Liu et al., 2024). The MS/CAS PENS dataset (Ao et al., 2021), derived from the MIND dataset (Wu et al., 2020), is a key benchmark for training and evaluating SOTA personalized summarization models. It comprises user preference histories (i.e., sequences of *click* and *skip* interactions with news articles) along with user-specific gold-reference summaries. Although user-encoders trained on PENS/MIND have been employed in various frameworks (Song et al., 2023; Yang et al., 2023; Lian et al., 2025), evaluations have focused on accuracy and, more recently, on the *degree-of-personalization* (Dasgupta et al., 2024), overlooking the evaluation of the effectiveness of preference datasets as training data. To address this challenge, we propose **PerAugy** – a perturbation-based augmentation of historical user preference data for personalized summarization. A seed preference dataset is first modeled as a **User Interaction Graph** (**UIG**). The nodes of a UIG represent users' clicked and skipped documents (**d-nodes**) as well as their generated/expected summaries (**s-nodes**). A path, called **trajectory**, of the UIG starts with a specific user node (**u-nodes**) and terminates in d/s-nodes. The edges denote *click*, *skip*, *generate-summary*, and *click-summary* actions. The UIG, hence, is a pool of user trajectories representing dynamic user behavior histories. We apply `PerAugy` on this UIG.

The key design principle behind `PerAugy` is guided by recent findings in recommendation systems, which show that higher diversity in training data directly enhances user interaction sequence representation (Hu et al., 2019; Fabbri et al., 2021). In other words, "*trajectory diversity of training data is directly proportional to personalization capabilities of summarizer models*". In line with this principle, `PerAugy` has been designed as a cross-trajectory augmentation technique in which we propose a novel controlled both-ways exchange of user trajectory segments (termed **Double Shuffling** (**DS**)) between sampled copies of the original trajectories to create a new pool of diverse synthetic user trajectories. There are two primary controls in DS – (i) the **gap-length** that determines how much the new trajectory should resemble the original seed, and (ii) the **trajectory-length** that determines the number of diverse profiles to be synthesized. The DS operation mimics the stochastic diffusion of a users interest into diverse themes, reflecting naturalistic behavior. A perturbation operation is then applied to the exchanged s-nodes to eliminate unnatural thematic jitters at the boundaries of two segments on the new synthetic trajectory, enhancing realistic nature of the synthetic trajectories. The content of every exchanged s-node is replaced by its corresponding d-node content that most closely matches the nearest prior nodes of the new synthetic trajectory. The substitutions influence diminishes over a $k$-step context window that represents what is to be considered within proximity. Since this perturbation follows a $k$-order Markov Chain, we term this as **Stochastic Markovian Perturbation** (**SMP**). Through the DS and SMP operations, `PerAugy` generates synthetic profiles that capture a richer spectrum of thematic transitions and behavioral patterns. This controlled diversification ensures that user-encoders are exposed to varied histories and context shifts during training, thereby strengthening their ability to generalize and capture user preferences effectively.

We evaluate `PerAugy` across two core dimensions: (i) improvements in accuracy for SOTA user-encoder models trained on `PerAugy`-augmented data, and (ii) downstream gains in personalized summarization frameworks. For the first objective, we use the PENS dataset Ao et al. (2021) as the seed. We find that `PerAugy`-augmented PENS data consistently improves models like NAML, EBNR, and NRMS, showing average gains of 24%, 25%, and 18% across AUC, MRR, and nDCG@5&10, respectively. `PerAugy` outperforms the other baseline augmentation strategies including PENS-SH Song et al. (2023), S3 Grover et al. (2024), SDAInter Jiao et al. (2024), and SOTA LLM-as-augmentors (Llama-13B, Mistral-7B, DeepSeek-R1-14B). Additionally, `PerAugy` generalizes effectively to cross-domain datasets (e.g., OpenAI-Reddit), yielding consistent encoder gains of 19%, 25%, and 17% across accuracy metrics[1]. In the downstream task of personalized summarization, GTP and PENS frameworks show an average improvement of 61.2%↑ in PSE-SU4, with setups like PENS+NRMS+T2 reaching up to 75% in PSE-RG-SU4.

---

[1]We have released the augmented datasets across `PerAugy` and all the baselines as a part of the **AugPerSumm** collection.

To have a deeper analysis of the performance of `PerAugy` and its relationship with the achieved diversity, we use two simple diversity evaluation metrics, Unique Topics per Trajectory (TP) and Rate of Topic Change (RTC). While TP and RTC capture topical richness and frequency of topical shifts in a trajectory, they fail to capture the effect of inserted s-nodes into trajectories, and their alignment with the corresponding d-nodes. This motivates to design of a more effective embedding-based metric `DegreeD`. `DegreeD` is designed to capture the proportional shift of successive s-nodes to that of the d-nodes, as well as the faithfulness of s-nodes towards the corresponding d-node. We evaluate the diversity of `PerAugy` generated datasets in comparison to baseline datasets. We find a strong correlation between enhanced `DegreeD` and user-encoder performance, with Pearson $r$: 0.68, Spearman's $\rho$: 0.73, and Kendall's $\tau$: 0.57. These results collectively demonstrate `PerAugy`'s effectiveness in enhancing data diversity, which in turn, enhances user modeling via user-encoders, and boosts downstream personalization[2].

## 2 Background

### 2.1 Dynamic User Preference (vs. Static User Persona)

It is crucial to distinguish between user persona and user-preference history in the context of preference datasets. Persona information, such as address, nationality, or broad interests like genres, tends to remain *relatively static over time.* In contrast, preference histories are *highly dynamic*, since they constitute interaction (or reading) behavior as a temporal sequence that is complex, and spans across multiple subtopics and discourses. A user is unlikely to display consistent behavioral repetition; for instance, it is improbable that Alice's weekly reading consistently centers only on European soccer highlights while predictably skipping U.S. politics or film updates. This distinction reinforces the need for training datasets that consist of dynamic user preferences rather than static persona features.

### 2.2 Personalized Summarizers

Most research on personalized summarization assumes a static user persona (i.e., user profile information that is relatively time-invariant). These works leverage the simplicity of guided (or controlled) summarization. In this direction, Dou et al. (2021) proposed *GSUM*, where the goal was to inject a *generic guidance* in terms of *explicit* user-provided key-phrases that are restricted to the query-document only and do not account for the dynamic shift in user preference. *CTRLSum* and *TMWIN* were also proposed on similar lines, where either static control signals were given explicitly or extracted from dialogue sessions (He et al., 2022; Kirstein et al., 2024). Xiao et al. (2024b) proposed the *Tri-Agent* personalized summarizer that was iteratively trained under an RL setup using an *oracle-as-an-instructor* that knows historical user-edits of previous summaries. However, the user-edit-preference does not entail subjectivity and is also static. Static user preference is unrealistic in most situations, while a shift in topics of interest is the norm.

In the more realistic context of dynamic user preference, *personalized summarization* refers to the extent to which a summarization model aligns its outputs with a reader's subjective expectations. The subjectivity is a function of the user's characteristic shift in preference as reflected through the *reading history* – a temporal trajectory of the reading and skipping actions of the user on a sequence of documents. It is important to note that this trajectory may occasionally be interleaved by the actions of generating and reading summaries instead of the full-length documents. The PENS framework (with external user-encoders such as NRMS, NAML, EBNR (Wu et al., 2019b;a; Okura et al., 2017)) is an early example that attempts to address this (Ao et al., 2021). The plugged user-encoders embed the user behavior trajectories from the PENS dataset. However, the encoders do not capture the dynamic temporal behavioral trend and are also tightly-coupled with the three injection techniques (T-1/2/3) of the encoder-decoder-based pointer-generator summarizer. Song et al. (2023) proposed the *GTP* framework that follows a similar summary-editing approach as *Tri-Agent*, except there is no explicit static guidance but rather the editing (latent) control is generated from the user trajectory. However, the internal user encoder, TrRMIo does not encode the dynamically shifting user trajectory without differentiating short vs. long-term influences. Also, so far we have not found any work that explicitly differentiates the various semantics of the user actions – *click, skip, read-summary.*

---

[2]**Codebase**: `https://github.com/KDM-LAB/PerAugy-TMLR/`; **Presentation**: `https://tinyurl.com/PerAugy-KDMLab`.

| UIG Symbols | |
|---|---|
| $\text{UIG} = \langle N, E \rangle$ | Directed acyclic graph of user-document-summary interactions. |
| $u_j^{(t_0)}$ | User node $j$ at time $t_0$. |
| $d^{(t_p)}$ | Document node at time $t_p$. |
| $s_j^{(t_q)}$ | Summary node for user $j$ at time $t_q$. |
| $\tau_{u_j}$ | User $j$'s interaction trajectory. |
| $\mathcal{T}$ | Trajectory pool dataset. |
| **PerAugy Symbols** | |
| $\mathcal{T}^{\text{syn}}$ | Synthetically created trajectory pool by incorporating s-nodes. |
| $\tau_{u_j, \text{target}}$ | Trajectory selected for augmentation. |
| $\tau_{u_i, \text{seg}}$ | Segment extracted from a source trajectory. |
| $g_l$ | Gap length between inserted segments. |
| $\tau_{\text{DS}}^{u_j}$ | Final trajectory after double shuffling. |
| $\tau_{\text{SMP}}^{u_j}$ | Final trajectory after SMP. |
| $k$ | Context window size for computing influence. |
| $\lambda$ | Exponential decay constant for context weighting. |
| $p_{\text{SMP}}$ | Probability of perturbing a summary node. |

Table 1: Notations and denotations used in the paper.

## 2.3 Personalized Summarization Datasets

A key challenge in the task of personalized summarization is the lack of suitable training data across varied domains that covers the three key conditions: (i) chronological ordering of *evolving* user actions, i.e., *historical trajectories*, (ii) subjective summary expectations (i.e. gold references/ratings) for same document by multiple users, and (iii) diversity and dynamicity w.r.t topics and topic transition. Standard summarization datasets like CNN/DM, Multinews (Hermann et al., 2015; Fabbri et al., 2019) do not qualify. Only a few real-world datasets, notably PENS (Ao et al., 2021) and PersonalSum (Zhang et al., 2024), meet these criteria[3].

## 2.4 Personalized Summarization Evaluation

Vansh et al. (2023) introduced the notion of *degree-of-personalization* as a measure of the *subjective* user experience (UX), which is inversely related to both information overload and lack of expected information. They proved, arguably for the first time, that *accuracy metrics are unsuitable for measuring UX*, i.e., there are real-world cases where we find low UX even with high accuracy. As a solution, they proposed EGISES as a metric to quantify the degree-of-personalization. EGISES was further modified and completed by Dasgupta et al. (2024) to incorporate a penalty due to accuracy drop and the PerSEval metric was proposed (detailed exposition in Appendix A). In this paper, we adopt PerSEval to demonstrate that SOTA user-encoders trained on a `PerAugy`-generated augmented dataset enhance the performance of SOTA personalized summarizers.

## 3 Modeling User Preference Datasets

In the following section, we first introduce User-Interaction-Graph (UIG) – a temporal knowledge graph-inspired data model for capturing dynamic user behavior trajectories. `PerAugy` operates on UIG to generate diverse synthetic trajectories, as discussed in depth in Section 4.

### 3.1 User-Interaction Graph (UIG)

A UIG is a Directed Acyclic Graph UIG: $\mathcal{G} = \langle N, E \rangle$ where $N(\text{nodes}) \equiv \{u_j^{(t_0)}\} \cup \{d^{(t_p)}\} \cup \{s^{(t_q)}\}$ and $E$ (edges) $\equiv \{a_d^{(t_p)}\} \cup \{a_s^{(t_q)}\}$. The node set $N$ contains 3 disjoint types:

---

[3]PersonalSum was skipped because of insufficient samples and it being in Norwegian, makes it infeasible to test performance boost of summarizers that are not pre-trained in Norwegian.

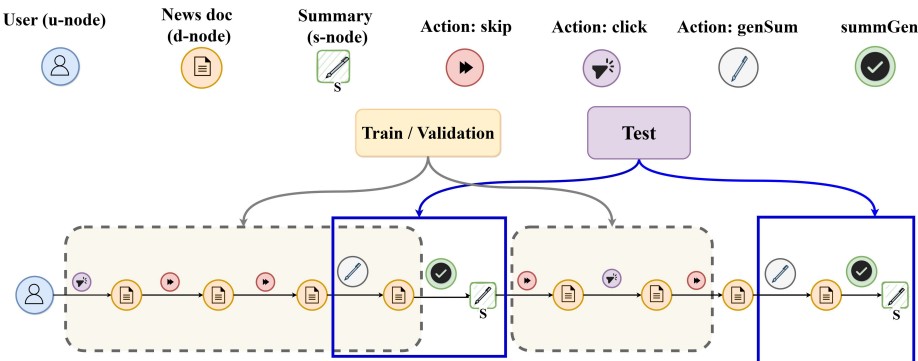

Figure 1: **UIG construction pipeline for PENS-styled datasets: Step 1**: *Documents from train/valid data are sequenced as d-nodes*; **Step 2**: *Reference personalized headlines for an intersecting d-node from test data are interleaved as s-nodes based on time-step*; **Step 3**: *If no intersecting d-node is found, the s-node along with corresponding d-node from test data are simply appended at their respective time-step.*

- **u-nodes** $u_j^{(t_0)}$: the j-th user at the initial time-step $t_0$ (source/start node of $\mathcal{G}$);

- **d-nodes** $d^{(t_p)}$: documents the user interacts with at time-step $t_p$; d-node may re-appear at multiple time-steps.

- **s-nodes** $s_j^{(t_q)}$: user-specific summaries requested or produced at time $t_q$ for a d-node viewed at time-step $t_{q-1}$.

Edges $E$ represent user interactions on the nodes:

- $a_d^{(t_p)}$: a user interaction on a d-node $d^{(t_p)}$ at time-step $t_p$ such that $a_d \in \{$*click, skip, summarize*$\}$, where *click* denotes positive engagement (interest), *skip* denotes non-engagement (disinterest) and *summarize* (also called *genSumm*) explicitly captures the interest to read a summarized version of the d-node (during inference, *genSumm* represents user command to generate a personalized summary of $d^{(t_p)}$);

- $a_s^{(t_q)}$: the follow-up edge of *summarize* denoting *summarized* version of $d^{t_{q-1}}$, acting on $s^{t_q}$ (also termed as *summGen*), indicating either a gold-reference summary written by user (in case UIG is treated as a training data), or a desired summary to be generated by the model (during inference).

**Trajectory**: Given a UIG, the preference history (termed *trajectory*) of $u_j$ is a sequence of interactions, denoted $\tau^{u_j}$, starting at $t_0$ and ending at a d-node or s-node at $t_{l-1}$, where $l$ is the trajectory length. Hence, a UIG is a pool of trajectories $\mathcal{T}$.

A UIG can hence be seen as a dynamic temporal knowledge graph (TKG) of user behavior. We now formally define Personalized Summarization as follows:

**Definition 1. *Personalized Summarization*** *Given a user trajectory $\tau^{u_j}$ of length $l$, a personalized parameterized ($\theta$) summarizer model $\mathcal{M}_\theta$ takes a query document node ($d_q^{(t_l)}$) and generates a corresponding user ($u_j$) specific summary $s_{(q,u_j)}^{(t_{l+1})}$, where $\mathcal{M}_\theta$ is trained on $\mathcal{T}_{train}$ to approach the upper-bound of a chosen personalization metric (in our case, we use PerSEval (PSE)).*

### 3.2 UIG Construction from Preference Data

In the parlance of UIG, preference datasets suitable for personalized summarization training and evaluation are of two categories – (i) those which can be directly modeled into a trajectory pool $\mathcal{T}$ (e.g., the PENS

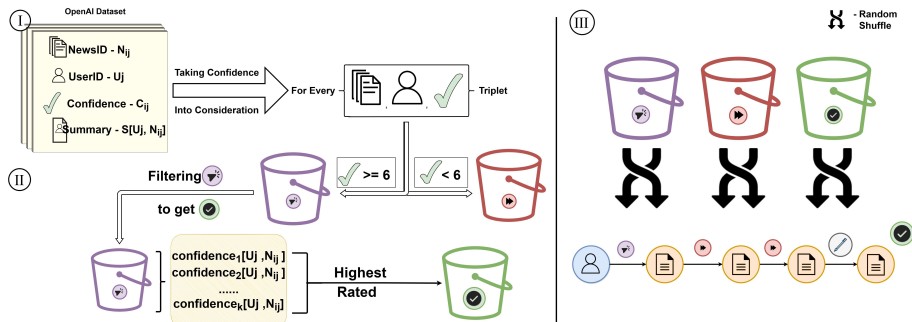

Figure 2: **UIG construction pipeline for OpenAI-styled datasets: Step 1:** *Extract NewsID, UserID, confidence, and summary*; **Step 2:** *Select top-rating* $< U_j, N_{ij} >$ *click pairs from filtered confidences*; **Step 3:** *Shuffle clicks, skips, and summaries to form trajectories.*

dataset (Ao et al., 2021)), dataset statistics are in Table 8, and (ii) those which lack user trajectories but contain discrete d-nodes, *model-generated* s-nodes (in contrast to user-generated s-nodes as per UIG definition), and *subjective* user feedback in the form of rating and the associated confidence score for that rating (e.g. OpenAI-Reddit dataset (Völske et al., 2017)), statistics in Table 9. [4]. In our experiments, we select the OpenAI-Reddit dataset for establishing cross-domain generalizability of `PerAugy` and its broader applicability in more widely available datasets. We describe the UIG construction method for both types as follows:

**PENS-styled Datasets** The construction of UIG is straightforward in the first case and is done in two steps. In the first step, *click* and *skip* interactions in the train dataset are mapped to document nodes (d-nodes) as incoming edges, forming the corresponding u-tier pool $\mathcal{T}$. As an example, for the PENS dataset, the *clkNews* interaction corresponds to a *click* edge and *uclkNews* to a *skip* edge, forming $\mathcal{T}_{\text{base}}^{\text{P}}$. However, the PENS dataset lacks user-specific s-nodes (i.e., true interest evolution over time), rendering $\mathcal{T}_{\text{base}}^{\text{P}}$ an *incomplete representation of the user dynamic preference*. Despite this, most recent frameworks train on $\mathcal{T}^{\text{PENS}}$ using history or document titles as "pseudo-targets" or via unsupervised learning (Ao et al., 2021; Song et al., 2023; Yang et al., 2023; Lian et al., 2025). We address this in the second step, where we incorporate the s-nodes from the test dataset ($\mathcal{T}_{\text{test}}$) at their associated time-steps into $\mathcal{T}$ with the addition of *genSumm* (directed to the d-node whose corresponding s-node is incorporated) and *summGen* (to the incorporated s-node) edges, forming a derived (and more diverse) user-profile pool $\mathcal{T}_{\text{base}}^{\text{syn-P}}$ (see Figure 1).

**OpenAI-styled Datasets** For the second category of datasets, we first do a pre-construction classification of clicked and skipped d-nodes for every human rater $u_j$. This is done based on a simple heuristic of selecting those d-nodes as clicked which has at least one corresponding model-generated summary (NT: there can be multiple models) that received a confidence score above a chosen threshold. In the case of OpenAI-Reddit, we chose the threshold for clicked d-nodes to be 6 out of 9 (see Figure 2-II), forming $\mathcal{T}_{\text{base}}^{\text{OAI}}$. We then select the best model-generated summary (i.e., the highest rated one by $u_j$) as the surrogate expected s-node for $u_j$ (Figure 2-II). We then randomly sequence all such $(d - s)$-node pairs along with the skipped d-nodes to form $\tau^{u_j}$ (thereby $\mathcal{T}_{\text{base}}^{\text{syn-OAI}}$; Figure 2-III). This method makes UIG-modeling *compatible with most summarization datasets that are not PENS-styled.* Additionally, it also addresses cold-start problem as $\mathcal{T}_{\text{base}}^{\text{syn-OAI}}$ itself is synthetically designed as a random sequence. A detailed pseudo-code of UIG construction is given Algorithm 1.

## 4 PerAugy: Augmentation of Base UIG

In this section, we introduce `PerAugy`, a novel data augmentation method designed to improve personalization in summarizers by enhancing the accuracy of user-encoders. The design is motivated by the objective to

---

[4]For detailed exposition of datasets, see Appendix B

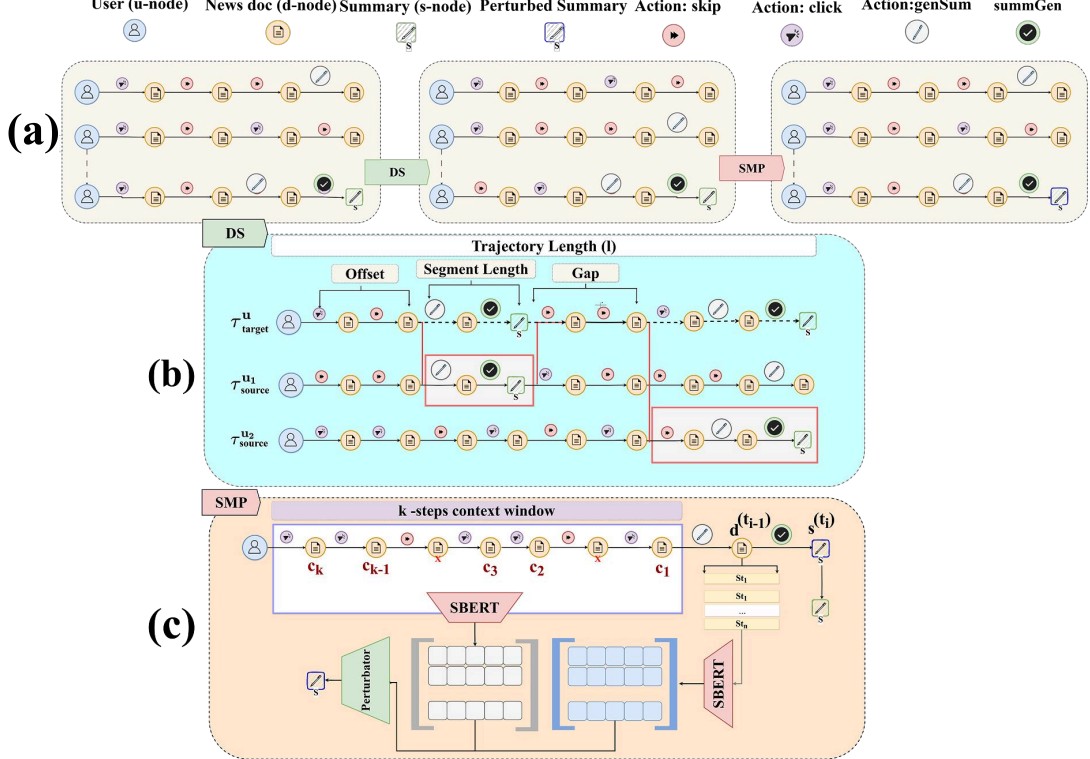

Figure 3: **PerAugy:** our proposed framework – (a) *Pipeline overview depicting the two-step augmentation,* (b) *Double Shuffling (DS) to ensure cross-trajectory augmentation and induce diffusion,* (c) *Stochastic Markovian Perturbation (SMP) to smoothen the s-nodes and modulate random diffusion incorporated in DS stage.*

create diverse realistic user trajectories for training, under the hypothesis that "*trajectory diversity of training data is directly proportional to personalization capabilities*".

## 4.1 PerAugy Pipeline: Overview

The PerAugy (**Per**turbation-based **Aug**mentation) pipeline has four steps. In the first step, we randomly sample (without replacement) $m$ seed trajectories ($\mathcal{T}_{\text{sample}}^{m}$) from a given UIG (i.e., trajectory pool $\mathcal{T}_{\text{base}}^{syn}$). In the next step, we perform the "*Double Shuffling* (DS)" operation that selects each of the sampled trajectories as the target and **substitutes** trajectory-segments at different time-steps of the target with that of segments from the other $m-1$ trajectories. This leads to a modified "shuffled" sample $\mathcal{T}_{\text{DS}}^{m}$. We describe the DS operation in details in section 4.2. We then select the s-nodes of each trajectory $\tau \in \mathcal{T}_{\text{DS}}^{m}$ via a Bernoulli Trial and **perturb** the summary content on the basis of the corresponding d-node's similarity with preceding d-nodes. The details of the perturbation method, called "*Stochastic Markovian Perturbation* (SMP)", is given in section 4.3. The resulting perturbed trajectories form $\mathcal{T}_{\text{SMP}}^{m}$ which represents a *new set of synthetic user preference history data*. Note that $\mathcal{T}_{\text{DS}}^{m}$ is added back into the trajectory pool ($\mathcal{T}_{base}^{syn} - \mathcal{T}_{\text{sample}}^{m}$) before the next iteration of sampling is done. PerAugy terminates after $|\mathcal{T}_{\text{base}}^{syn}|/m$ sampling iterations. We depict the pipeline in Figure 3(a).

## 4.2 Double Shuffling

In this section, we detail the Double Shuffling (DS) operation. The key design heuristics behind DS are that behavioral subsequences within real-life trajectories can be stitched together to form more diverse, realistic synthetic trajectories. That is, a sequence of interactions of Alice can be stitched with that of Bob and Joe to result in a realistic synthetic profile that behaves like Alice during the first session, then Bob in the second, and then finally Joe. To simulate this, following the sampling method

described in the previous section, given a target trajectory $\tau_{\text{target}}^{u_j} \in \mathcal{T}_{\text{sample}}^m$ corresponding to a user $u_j$ (i.e., an existing **real** user), we first randomly select an "*Offset*" **O**. This offset determines the **early-stage behavior sequence** (behavior sequence is termed "*trajectory segment*" $\tau_{seg}$) that **should remain the same** as $u_j$. **O** helps to make sure that a new trajectory does not start with an unrealistic initial segment. The random selection helps to generate early-stage trajectory segments of varying lengths in the augmented dataset. For example, if the original trajectory of Alice $\tau_{\text{base}}^{Alice}$ looks like: CLK:MT→CLK:YR→SKP:PR→CLK:GW→SKP:TA→SKP:CR→CLK:MB→SKP:DK→CLK:WC. Alice's first three interactions involve reading "*Meditation tips*" (CLK:MT) and "*Yoga retreat guides*" (CLK:YR), and skipping "*Political tension rises in Russia*" (SKP:PR), an offset $O = 3$ ensures these three steps remain unchanged in her synthetic trajectory, preserving a natural beginning. The random selection of $O$ also helps to generate early-stage trajectory segments of varying lengths in the augmented dataset.

In the next step, we select $m - 1$ trajectory segments ($\tau_{seg}^{u_{i=1:m-1}}$) from each of the remaining $m - 1$ "*source*" trajectories $\tau_{\text{source}}^{u_{i=1:m-1}} \in \mathcal{T}_{\text{sample}}^m$ corresponding to $m - 1$ users and substitute corresponding target segments having same length as that of these source segments. After Alice's preserved $O = 3$ steps, a segment of length $l_{\text{seg}_B} = 2$ from Bobs trajectory where he was browsing "*Concert schedules*" (CLK:CS) and "*Band interviews*" (CLK:BI) gets substituted in Alice's synthetic trajectory $\tau_{\text{DS}}^{Alice}$.

The $m - 1$ time-steps on $\tau_{\text{target}}^{u_j}$ where the substitutions occur are controlled by "*Gap*" (**G**), which determines the number of time-steps that should be kept intact (i.e., same intermediate behavior sequences as the original $u_j$). If $G = 2$, then after Bobs exchanged segment ends, two of Alice's untouched interactions, say, "*Cooking recipes*" (SKP:CR) and "*Mindfulness blogs*" (CLK:MB) – are preserved in $\tau_{\text{DS}}^{Alice}$ before the next substitution. Following this, another source segment may be introduced. For example, a segment of length $l_{\text{seg}_J} = 2$ from Joe could be substituted, representing his interactions with "*Sports highlights*" (CLK:SH) and "*Malaria Outbreaks*" (SKP:MO). The final $\tau_{\text{DS}}^{Alice}$ will therefore look like: CLK:MT→CLK:YR→SKP:PR→**CLK:CS**→**CLK:BI**→SKP:CR→CLK:MB→**CLK:SH**→**SKP:MO**.

Gap length is a hyper-parameter that we ablate on in our experiments (see section I.1). Longer gaps would signify that the new synthetic trajectory $\tau_{\text{DS}}^{u_j}$ is more similar to the corresponding original $\tau_{\text{sample}}^{u_j}$, while longer source segments and higher $m$ value would lead to longer $\tau_{\text{DS}}^{u_j}$. The random selection of $O$ also helps to generate early-stage trajectory segments of varying lengths in the augmented dataset.

Although DS introduces diversity by aggregating trajectory segments from different users and threads them up at different time-steps thereby altering their original positions, it *fails to ensure that the s-nodes that come intact along with the source segments have realistic coherence with the preceding nodes*. This is because the source s-node has been influenced by the **preceding source nodes** that **form its "*history*"**, and hence, may not be compatible with the new history in the target trajectory. To address this, we introduce a subsequent operation on each double-shuffled trajectory in $\mathcal{T}_{\text{DS}}^m$ called "*Stochastic Markovian Perturbation*" that we describe in the next section.

## 4.3 Stochastic Markovian Perturbation (SMP)

SMP smoothens the newly substituted incompatible s-node $s^{(t_i)}$ at time-step $t_i$ by operating over a backward sliding *context window* $\tau^{u_{\text{target}}}c_k$ of $k$ time-steps, derived from the corresponding d-node at $t_{i-1}$. This process refines $s^{(t_i)}$ by perturbing it; i.e. by replacing it with a sentence that better aligns with the temporal context of the target user. Specifically, SMP selects the top-$p$ sentences from $d^{(t_{i-1})}$ that are most *influenced* by the context nodes $c_{q=1:k}$ within the window. In our experiments, we use top-1 selection, since the s-node represents a summary-level headline, making a single representative sentence sufficient. The notion of "influence" is quantified through the Root Mean Square Distance (RMSD) between each candidate sentence $s_{t_p}$ in $d^{(t_{i-1})}$ and each context node $c_q$ in $\tau^{u_{\text{target}}}c_k$. Sentence and context representations are computed using SBERT embeddings (as detailed in Section E). To reflect temporal relevance, this influence is weighted by an exponential decay factor $e^{-\lambda \cdot pos(c_q)}$, where $pos_{c_1} = 0$ denotes the most recent context and thus receives maximum weight. As a result, earlier context nodes contribute less to the influence score. This formulation characterizes SMP as a $k$-order Markov process, where the prediction at $t_i$ depends on a weighted combination

of the preceding $k$ time steps. A compact representation of SMP follows:

$$\text{SMP}(s^{(t_i)} \in \tau^{u_{\text{target}}}) = \hat{s}^{(t_i)} = \arg\min([\boldsymbol{\Sigma}]_{(n_{d^{(t_{i-1})}} \times k)} \cdot [e^{-\lambda \cdot pos_{(c_q)}}]_{(k \times 1)});$$

$$\text{where: } \boldsymbol{\Sigma}_{(p,q)} = \sigma(\mathbf{e}_{st_p}, \mathbf{e}_{c_q}); n_{d^{(t_{i-1})}} : \text{No. of sentences in } d^{(t_{i-1})}; \sigma : \text{RMSD} \tag{1}$$

Once the minimum-scoring sentence $\hat{s}^{(t_i)}$ is identified via this influence-weighted RMSD computation, the original s-node $s^{(t_i)}$ is replaced with $\hat{s}^{(t_i)}$. RMSD is chosen over other metrics because it provides a straightforward, geometry-based measure of dissimilarity between embeddings, capturing overall distributional mismatch rather than just directional alignment (as in cosine similarity). This makes it sensitive to subtle semantic shifts, which is desirable when ensuring that substituted s-nodes remain contextually coherent. The DS operation mimics the stochastic diffusion of a user's interest into diverse themes, reflecting natural behavior; hence, **smoothing every s-node may inhibit the intended thematic diffusion**. Consequently, s-node $s^{(t_i)}$ is selected for perturbation via a Bernoulli trial with perturbation probability $p_{\text{SMP}}$ [5]. As an illustrative example, suppose at time-step $t_i$ Alices double-shuffled trajectory inserts an s-node from Bob, e.g., "*Booked tickets for a rock concert*", but her recent context window ($k = 3$) contains "*Searching for yoga retreats*", "*Reading reviews of meditation centers*", and "*Browsing healthy recipes*". SMP looks back to the previous d-node, extracts candidate sentences, and scores them against this context with exponential decay weighting. A more coherent candidate, such as "*Workshops focus on mindfulness practices*" achieves the lowest weighted RMSD and replaces the original s-node, ensuring the trajectory flows naturally with Alice's wellness theme instead of abruptly shifting to concerts.

The DS and SMP operations generate synthetic user profiles $\hat{u}$ and trajectories $\tau^{\hat{u}}_{\text{SMP}} \in \mathcal{T}^m_{\text{SMP}}$. We ablate on the context window size $k$, decay constant $\lambda$, and $p_{\text{SMP}}$ in Section I.1. The pseudo-code for DS is provided in Algorithm 3 and that of SMP is provided in Algorithm 4.

## 5 Evaluation

To evaluate the overall efficacy of `PerAugy`, we frame our investigation around two central research questions: **RQ-1**: Does applying `PerAugy` on seed training data lead to higher accuracy in SOTA user-encoders? **RQ-2**: If so, then do the enhanced user-encoders, in turn, improve SOTA personalized summarization frameworks?

### 5.1 Experiment Setup

In this section, we describe our detailed experimental setup: The training and testing datasets, Training procedure, Baseline augmentations, user-encoders, and summarizers, and the Evaluation metrics utilized to assess the effectiveness and utility of `PerAugy`.

#### 5.1.1 Augmented Synthetic Datasets

**Training Data.** We create two training datasets using `PerAugy`: $\mathcal{T}^{\mathcal{E}}_{\text{DS}}$ and $\mathcal{T}^{\mathcal{E}}_{\text{DS+SMP}}$. The dataset $\mathcal{T}^{\mathcal{E}}_{\text{DS}}$ is a *mixed bag* of trajectories sampled from ten different augmented datasets generated with varying <Gap-length $g_l$, Trajectory-length $l$> configurations *without SMP operations*. Among them, five datasets $\{\mathcal{T}^{i=1:5}_{\text{DS}}\}$ use a fixed $l = 150$ and vary $g_l \in \{10, 15, 20, 25, 40\}$, while the remaining five $\{\mathcal{T}^{i=6:10}_{\text{DS}}\}$ use a fixed $g_l = 25$ and vary $l \in \{50, 100, 125, 175, 200\}$. Each of the ten datasets contains 400K trajectories. From each, we sample 10% to construct the final $\mathcal{T}^{\mathcal{E}}_{\text{DS}}$.

Similarly, $\mathcal{T}^{\mathcal{E}}_{\text{DS+SMP}}$ is constructed via the same mixed-bag sampling strategy, but using SMP operations applied to each of the ten augmented datasets with a decay constant $\lambda = 0.3$, perturbation probability $p_{SMP} = 0.8$, and context length $k = 10$. Proportional sampling from diverse configurations helps mitigate overfitting and increases the generality of preference histories. We generate four variants in total: (i) $\mathcal{T}^{\mathcal{E}-P}_{\text{DS}}$, (ii) $\mathcal{T}^{\mathcal{E}-OAI}_{\text{DS}}$, (iii) $\mathcal{T}^{\mathcal{E}-P}_{\text{DS+SMP}}$, and (iv) $\mathcal{T}^{\mathcal{E}-OAI}_{\text{DS+SMP}}$, derived respectively from $\mathcal{T}^{\text{syn-P}}_{\text{base}}$ and $\mathcal{T}^{\text{syn-OAI}}_{\text{base}}$.

---

[5]All symbols used are described in Table 1.

**Test Data: User-Encoder Evaluation.** We construct the test dataset $\mathcal{T}_{\text{test}}^P$ to evaluate user encoders under realistic interaction conditions accurately. The PENS validation set lacks skipped d-nodes (thereby rendering it less effective to evaluate user-encoders). The PENS test set lacks explicit negative dnodes in the target bin, we merge Phase-1 clicks ($\tau_{s\acute{e}g_h}^{u_j}$) with Phase-2 $(d, s)$ pairs ($\tau_s^{u_j}$) in sequential order. We split $\tau_s^{u_j}$ in half: the *first half* is appended to $\tau_{s\acute{e}g_h}^{u_j}$ to form the user history $\tau_{h_{\text{test}}}^{u_j}$, while the *second half* serves as the candidate set for next-click prediction. To ensure balance, we augment positive samples with *negative samples*-documents not clicked by user $u_j$ in either Phase-1 or 2 –randomly drawn from the index range $[50 : n_s^{u_j}]$, where $n_s^{u_j}$ is the total number of s-nodes in $\tau_s^{u_j}$. This process yields a realistic distribution of clicked and non-clicked items, enabling a fair **next click prediction** evaluation. The final test set contains 103 trajectories with $\approx 20K$ candidate pool (both positive and negative samples) of d-nodes with $10K$ target d-nodes, thereby not altering the settings of the PENS test dataset. (Table 8).

**Test Data: Personalized Summarizer Evaluation.** We evaluate personalized summarizers using the original PENS test set $\mathcal{T}_{\text{test}}$. For each user $u_j$, we retain Stage-1 click history $\tau_{s\acute{e}g_h}^{u_j}$ and use the 200 $(d, s)$ pairs from Stage-2 as summarization queries. The model generates personalized summaries $\hat{s}^{u_j}$ conditioned on the query document $d_{\text{query}}$ and $\tau_{s\acute{e}g_h}^{u_j}$ alone; the intermediate $(d, s)$ pairs are not appended, resulting in inferred summaries $\hat{s}_{1:200}^{u_j}$ based solely on $\tau_{s\acute{e}g_h}^{u_j}$.

### 5.1.2 User-Encoder Training

We train the user-encoder models *from scratch* on each of the mixed sets $\mathcal{T}_{\text{DS}}^{\mathcal{E}}$ and $\mathcal{T}_{\text{DS+SMP}}^{\mathcal{E}}$ (with batch size: 128 trajectories; epochs 2; Adam Optimizer ($\alpha : 1e - 4;\ \beta_1 = 0.9;\ \beta_2 : 0.999; \epsilon = 1e - 8$)), in contrast to standard fine-tuning, to analyze: (a) the extent to which synthetic datasets can replace real datasets, especially when such datasets are extremely scarce, and (b) to clearly understand the effect of the hyperparameters of `PerAugy` under ablation. During training, we split a user's trajectory $\tau_{\text{DS/DS+SMP}}^{u_j}$ into an input segment, also termed **train history-segment** ($\tau_{h_{\text{train}}}^{u_j}$), and a target segment $\tau_{\text{target}}^{u_j}$ at random time-step within the interval $[l^{u_j}/2, l^{u_j} - 3]$. The nodes of $\tau_{\text{target}}^{u_j}$ form a target candidate bin for the next node prediction task. We ablate on $\tau_{h_{\text{train}}}^{u_j}$ length in section 6.1. We further fine-tune TrRMIo (with epoch 1) on the top of $\mathcal{T}_{\text{base}}^{\text{syn-P}}$ to ascertain the impact of fine-tuning by each baseline augmentations (as discussed in Section 5.1.3).

### 5.1.3 Baseline Augmentation Methods

We evaluate `PerAugy` against three state-of-the-art (SOTA) algorithmic augmentation methods. We select PENS-SH as a strong baseline, as it is specifically designed for personalized summarization. We also choose S3-Aug as intrea-trajectory baseline, and SDAInter as inter-trajectory baseline, along with three LLM-based augmentation setups. PENS-SH merges multiple user interaction trajectories from $\mathcal{T}_{\text{train}}^P$ into synthetic ones by aligning common d-nodes to create diverse pseudo-users. S3-Aug applies intra-trajectory segment-shuffle-stitch operations to perturb temporal structure while preserving local coherence in $\mathcal{T}_{\text{train}}^{\text{S3}}$. SDAInter swaps interchangeable sub-sequences between user trajectories based on anchor overlaps and IoU (Interaction over Union) confidence, producing cross-user hybrids in $\mathcal{T}_{\text{train}}^{\text{SDA}}$. We convert each of these into UIG-compatible datasets, $\mathcal{T}^{\text{syn-PSH}}$, $\mathcal{T}^{\text{syn-S3}}$, and $\mathcal{T}^{\text{syn-SDA}}$, by injecting s-nodes from test dataset summaries to evaluate diversity. In the LLM-as-augmentor setup, we use LLaMA-2-13B, Mistral-v2-Instruct, and DeepSeek-7b-chat with two prompt strategies. **Chain-of-Thought (CoT)** prompts guide LLMs to reason step-by-step through user preferences to generate personalized summaries. **Prompt-Chaining (PC)** splits the task: first generating user behavior, then using that to generate personalized summaries. All augmented datasets are used to train user encoders, and we evaluate them to assess their diversity and impact on user-encoders.[6] The baseline augmentations are discussed in details in Appendix C.1, and prompt details are depicted in Figure 9 (Chain-of-thoughts) and Figure 10 (Prompt Chaining).

---

[6]UIG statistics are detailed in Table 10.

### 5.1.4 User-Encoder Baselines

To study RQ-1, we evaluate four SOTA user-encoder models originally trained on PENS or its source dataset MIND (Wu et al., 2020): **NAML** (Wu et al., 2019a), **NRMS** (Wu et al., 2019b), **EBNR** (Okura et al., 2017), and **TrRMIo** (Song et al., 2023) since these encoders form four structural variations of history sequence modeling. **NAML** employs a multi-view additive attention approach to integrate news features (e.g., titles, categories) and models user interests via attention over browsing history. **NRMS** applies multi-head self-attention in both news and user encoders to capture semantic relations in titles and personalize based on attended browsed content. **EBNR** is click-order GRU that incorporates dwell-time-based implicit negatives, combining transformers and attention to model user preferences from both positive and negative signals. **TrRMIo** leverages pre-trained full-sequence transformers with attention pooling, defining user interest through CTR-based filtering that emphasizes low-CTR news as indicators of core user preference. All baseline encoders are described in Appendix C.2.

### 5.1.5 Personalized Summarization Baselines

Most recent frameworks for personalized summarization are trained on the trajectory dataset $\mathcal{T}_{\text{base}}^{P}$ using either of two methodological paradigms. The first involves reinforcement learning setups, where models are optimized using a **"pseudo-target"** such as user history (Ao et al., 2021) or document title (Song et al., 2023) to approximate personalized summaries. The second involves unsupervised setups, where the training objective is not based on explicit summaries but instead aims to reduce the *surprise* in the generated summary (Yang et al., 2023) or to align user representations with the style-preference centroid of similar users in their neighborhood (Lian et al., 2025). We could not experiment with models of the second paradigm since they are closed[7]. To systematically investigate RQ-2, we adopt two state-of-the-art personalized summarization frameworks. The first is **PENS** (Ao et al., 2021), a pointer-network-based model trained on the PENS dataset using policy gradientbased reinforcement learning. PENS utilizes user embeddings derived from third-party user-encoders such as NAML, NRMS, and EBNR, injecting them into the generation process to personalize summaries. The second is **GTP** (Song et al., 2023), which follows a two-stage late-fusion approach trained on PENS-SH. In this framework, a general headline is first generated using a transformer-based encoderdecoder model, and then personalized in a second stage using TrRMIogenerated user embeddings to control stylistic and semantic refinements. Baseline frameworks are detailed in Appendix C.3.

## 5.2 Evaluation Metrics

### 5.2.1 Encoder Evaluation

To evaluate user-encoders on the task of *Next-item Prediction*, we use standard metrics: AUC, MRR, and nDCG@5&10. AUC (Area Under the ROC Curve) measures the models ability to rank a positive item higher than negative ones, indicating overall ranking quality. MRR (Mean Reciprocal Rank) evaluates the position of the first relevant item in the ranked list, giving higher scores when relevant items appear earlier. nDCG (normalized Discounted Cumulative Gain) at cutoff positions 5 and 10 assesses both the relevance and position of items, rewarding models that rank relevant items higher in the top-K predictions.

### 5.2.2 Personalization Evaluation

We adopt PSE-SU4 as the PerSEval (only existing personalized summarization evaluation metric known so far) variant to measure the boost in *degree-of-personalization* for both frameworks, owing to its high humanjudgment correlation (Pearson's $r$: 0.6; Spearman's $\rho$: 0.6; Kendall's $\tau$: 0.51) and computational efficiency (Dasgupta et al., 2024). PerSEval measures how well a summarization model personalizes its outputs to individual user preferences (responsiveness) while also penalizing it for poor or inconsistent accuracy across users. It balances the trade-off between generating diverse, user-specific summaries and maintaining relevance to the expected content. The detailed formulation is described in Appendix A.

---

[7]We are yet to receive the codebase from the authors.

| Method | NAML | | | | EBNR | | | | NRMS | | | | TrRMIo (ft) | | | |
|---|---|---|---|---|---|---|---|---|---|---|---|---|---|---|---|---|
| | AUC | MRR | nDCG@5 | nDCG@10 | AUC | MRR | nDCG@5 | nDCG@10 | AUC | MRR | nDCG@5 | nDCG@10 | AUC | MRR | nDCG@5 | nDCG@10 |
| PENS(Original) | 0.48 | 0.74 | 0.81 | 0.81 | 0.45 | 0.72 | 0.77 | 0.77 | 0.47 | 0.73 | 0.80 | 0.80 | 0.47 | 0.7 | 0.78 | 0.78 |
| PENS-SH | 0.48 | 0.67 | 0.79 | 0.79 | 0.46 | 0.68 | 0.79 | 0.79 | 0.48 | 0.72 | 0.81 | 0.81 | 0.62 | 0.89 | 0.94 | 0.94 |
| S3 | 0.47 | 0.71 | 0.79 | 0.79 | 0.46 | 0.69 | 0.78 | 0.78 | 0.47 | 0.70 | 0.81 | 0.81 | 0.51 | 0.75 | 0.8 | 0.8 |
| SDAInter | 0.53 | 0.78 | 0.83 | 0.83 | 0.51 | 0.75 | 0.79 | 0.79 | 0.52 | 0.77 | 0.83 | 0.83 | 0.56 | 0.87 | 0.95 | 0.95 |
| LLaMA2(13B) | 0.43 | 0.62 | 0.68 | 0.68 | 0.46 | 0.68 | 0.74 | 0.74 | 0.41 | 0.53 | 0.58 | 0.58 | 0.42 | 0.67 | 0.73 | 0.73 |
| Mistral(7B) | 0.45 | 0.64 | 0.71 | 0.71 | 0.48 | 0.70 | 0.74 | 0.74 | 0.45 | 0.59 | 0.68 | 0.68 | 0.56 | 0.73 | 0.76 | 0.76 |
| DeepSeek-R1 | 0.43 | 0.61 | 0.65 | 0.65 | 0.45 | 0.64 | 0.72 | 0.72 | 0.44 | 0.54 | 0.65 | 0.65 | 0.47 | 0.69 | 0.77 | 0.77 |
| **PerAugy DS(ours)** | _0.57_ | _0.78_ | _0.84_ | _0.84_ | _0.54_ | _0.77_ | _0.84_ | _0.84_ | _0.55_ | _0.78_ | _0.83_ | _0.83_ | _0.74_ | _0.9_ | _0.95_ | _0.95_ |
| **PerAugy DS+SMP(ours)** | **0.59** | **0.79** | **0.87** | **0.87** | **0.59** | **0.81** | **0.88** | **0.88** | **0.59** | **0.83** | **0.86** | **0.86** | **0.76** | **0.91** | **0.97** | **0.97** |

Table 2: **User encoder performance (*trained-from-scratch*):** Models trained on PENS and its augmented variants, including PerAugy (DS+SMP). **Observation-1:** PerAugy *outperforms all baselines across models (NAML, EBNR, NRMS) and metrics (AUC, MRR, nDCG@5/10), when trained-from-scratch.* **Observation-2:** *When finetuned on TrRMIo,* PerAugy *consistently outperforms all baseline augmentation strategies, as compared to their fine-tuned versions.* **Observation-3:** *While some methods (e.g., SDAInter) help, others (e.g., S3, LLaMA2) degrade performance, showing the impact of augmentation and UIG quality.*

### 5.2.3 Human-Judgment based Evaluation

Direct human evaluation of personalized summarization faces fundamental feasibility issues. A third-party annotator cannot reasonably adopt the evolving preferences of a target user after parsing through extensive and often noisy interaction histories, ranging from raw click headlines and skipped articles to long Reddit threads. Personalization hinges on nuanced, longitudinal signals such as shifting stances, sub-topic interests, and stylistic inclinations, which are often subtle and subjective. Any attempt to reduce this complexity into a simplified abstraction risks erasing the contributions of more expressive personalization models that can capture temporal shifts in preference history, collapsing the goal of personalized summarization to persona-centric (static) summarization (as described in Section 2.1). Thus, to assess the cognitive validity of PerSEval, Dasgupta et al. (2024) designed a survey-based meta-evaluation simulating how human evaluators perceive personalization. Participants rated the similarity of summary pairs (model-generated and gold-reference) without knowing their source. From these ratings, they constructed DEGRESS-HJ (a human-judged version of DEGRESS) using normalized similarity as divergence and compared it against DEGRESS using correlation metrics (Pearsons $r$, Spearmans $\rho$, Kendalls $\tau$), and further evaluated whether applying standard accuracy metrics as discounting factors over DEGRESS-HJ (mimicking EDP) aligns with PerSEval scores. Strong correlations in both stages confirm that *human evaluators intuitively align with PerSEval's ratio-based responsiveness and factor-based accuracy penalty* indicating that PerSEval has strong human-judgment validity and does not require further human evaluation in this setup.

## 6 PerAugy Performance Results and Insights

In this section, we discuss the results of each of the research questions outlined in Section 5.

### 6.1 RQ-1: PerAugy's Effect on User-Encoder Accuracy

**Comparison with SOTA augmentation strategy.** We observe a *significant improvement in accuracy* across all the trained-from-scratch baseline encoders[8] when trained on our proposed dataset $\mathcal{T}^{\mathcal{E}-\mathrm{P}}_{\mathrm{DS+SMP}}$, compared to their original performance using the standard preference training set $\mathcal{T}^{P}_{\mathrm{train}}$. The best results obtained using PerAugy show notable relative gains of 0.139 ↑ in AUC, 0.108 ↑ in nDCG@5/10 (on the EBNR encoder), and 0.096 ↑ in MRR (on the NRMS encoder), clearly demonstrating that the PerAugy augmented dataset can effectively *substitute for scarce preference training data*. We also find that PerAugy significantly outperforms all baseline augmentation methods (S3, PENS-SH, and SDAInter) when the user encoders (NAML, EBNR, and NRMS) are trained-from-scratch. Specifically, we observe average gains of 0.127 ↑, 0.143 ↑, and 0.09 ↑ over S3, 0.117 ↑, 0.12 ↑, and 0.07 ↑ over PENS-SH, and 0.103 ↑, 0.09 ↑, and 0.067 ↑ over SDAInter in terms of AUC, MRR, and nDCG@5/10 respectively. $T^{\mathcal{E}-\mathrm{P}}_{\mathrm{DS+SMP}}$ upgrades performance w.r.t. $\mathcal{T}^{\mathcal{E}-\mathrm{P}}_{\mathrm{DS}}$ across all metrics throughout the user encoders (average boost of 0.03 ↑ in AUC, nDCG@5&10, and 0.02 ↑ in MRR),

---

[8]All results are statistically significant with $p < 0.01$ (test size: 18.1K positive and negative decision nodes).

| Personalized Summarizer | User-Encoder | PENS (Original) | DS (Scratch) | DS+SMP (Scratch) | DS+SMP (Encoder FT / End-to-End) |
|---|---|---|---|---|---|
| **PENS** | NAML (T1) | 0.013 | 0.008 | 0.014 | 0.015 / 0.017 |
| | EBNR (T1) | 0.008 | 0.009 | 0.010 | 0.010 / *NA* |
| | EBNR (T2) | 0.005 | 0.008 | 0.008 | 0.010 / 0.009 |
| | NRMS (T1) | 0.011 | 0.008 | 0.010 | 0.011 / *NA* |
| | NRMS (T2) | 0.004 | 0.007 | 0.007 | 0.005 / *NA* |
| **GTP** | TrRMIo (title) | 0.006 | 0.017 | **0.017** | 0.020 / 0.022 |

Table 3: **Personalized Summarizer Performance (PSE-SU4).** All models are injected with the same user-encoder used during their original training. **Observation-1:** *GTP utilizes improved user embeddings best in both encoder finetuning and end-to-end tuning.* **Observation-2:** *In PENS, NAML-T1 shows a clear boost, while other variants fail to capitalize on the finetuned encoder.* **Observation-3:** *SMP consistently improves over DS, indicating its necessity for maximizing gains.*

establishing that *Double-Shuffling alone is not sufficient to yield significant performance lift*( Table 2). ablate on the mixed training data $\mathcal{T}_{\text{DS}}^{\mathcal{E}-\text{P}}$ to analyze the effect of DS hyper-parameters– gap-length $g_l$ (section 5.1.1) and train history-segment length $\tau_{h_{\text{train}}}$: $\{l/2, 5l/8, 3l/4, 7l/8, l-3\}$ ($l$: trajectory length). For SMP hyper-parameters ($k$: $\{10, 15, 20\}, \lambda$: $\{0.3, 0.8, 1\}, p_{\text{SMP}}$: $\{0.5, 0.8, 1\}$), we ablate on $\mathcal{T}_{\text{DS+SMP}}^{\mathcal{E}-\text{P}}$. Results are in Appendix I.1; Figure 6.

**Comparison with LLM-generated Train sets.** Furthermore, to assess the effectiveness in comparison to LLM-generated training data, we train the same user encoders from scratch using $\mathcal{T}_{\text{train}}^{\text{LLaMA-2}}$, $\mathcal{T}_{\text{train}}^{\text{Mistral}}$, and $\mathcal{T}_{\text{train}}^{\text{DeepSeek-R1}}$. In comparison to these LLM-generated baselines, our PerAugy training set $\mathcal{T}_{\text{DS+SMP}}^{\mathcal{E}-\text{P}}$ yields average performance gains of 0.157 ↑, 0.2 ↑, and 0.203 ↑ over LLaMA2; 0.13 ↑, 0.167 ↑, and 0.16 ↑ over Mistral; and 0.15 ↑, 0.213 ↑, and 0.197 ↑ over DeepSeek-R1 – again reported in terms of AUC, MRR, and nDCG@5/10 respectively. It is important to understand that LLMs have limited capacity to generate a large number of diverse trajectories, and thus we scale down to $800-1500$ unique trajectories generated. These consistent improvements across all metrics and models further validate the effectiveness of the PerAugy as an augmentation strategy (For details, see Table 2).

**Effect of Finetuning.** We reserve TrRMIo to analyze the fine-tuning performance of PerAugy and find it to outperform the best performing PENS-SH-based fine-tuning (0.14 ↑ w.r.t AUC, 0.034 ↑ w.r.t MRR, & 0.026 ↑ w.r.t nDCG@5/10) (Results in Table 2).

**Cross-domain Study.** When trained on $\mathcal{T}_{\text{DS/DS+SMP}}^{\mathcal{E}-\text{OAI}}$ and evaluated on the PENS test set $\mathcal{T}_{\text{test}}$, we observe that PerAugy reliably applied to OpenAI (Reddit) like datasets that do not contain preference histories (best (NRMS): 0.163 ↑ w.r.t AUC, 0.112 ↑ w.r.t MRR, 0.09↑ w.r.t nDCG@5/10). Importantly, our primary claim is of *cross-domain generalizability* and not *transferability*. By generalizability, we mean that PerAugy can be trained on datasets (as trained on $\mathcal{T}_{\text{DS/DS+SMP}}^{\mathcal{E}-\text{OAI}}$) that are very different in domain and structure (for instance, is a **non-news, multi-domain** dataset), and still sustain its boosting performance on their corresponding test sets, while testing the $\mathcal{T}_{\text{DS/DS+SMP}}^{\mathcal{E}-\text{OAI}}$-trained encoders on $\mathcal{T}_{\text{test}}$ sufficiently validates cross-domain transferability. The results are detailed in Figure 8.

## 6.2 RQ-2: PerAugy's Effect on Personalization

We examine how effectively the baseline personalized summarization frameworks, GTP and PENS, leverage their corresponding improved user encoders, focusing on the frameworks' sensitivity to enhanced user history embeddings. To isolate the contribution of user encoder improvements, we first replace the fine-tuned encoders with their *train-from-scratch* counterparts. This setup helps us assess the raw effect of our augmentation strategy, PerAugy. With DS, the highest improvement in the PSE-SU4 score is for the GTP framework (+TrRMIo), achieving a gain of 0.012 ↑ over its baseline (original PSE-SU4: 0.006). On the PENS framework, the PENS(+EBNR+T2) variant shows a notable performance increase of 0.003↑ under DS. PENS variants that utilize NAML and NRMS encoders with T1 injection appear to benefit more from the SMP augmentation, yielding additional gains of 0.006↑ and 0.002↑ respectivelyon top of what DS alone provides. This further establishes that *Double Shuffling alone is not sufficient to boost downstream per-*

*formances.* When we switch to using the fine-tuned versions of the user encoders, the improvements in PSE-SU4 become more pronounced across models. The best results with these encoders include a boost of 0.014 ↑ for GTP+TrRMIo, 0.005 ↑ for PENS+EBNR-T2, and 0.002 ↑ for PENS+NAML-T1. Finally, to evaluate the full effect of end-to-end fine-tuning of the summarization frameworks with `PerAugy`, we utilize NAML(T1), EBNR(T2), and TrRMIo (since these encoders had shown relatively greater improvements in their fine-tuned versions). This end-to-end training leads to further performance boosts for NAML(T1) and GTP, both gaining 0.002 ↑, whereas EBNR(T2) shows a slight performance decline of 0.002 ↓ (Table 3). This indicates that the architecture and the injection method play a major role. This is evident from the best-performing GTP, which is trained with the *additional loss on aligning generated summary embedding with the history embedding* by the TrRMIo user-encoder. This results in superior personalized summaries, while the PENS framework falls behind due to a lack of alignment with the user's preference history.

## 7 Dataset Diversity Boosts Performance: Quantitative Analysis

We observe that DS and SMP operations lead to significantly higher performance of the SOTA user encoders (and thereby, the personalized summarization frameworks) when trained on `PerAugy`-generated datasets (see Tables 2 & 3). This empirically confirms our hypothesis that "*trajectory diversity of training data is directly proportional to personalization capabilities*". In this section, we further analyze the extent to which `PerAugy` achieves diversity via the DS and subsequent SMP operations w.r.t. three different **post-hoc diagnostic** diversity metrics. We then show that these metrics have high positive correlation with the accuracy metrics, thereby hinting (as a part of a *preliminary study*) that such metrics may be reliably used to estimate how good the training data (both real and synthetic) might be for the personalized summarization task. *The analysis outlined is a strongly suggestive verification method to show that diversity is the primary cause of the performance boost.*

### 7.1 Trajectory Diversity Metrics

In order to analyze the extent of diversity in `PerAugy`-generated datasets, we first define three diversity metrics – Topics per Trajectory (TP), Rate of Topic Change per Trajectory (RTC), and Degree-of-Diversity (`DegreeD`) as follows:

**Topics per Trajectory (TP).** TP is a simple trajectory-level metric that is commonly used to quantify topical variety and diversity in userinteraction graphs (UIGs):

$$\text{TP}(\tau^{u_j}) \;=\; \left| \left\{ \texttt{topic}\left( d^{(t_i)} \right) \right\}_{i=1}^l \right| ; \text{TP}(\mathcal{D}) \;=\; \frac{1}{|\mathbf{U}|} \sum_{j=1}^{|\mathbf{U}|} \text{TP}(\tau^{u_j}) \tag{2}$$

For a user trajectory $\tau^{u_j}$ with length $l$, where $d^{(t_i)}$ is the $i$-th document (d-node) consumed at time $t_i$, and let $\texttt{topic}(\cdot)$ map $d^{(t_i)}$ to its discrete topic (given as ground-truth). Higher TP indicates that a user engages with a wider variety of unique topics over time. However, TP alone cannot distinguish between *drift* and *diffusion*. A trajectory where topics change gradually over time (drift) and one where topics switch abruptly in a scattered manner (diffusion) can yield the same TP score, since TP only counts distinct topics. This limitation motivates the need for a complementary metric that accounts for the *frequency* of topical shifts.

**Rate of Topic Change per Trajectory (RTC).** This captures how frequently the topic changes between consecutive steps:

$$\text{RTC}(\tau^{u_j}) \;=\; \frac{1}{l-1} \sum_{i=1}^{l-1} \mathbf{1}\left[ \texttt{topic}\left( d^{(t_i)} \right) \neq \texttt{topic}\left( d^{(t_{i+1})} \right) \right] ; \text{RTC}(\mathcal{D}) \;=\; \frac{1}{|\mathbf{U}|} \sum_{j=1}^{|\mathbf{U}|} \text{RTC}(\tau^{u_j}), \tag{3}$$

where $\mathbf{1}[\cdot]$ is the indicator function and $|\mathbf{U}|$ is the number of user trajectories in dataset $\mathcal{D}$. A higher RTC indicates more frequent topic switching within a trajectory. By construction, $\text{RTC}(\tau^{u_j}) = 1$ if every successive pair of interactions involves different topics.

While TP and RTC provide fast, interpretable signals of diverse topics and frequent shifts of users' interest over timesteps, they do not capture several aspects crucial to a personalized summarization dataset: (i) *Faithfulness of s-node to source d-node:* they ignore how closely each subjective expected summary (s-node) aligns with the central discourse of the corresponding d-node. (ii) *Magnitude of thematic shifts:* a change of topic label is treated equally regardless of the actual semantic distance between the topics – i.e., small and large shifts are indistinguishable. (iii) *Consistency of user focus:* they do not penalize cases where s-nodes drift away from the core content of the corresponding d-nodes across time, i.e., when the faithfulness decreases. (iv) *Sparsity of s-nodes:* practical UIGs often lack s-nodes at many time-steps; these metrics offer no principled way to account for the lack of gold-reference s-nodes in training data. Also, RTC is insensitive to the *uniqueness and breadth of topical transitions*: a trajectory that merely switches with high frequency between a small subset of topics (periodic alternation) can yield a high RTC score, even though the topical coverage remains narrow. These limitations motivate a metric that jointly captures over (a) the *relative* alignment between documentdocument and summarysummary shifts, (b) the *absolute* thematic divergence, and (c) changes in *faithfulness* over time.

**Degree-of-Diversity (DegreeD).** To address the above drawbacks, we propose a novel metric called DegreeD (***Degree*-of-*D***iversity*) and then briefly discuss the method to compute it given any UIG. As a building block, we first define ***Degree*-of-*P***reference-*S***hift** (DePS) in a given UIG trajectory $\tau^{u_j}$ corresponding user $u_j$, where DePS quantifies the shift in a user's interest across $\tau^{u_j}$.

**Definition 2** (**Degree of Preference Shift (DePS)**). DePS *is the ratio of the thematic divergence between consecutive d-nodes $d^{(t_i)}$ and $d^{(t_{i+1})}$ (denoted as $\delta[X]_d$) over a time span (or interval) $\Delta_{(t_i, t_{i+1})} = t_{i+1} - t_i$ and that between the corresponding s-nodes (user's subjective expected summaries) $s_j^{(t_i)}$ and $s_j^{(t_{i+1})}$ (denoted as $\delta[X]_s$).*

The thematic divergence $\delta[X]_\bullet$ ($\bullet \in \{d, s\}$) is calculated as $\sigma(\bullet_j^{t_i}, \bullet_j^{t_{i+1}})$ where $\sigma$ is a distance measure on a chosen metric space. As per the definition, DePS for the $j$-th user at any unit interval $\Delta_{(t_i, t_{i+1})}$ is:

$$\text{DePS}_j^{\Delta_{(t_i, t_{i+1})}} = \frac{min(\delta[X]_d, \delta[X]_{s_j}) + \epsilon}{max(\delta[X]_d, \delta[X]_{s_j}) + \epsilon} \tag{4}$$

The Expected DePS, $\mathbb{E}_j[\text{DePS}]$, for $j$-th user over the trajectory $\tau^{u_j}$ having length $l$ is:

$$\mathbb{E}_j[\text{DePS}] = \frac{1}{l-1} \sum_{i=1}^{l-1} \text{DePS}_j^{\Delta_{(t_i, t_{i+1})}} \tag{5}$$

$\mathbb{E}_j[\text{DePS}]$ penalizes the disproportionate alignment between the expected summaries ($\delta[X]_{s_j}$) and the document divergence ($\delta[X]_d$) for $\tau^{u_j}$ (Figure 4a (a)). We adopt the minmax normalization in Eq. 4 to ensure that DePS is scale-invariant across embedding spaces while remaining naturally bounded in $[0, 1]$, providing a consistent measure of alignment between document and summary divergences. Consider that at $t_1$, Alice clicks on "*Yoga retreat in Bali*" ($d^{(t_1)}$) and her expected summary is "*Yoga travel guide*" ($s^{(t_1)}$), which is well aligned. At $t_2$, the document shifts to "*Top 10 meditation apps*" ($d^{(t_2)}$), but her summary remains almost unchanged as "*Yoga travel blogs*" ($s^{(t_2)}$). Here, the document divergence $\delta[X]_d$ is non-trivial (topic shifted within wellness), while the summary divergence $\delta[X]_s$ is very less. This disproportion leads to a low $\text{DePS}^{\Delta_{(t_1, t_2)}}$, signaling that Alice's summaries are not faithfully tracking her document-level preference shifts. However, it fails to penalize the case when any expected summary (s-node) in $\tau^{u_j}$ is not consistently faithful (in the sense of centrality to the core topic) to the corresponding document (d-node) (Figure 4a (b)). It also fails to penalize the case when the absolute thematic divergence is small (Figure 4a (c)). Both these cases can happen even with a high $\mathbb{E}_j[\text{DePS}]$. To address the first issue, we modify $\mathbb{E}_j[\text{DePS}]$ to incorporate the necessary penalties (i.e, penalized $\mathbb{E}_j[\text{DePS}]$ or $\mathbb{E}_j[\text{DePS}^{\mathcal{P}}]$) as follows:

$$\mathbb{E}_j[\text{DePS}^{\mathcal{P}}] = \frac{\sum_{i=1}^{l-1} \left( \text{DePS}_j^{\Delta_{(t_i, t_{i+1})}} \cdot \frac{\sigma(d^{(t_i)}, s_j^{(t_i)})}{\sigma(d^{(t_{i+1})}, s_j^{(t_{i+1})})} \right)}{l-1} \tag{6}$$

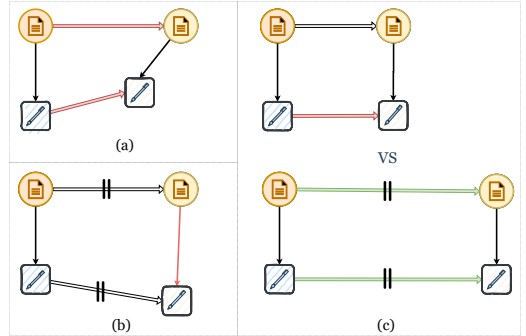

| Symbol | Description |
|--------|-------------|
| $\text{TP}(\tau^{u_j})$ | Unique topics in trajectory $\tau^{u_j}$. |
| $\text{RTC}(\tau^{u_j})$ | Rate of topic change in $\tau^{u_j}$. |
| $\delta[X]_d$ | Thematic Divergence between consecutive documents. |
| $\delta[X]_{s_j}$ | Thematic Divergence between consecutive summaries for user $j$. |
| $\Delta(t_i, t_{i+1})$ | A unit time-span. |
| $\sigma$ | Divergence metric in embedding space. |
| $\mathbb{E}_j[\delta]$ | Expected preference shift across $\tau^{u_j}$. |
| $\mathbb{E}_j[\delta^{\mathcal{P}}]$ | Penalized expected preference shift for $j$. |
| $\alpha$ | Regulator that controls the influence of Penalized DePS. |
| $|\mathbf{U}|$ | Number of trajectories in the dataset. |

(a) **DegreeD** penalization: (a) *Disproportionality* between successive s-node and d-node divergences, (b) *Proportionate but* **unfaithful** *alignment of s-node & d-node*, (c) **Lack of thematic divergence** *between consecutive d-s pairs.*

(b) **Symbols used in the definition of TP, RTC, and DegreeD.**

In the above equation, the second factor penalizes *negative shift* in faithfulness where the s-node starts deviating away from the corresponding d-node as compared to the deviation at previous time-steps. Note that the **second factor rewards** *positive shift* **where s-node comes closer to its corresponding d-node**. As can instance, consider at $t_1$, Alice clicks "*Yoga retreat in Bali*" ($d^{(t_1)}$) and her expected summary is "*Yoga travel guide*" ($s^{(t_1)}$). At $t_2$, she clicks "*Top 10 meditation apps*" ($d^{(t_2)}$), but her expected summary becomes "*App technology and efficiency*" ($s^{(t_2)}$). The two documents are semantically close (both wellness-related), but the new s-node drifts away, showing inconsistency. $\mathbb{E}_j[\text{DePS}]$ would still look high because of normalization, but the penalized $\mathbb{E}_j[\text{DePS}^{\mathcal{P}}]$ detects this faithfulness drop and penalizes it. To address the second issue, we inject an additional factor $\delta[X]_{s_j}$ that penalizes lack of thematic divergence in terms of user's actual interest/focus (hence, $\delta[X]_{s_j}$ instead of $\delta[X]_{d_j}$). $\delta[X]_{s_j}$ is regulated by the hyper-parameter $\alpha = (0, 1]$. The final DegreeD formulation for a dataset $\mathcal{D}$ containing $|\mathbf{U}|$ unique user trajectories is:

$$\text{DegreeD}(\mathcal{D}) = \frac{\alpha}{|\mathbf{U}|} \cdot \sum_{j=1}^{|\mathbf{U}|} \delta[X]_{s_j} \cdot \mathbb{E}_j[\text{DePS}^{\mathcal{P}}] \tag{7}$$

To continue illustration from the last example itself, consider at $t_3$, Alice clicks "*Meditation workshop details*" ($d^{(t_3)}$), and her expected summary is "*Workshop information*" ($s^{(t_3)}$). Both $d^{(t_2)}$ and $d^{(t_3)}$ are very similar, and the corresponding summaries are almost identical. Here, $\mathbb{E}_j[\text{DePS}^{\mathcal{P}}]$ might appear high since s-nodes are faithful towards their respective d-nodes, along with being proportionate. But $\delta[X]_{s_j}$ penalizes such low divergence in user interest, ensuring that datasets with repetitive, uninformative shifts are not overvalued. A dataset $\mathcal{D}$ is suitable for personalization training if it has a high DegreeD score. Mathematical proof of robustness of DegreeD under a variant choice of $\sigma$ is given in Appendix F.

## 7.2 Effect of Augmentation on Dataset Diversity

We analyze PerAugy's effect on dataset diversity. We observe that $\mathcal{T}_{\text{DS+SMP}}^{\mathcal{E}-P}$ achieves a boost of $4.2/0.09/0.28 \uparrow$ w.r.t TP/RTC/DegreeD compared to the PENS original dataset and a boost of $3.2/0.30/0.113 \uparrow$ w.r.t TP/RTC/DegreeD compared to the OAI original dataset (Table 4). We ablate the effect of the hyperparameters of PerAugy on DegreeD, including gap length $g_l$ and trajectory length $l$ for $\mathcal{T}_{\text{DS}}^{\mathcal{E}-\text{P}}$, as well as context length $k$, decay constant $\lambda$, and perturbation probability $p_{SMP}$ for $\mathcal{T}_{\text{DS+SMP}}^{\mathcal{E}-\text{P}}$ (see Appendix I.2, Figure 7).

## 7.3 Dataset Diversity Metrics as a Potential Predictor of Performance Gain

In order to draw a conclusive statement on dataset diversity being a primary cause of performance gains across models, we first have to establish the reliability and stability of the three diversity metrics, along with

| Augmentation Baselines | TP | RTC | DegreeD |
|---|---|---|---|
| PENS (original) | 7.3 | 0.56 | 0.009 |
| PENS-SH† | 7.4 | 0.54 | 0.067 |
| S3† | 7.3 | 0.56 | 0.019 |
| SDAInter† | 7.8 | 0.63 | 0.083 |
| LLaMA-2-13B† | 8.8 | 0.61 | 0.113 |
| Mistral-7B† | 8.6 | 0.63 | 0.144 |
| DeepSeek-R1† | *9.4* | *0.68* | 0.219 |
| PerAugy DS† | **13.6** | **0.77** | *0.232* |
| PerAugy DS+SMP† | **13.6** | **0.77** | **0.289** |
| OpenAI (Reddit) (OAI) | 8.5 | 0.42 | 0.008 |
| PerAugy-OAI⋆ | 11.7 | 0.72 | 0.121 |

Table 4: **Diversity analysis:** Comparison of DegreeD, TP, and RTC across datasets. † indicates augmentation followed by UIG abstraction on PENS, and ⋆ indicates augmentation on seed OpenAI(Reddit). **Observation-1:** PerAugy *shows higher diversity in terms all 3 metrics than its seeds and augmentation baselines.* **Observation-2:** *While* TP *and* RTC *relate to diversity, they fall short in capturing preference shifts effectively (e.g. PENS-SH has lower* RTC *than PENS, although it leads to higher user-encoder accuracy; see Table 2).***Observation-3:** PerAugy *DS and* PerAugy *DS+SMP yields same across* TP *and* RTC *because they are agnostic of the inserted s-nodes, while SMP changes s-node contents only on the top of DS, thereby making* DegreeD *a better evaluator of the pertubation w.r.t. diversity.*

| Diversity Metric Pair | Pearson | Spearman | Kendall |
|---|---|---|---|
| TP vs RTC | 0.75 | 0.78 | 0.62 |
| TP vs DegreeD | 0.79 | 0.81 | 0.67 |
| RTC vs DegreeD | **0.84** | **0.90** | **0.75** |

Table 5: **Inter-correlation (Pearson $r$, Spearman $\rho$, Kendalls $\tau$) between Diversity Metrics:** **Observation-1:** DegreeD *is effective diversity evaluation metric that gives additional insights, which are not captured by* TP *and* RTC; **Observation-2:** DegreeD *has high compatibility with text-based diversity evaluation metrics;* **Observation-3:** RTC *exhibits higher correlation than* TP *across all datasets, which suggests that* RTC *might fail to capture true diversification of sequence, and misevaluate a frequently shifting trajectory limited to fewer topics.*

whether they have inter-metric alignment. In this section, we first check whether the three metrics align with each other, and then show the meta-evaluation of diversity metrics to establish reliability.

### 7.3.1 Inter-metric Alignment

We also see a positive correlation between the diversity metrics within themselves, with an average correlation of 0.7 between TP and RTC, 0.75 between TP and DegreeD, and 0.8 between DegreeD and RTC, as in Table 5. This suggests that at a broader design level, DegreeD although being a more nuanced metric, is compatible with simpler diversity metrics (correlation computation details have been provided in Appendix H).

### 7.3.2 Meta-evaluation: Diversity Metric Reliability

As per the empirical evidences of Tables 2 and 3, dataset diversity should lead to an increase in accuracy. Therefore, *any diversity metric should be consistent with these empirical results.* Hence, we conduct the meta-evaluation of the diversity metrics w.r.t *reliability.* To this end, we compute the correlation of the dataset diversity (both original and synthetic generated by the augmentation methods) as shown in Table 4 with the average user-encoder accuracy across the encoder models when trained on these datasets[9]. We observe a strong positive correlation, thereby high reliability w.r.t consistency, for TP and DegreeD across all accuracy metrics (Pearson: 0.72/0.69, Spearman: 0.7/0.73, and Kendall: 0.53/0.58 w.r.t. nDCG), while RTC has low correlation (see Table 6). This indicates that RTC, which only focuses on the *absolute* frequency of topic shifts

---

[9]Since LLM-generated trajectories have been preprocessed to track unique sequences to address redundancies (thus making the number of trajectories quite low and yielding significantly lower user-encoder performance). For details of correlation measures, see Appendix H.

| Aggregate Mean Correlation of Diversity Metrics with User Encoder Performance | | | | |
|---|---|---|---|---|
| Diversity Metric | Eval Metric | Pearson $r$ (incl. PerAugy/ excl. PerAugy) | Spearman $\rho$ (incl. PerAugy/ excl. PerAugy) | Kendall's $\tau$ (incl. PerAugy/ excl. PerAugy) |
| TP | AUC | 0.606 / 0.543 ($\Delta = 0.063$) | 0.710 / 0.562 ($\Delta = 0.148$) | 0.518 / 0.461 ($\Delta = 0.057$) |
| | MRR | **0.765** / **0.686** ($\Delta = 0.079$) | 0.704 / 0.637 ($\Delta = 0.067$) | 0.542 / 0.531 ($\Delta = 0.011$) |
| | nDCG@5 | **0.719** / 0.612 ($\Delta = 0.107$) | 0.701 / 0.635 ($\Delta = 0.066$) | 0.529 / 0.424 ($\Delta = 0.105$) |
| | nDCG@10 | **0.719** / 0.612 ($\Delta = 0.107$) | 0.701 / 0.635 ($\Delta = 0.066$) | 0.529 / 0.424 ($\Delta = 0.105$) |
| RTC | AUC | 0.448 / 0.392 ($\Delta = 0.056$) | 0.243 / 0.201 ($\Delta = 0.042$) | 0.112 / 0.094 ($\Delta = 0.018$) |
| | MRR | 0.642 / 0.581 ($\Delta = 0.061$) | 0.251 / 0.213 ($\Delta = 0.038$) | 0.132 / 0.110 ($\Delta = 0.022$) |
| | nDCG@5 | 0.523 / 0.472 ($\Delta = 0.051$) | 0.234 / 0.196 ($\Delta = 0.038$) | 0.106 / 0.090 ($\Delta = 0.016$) |
| | nDCG@10 | 0.523 / 0.472 ($\Delta = 0.051$) | 0.234 / 0.196 ($\Delta = 0.038$) | 0.106 / 0.090 ($\Delta = 0.016$) |
| **DegreeD** | AUC | **0.682** / **0.645** ($\Delta = 0.037$) | **0.737** / **0.712** ($\Delta = 0.025$) | **0.578** / **0.556** ($\Delta = 0.022$) |
| | MRR | 0.672 / 0.624 ($\Delta = 0.048$) | **0.725** / **0.684** ($\Delta = 0.041$) | **0.552** / **0.537** ($\Delta = 0.015$) |
| | nDCG@5 | 0.687 / **0.663** ($\Delta = 0.024$) | **0.731** / **0.693** ($\Delta = 0.038$) | **0.580** / **0.566** ($\Delta = 0.014$) |
| | nDCG@10 | 0.687 / **0.663** ($\Delta = 0.024$) | **0.731** / **0.693** ($\Delta = 0.038$) | **0.580** / **0.566** ($\Delta = 0.014$) |

Table 6: **Meta-evaluation of Diversity Metrics:** Correlation of dataset diversity with encoder accuracy (averaged) **Observation-1 (Reliability):** DegreeD *consistently shows the strongest correlation, confirming it as the most reliable diversity indicator.* **Observation-2:** RTC *exhibits low correlation throughout, indicating that frequency of topic switching alone is insufficient for capturing personalization-relevant diversity.* **Observation-3 (Stability):** *Correlation of* TP *and* DegreeD *remains strong even after excluding high-performing* PerAugy*, thereby confirming that overall metric reliability is not inflated by* PerAugy*.*

| Correlation of TP and **DegreeD** with each user encoder | | | | | | | |
|---|---|---|---|---|---|---|---|
| Metric | Corr. | NAML (TP / DegreeD) | EBNR (TP / DegreeD) | NRMS (TP / DegreeD) | TrRMIo (TP / DegreeD) | Aggr. Mean Corr. (TP / DegreeD) | Variance (TP / DegreeD) |
| AUC | Pearson | 0.318 / 0.848 | 0.303 / 0.380 | 0.616 / 0.700 | 0.570 / 0.607 | 0.606/0.682 | 0.044 / 0.031 |
| | Spearman | 0.602 / 0.905 | 0.429 / 0.571 | 0.464 / 0.567 | 0.357 / 0.500 | 0.710/0.737 | 0.069 / 0.035 |
| | Kendall | 0.371 / 0.786 | 0.229 / 0.357 | 0.371 / 0.500 | 0.257 / 0.429 | 0.518/0.578 | 0.049 / 0.030 |
| MRR | Pearson | 0.167 / 0.643 | 0.100 / 0.239 | 0.341 / 0.499 | 0.245 / 0.576 | 0.765/0.672 | 0.313 / 0.057 |
| | Spearman | 0.265 / 0.667 | 0.153 / 0.262 | 0.357 / 0.433 | 0.267 / 0.643 | 0.704/0.725 | 0.202 / 0.077 |
| | Kendall | 0.148 / 0.500 | 0.095 / 0.143 | 0.190 / 0.286 | 0.143 / 0.476 | 0.542/0.552 | 0.16 / 0.062 |
| nDCG@5/10 | Pearson | 0.205 / 0.772 | 0.143 / 0.288 | 0.382 / 0.468 | 0.279 / 0.638 | 0.719/0.687 | 0.226 / 0.054 |
| | Spearman | 0.530 / 0.786 | 0.365 / 0.524 | 0.530 / 0.467 | 0.414 / 0.690 | 0.701/0.731 | 0.063 / 0.029 |
| | Kendall | 0.297 / 0.643 | 0.185 / 0.286 | 0.334 / 0.389 | 0.260 / 0.524 | 0.529/0.580 | 0.071 / 0.033 |

Table 7: **Stability of Diversity Metrics:** The sensitivity of TP and DegreeD to strong positive user-encoder outliers is analyzed via model-specific correlation variance w.r.t aggregate mean correlation (reported in Table 6). **Observation-1:** Overall, TP an DegreeD consistently has low variance across models; **Observation-2:** DegreeD has stronger stability w.r.t MRR (the strictest accuracy metric) than TP.

rather than their uniqueness, can incorrectly quantify a low diversity dataset as high. It is to be noted that beyond PerAugy, multiple baseline augmentations (e.g., SDAInter, PENS-SH) also raise encoder accuracy over the original PENS set, indicating that increased and well-aligned diversity can improve learning.

### 7.3.3 Meta-evaluation: Diversity Metric Stability

Having established that the diversity-accuracy relationship holds across all augmentation methods (not only PerAugy), we next test whether the strong correlation observed between dataset diversity and encoder performance is inflated by two potential sources – (i) PerAugy itself that acts as a strong outlier and dominates the correlation trend, and (ii) a specific strong positive user-encoder outlier whose unusually high accuracy performance disproportionately boosts the correlation values.

**Stability w.r.t. PerAugy as strong outlier:** There is a possibility that the high performance of PerAugy acts as a positive strong outlier. This can inflate the reliability correlation of TP and DegreeD thereby giving a misleading metric reliability. We perform the same experiment as in Table 6 but excluding PerAugy-generated datasets. We observe that the correlation of both TP and DegreeD do not vary and lie in the same high correlation band, underscoring that the diversityaccuracy trend is not because of PerAugy alone

(see Table 6; $\Delta$: absolute difference in correlation values; Mean values $\mu_{\text{TP}} : 0.082$, $\mu_{\text{DegreeD}} : 0.023$; Standard Deviations $\sigma_{\text{TP}} : 0.03$, $\sigma_{\text{DegreeD}} : 0.01$).

**Stability w.r.t. strong outlier user-encoder:** The previous analysis cannot detect whether a user-encoder acts as a strong positive outlier and inflates the aggregate correlation results reported. To address this, we find the correlation between dataset diversity (w.r.t TP and `DegreeD`) and individual user-encoder (i.e., NAML, EBNR, NRMS, TrRMIo) accuracy performances (w.r.t AUC, MRR, and nDCG metrics). We then compute the correlation variance (averaged across the user-encoders w.r.t the aggregated mean reported in Table 6). We observe that `DegreeD` consistently demonstrates lower variance than TP (average gain of $0.07 \uparrow$). Also, the results indicate that TP as a diversity metric is not a stable indicator of the role of the underlying dataset diversity under the stricter condition of MRR of encoders. In general, both metrics are stable w.r.t outlier models.

Although `DegreeD` is more stable and robust across models, it is relatively less interpretable and computationally heavier. TP, on the other hand, is simpler and interpretable but fails to capture the magnitude and direction of semantic shifts, making it insensitive to subtle preference drifts.

### 7.4 Data Diversity Causes Personalization Boost

Having empirically established the reliability and stability of TP and `DegreeD`, we can conclude that `PerAugy` reliably outperforms other baseline augmentation methods by a significant margin in terms of injecting diversity in the seed datasets (outperforms best (DeepSeek-R1) by $4.2/0.03 \uparrow$ w.r.t TP/`DegreeD`; Table 4). However, the user-encoder and the downstream model architecture matter, we find that NAML, NRMS, and TrRMIo being able to capture the diversity more, showing consistent correlation with diversity metrics.

## 8 Related Work

Prior works on data augmentation have largely focused on generative approaches for dialogue summarization (Liu et al., 2022; Ouyang et al., 2023; Park et al., 2024), and document-level augmentation for generalized summarization tasks (Fabbri et al., 2021; Chen et al., 2023b; Sahu et al., 2025). However, to the best of our knowledge, **preference-oriented data augmentation** in the *context of personalized summarization* remains significantly underexplored. The most relevant effort in this domain is PENS-SH (Song et al., 2023), which constructs synthetic user trajectories by identifying and merging common d-nodes across multiple user interaction graphs (UIGs). While effective at preserving shared preferences, PENS-SH fails to retain temporal order information due to the loss of time-step data, and entirely lacks intermediate summary nodes (s-nodes), leading to an incomplete representation of user intent and preference evolution. Broadly, preference data augmentation methods for sequential recommendation can be categorized into two classes: *intra-trajectory* and *cross-trajectory* augmentation. Most of these techniques use sequential recommendation datasets like Amazon, MovieLens, where the goal is next interaction prediction, not preference-based generation.

### 8.1 Intra-Trajectory Augmentation

Most intratrajectory methods perturb each users history by locally manipulating nodes or segments, but within the same trajectory. For example, MBASR (Xiao et al., 2024a) employs an intra-trajectory augmentation technique by performing pairwise swapping of segments to generate diversity. But on highly monotonous trajectories, this yields minimal change and may even inject unrealistic temporal transitions. STEAM (Lin et al., 2023) operates by deciding whether to drop or insert nodes within a trajectory to create augmented data. However, the method is not scalable to longer trajectories, and the insertion or deletion of nodes can disrupt the historical sequence or break the natural flow of interactions. L2Aug (Wang et al., 2022) learns a policy to delete nodes, but deletions disrupt continuity and remove potentially informative context. The Heuristic SAMPLER model (Chen et al., 2023c) replaces single nodes via popularity or cooccurrence-based heuristics, but such isolated swaps fail to shift the sequences overall engagement degree. ASReP (Liu et al., 2021) extends trajectories by generating pseudo-prior nodes using reverse pretraining. Although it aims to enrich the trajectory, any kind of insertion technique (or segment extension) can incorporate unrealistic synthetic nodes that compromise the authenticity of time-step information of further interactions,

making it unsuitable for our application. Finally, BTBR (Li et al., 2023) incorporates masking strategies and swapping operation to train the model for "Next Novel Basket Recommendation". But it does not generate a diverse input sequence to make the existing encoders learn the representations.

`PerAugy` departs from these schemes by applying controlled perturbations that operate at both micro and macroscales while preserving strict temporal coherence since intra-trajectory augmentations does not solve the problem of monotonous user history. Rather than swapping or deleting isolated nodes, we perform shuffling that incorporates controlled shifts and apply perturbation to diffusion. This approach polishes the sequenceenriching diversitywithout sacrificing realistic timestep information or overburdening the augmentation pipeline, making it scalable to long trajectories.

## 8.2 Cross-Trajectory Augmentation

Crosstrajectory techniques draw patterns across users to forge new sequences and augments multiple trajectories. DR4SR (Yin et al., 2024) uses a transformer to learn global sequence regeneration. The pertaining task is constructed for to extract patterns from given set of sequences and feed the patterns to the model to regenerate other set of possible sequences, but applying interchangeable patterns across subjective s-node summaries risks injecting generic behaviors that dilute personalization. TiCoSeRec (Dang et al., 2024) ensures uniform time-interval distribution in the sequence based on the time-aware traditional operations like Crop, Mask, Insert, Reorder and Substitute. But our trajectories inherently assume consistent unitstep timing, thereby making such timeaware edits redundant. FDA (Chen et al., 2023a) generates synthetic user profiles from the realistic profiles to balance between realistic data and pseudo data. But it generates monotonic complemented sequences that mirror existing repetition rather than diversifying it.divSPA (Liu et al., 2023) swaps segments between similar users based on similarity metrics. Similaritybased exchanges often leave the overall interaction degree unchanged and introduce context mismatches.

Instead of wholesale regeneration or arbitrary segment swaps, `PerAugy` leverages crosstrajectory substitution to adapt its perturbation parameters dynamically. By analyzing interuser variance in snode distributions, our method addresses optimal perturbation scales and target positions, ensuring that borrowed structure enhances diversity without compromising each trajectorys unique summarization points w.r.t. realistic userbehaviors. This yields augmented sequences that are both personalized and informationrich, overcoming the homogenization (or unrealistic heterogenization) pitfalls of existing crosstrajectory approaches.

## 9 Discussions & Limitations

`PerAugy` enhances the diversity and expressiveness of the original user history data while preserving personalization fidelity. This approach is particularly beneficial for improving model robustness and generalization in sparse or skewed datasets. In deployment settings, where user-written summaries are typically unavailable, model-generated summaries can serve as effective proxies, especially during cold-start scenarios such as a new browsing session, which we address via experiments on the OpenAI(Reddit) dataset. Looking forward, we see promising opportunities in extending this augmentation framework using large language models (LLMs). In particular, we are investigating prompt-tuning-based augmentation techniques that could generate more semantically rich and user-aligned variations of preference histories. Such methods hold the potential to be especially impactful in low-resource or non-PENS-like domains, where user signals are limited or noisy. Additionally, we aim to ground perturbation modeling in more principled stochastic processes. One such candidate is the Itô process, which incorporates both deterministic trends (drift) and random fluctuations (diffusion). Modeling user preference evolution through such continuous-time stochastic frameworks may offer a more realistic approximation of human behavior, allowing for fine-grained control over the intensity and direction of perturbations. This could open avenues for theoretically grounded, temporally aware augmentation strategies that better reflect user dynamics in real-world settings.

## 10 Conclusion

In this paper, we introduced `PerAugy`, a novel data augmentation technique designed to enhance the personalization capabilities of summarization models. By addressing limitations in current personalized datasets like PENS, `PerAugy` generates synthetic, diverse user interaction trajectories, reducing overfitting and improving generalization across domains. Techniques like Double Shuffling (DS) and Stochastic Markovian Perturbation (SMP) ensure that the augmented data remains realistic and coherent, enabling models to better align with individual user preferences. Our evaluation demonstrated significant improvements in the personalization metric PSE (an average of 61.2% boost), particularly in models like PENS+NRMS+T2, which achieved a 75% performance increase (PSE-RG-SU4). `PerAugy` also improved user-encoders such as NAML, EBNR, and NRMS, enhancing their ability to capture user preferences with average boosts of 24%, 25%, and 18% over baseline augmentations (w.r.t AUC, MRR, and nDCG@5&10). We further demonstrated its potential as a reliable generator of synthetic datasets in low-resource domains like OpenAI (Reddit), with encoder boosts of 19%, 25%, and 17% in the same metrics, broadening its applicability. While `PerAugy` is a critical advancement in addressing data scarcity and generalization in personalized summarization, future work will refine its techniques and explore adaptability to more models and architectures.

## Acknowledgment

T. Chakraborty would like to thank the support of the Rajiv Khemani Young Faculty Chair Professorship in AI. We would also like to thank Lightning.ai for GPU credits and Prof. Ankush Chander of DAU for helping the team with implementation details.

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

# A  Measuring Degree-of-Personalization

## A.1  Motivation

Vansh et al. (2023) proposed EGISES– a metric to measure the degree of **in**sensitivity-to-subjectivity for relative benchmarking of how much models *lack personalization* (i.e., a lower score is better within the range $[0,1]$) instead of assigning an absolute goodness score. Based on this notion, they defined (summary-level) "**deviation**" of a model $M_{\boldsymbol{\theta},u}$ (later termed as ***Deg**ree-of-**R**esponsiven**ess*** (DEGRESS) by Dasgupta et al. (2024)) as follows:

**Definition 3. *Summary-level* DEGRESS.** *Given a document $d_i$ and a user-profile $u_{ij}$ (user $j$'s expected summary), the summary-level responsiveness of a personalized model $M_{\boldsymbol{\theta},u}$, (i.e., DEGRESS$(s_{u_{ij}}|(d_i, u_{ij}))$), is defined as the proportional divergence between model-generated summary $s_{u_{ij}}$ of $d_i$ for $j$-th user from other user-specific summary versions w.r.t a corresponding divergence of $u_{ij}$ from the other user-profiles.*

DEGRESS$(s_{u_{ij}}|(d_i, u_{ij}))$ is formulated as:

$$\text{DEGRESS}(s_{u_{ij}}|(d_i, u_{ij})) = \frac{1}{|\mathbf{U}_{d_i}|} \sum_{k=1}^{|\mathbf{U}_{d_i}|} \frac{min(X_{ijk}, Y_{ijk}) + \epsilon}{max(X_{ijk}, Y_{ijk}) + \epsilon}$$

$$X_{ijk} = \frac{\exp(w(u_{ij}|u_{ik}))}{\sum\limits_{l=1}^{|\mathbf{U}_{d_i}|} \exp(w(u_{ij}|u_{il}))} \cdot \sigma(u_{ij}, u_{ik}); \ \ Y_{ijk} = \frac{\exp(w(s_{u_{ij}}|s_{u_{ik}}))}{\sum\limits_{l=1}^{|\mathbf{U}_{d_i}|} \exp(w(s_{u_{ij}}|s_{u_{il}}))} \cdot \sigma(s_{u_{ij}}, s_{u_{ik}}) \tag{8}$$

$$w(u_{ij}|u_{ik}) = \frac{\sigma(u_{ij}, u_{ik})}{\sigma(u_{ij}, d_i)}; \ \ w(s_{u_{ij}}|s_{u_{ik}}) = \frac{\sigma(s_{u_{ij}}, s_{u_{ik}})}{\sigma(s_{u_{ij}}, d_i)}$$

Here, $|\mathbf{D}|$ is the total number of documents in the evaluation dataset, $|\mathbf{U}|$ is the total number of users who created gold-reference summaries that reflect their expected summaries (and thereby, their subjective preferences), and $|\mathbf{U}_{d_i}| \ (= |\mathbf{S}_{d_i}|)$ is the number of users who created gold-references for document $d_i$. $w$ is the divergence of the model-generated summary $s_{u_{ij}}$ (and the corresponding expected summary $u_{ij}$) from document $d_i$ itself in comparison to all the other versions. It helps to determine how much percentage (therefore, the softmax function) of the divergence (i.e., $\sigma(s_{u_{ij}}, s_{u_{ik}})$ should be considered for the calculation of DEGRESS. If $s_{u_{ij}}$ is farther than $s_{u_{ik}}$ w.r.t $d_i$ then DEGRESS$(s_{u_{ij}}|(d_i, u_{ij})) <$ DEGRESS$(s_{u_{ik}}|(d_i, u_{ik}))$, implying that $M_{\boldsymbol{\theta},u}$ is more responsive to the $k$-th reader. A lower value of DEGRESS$(s_{u_{ij}}|(d_i, u_{ij}))$ indicates that while reader-profiles are different, the generated summary $s_{u_{ij}}$ is very similar to other reader-specific summaries (or vice versa), and hence, is not responsive at the summary-level. The system-level DEGRESS and EGISES have been formulated as follows:

$$\text{DEGRESS}(M_{\boldsymbol{\theta},u}) = \frac{\sum\limits_{i=1}^{|\mathbf{D}|} \dfrac{\sum\limits_{j=1}^{|\mathbf{U}_{d_i}|} \text{DEGRESS}(s_{u_{ij}}|(d_i, u_{ij}))}{|\mathbf{U}_{d_i}|}}{|\mathbf{D}|} \tag{9}$$

## A.2 `PerSEval`: Formulation

As can be noted, the **DEGRESS formualtion does not enforce any penalty on accuracy drop**. To rectify this Dasgupta et al. (2024) proposed `PerSEval`. The design of `PerSEval` had two key goals: (i) to penalize models for poor accuracy, while simultaneously (ii) ensuring that the evaluation of responsiveness (i.e., `DEGRESS`) is not overshadowed by high accuracy. This penalty is referred to as the *Effective DEGRESS Penalty Factor* (EDP). If a model achieves 100% accuracy, no EDP will be applied, and the `PerSEval` score will equal the `DEGRESS` score. The following formulatiown of `PerSEval` guarantees these properties:

$$\text{PerSEval}(s_{u_{ij}}|(d_i, u_{ij})) = \text{DEGRESS}(s_{u_{ij}}|(d_i, u_{ij})) \times \text{EDP}(s_{u_{ij}}|(d_i, u_{ij}))$$

$$\text{where, } \text{EDP}(s_{u_{ij}}|(d_i, u_{ij})) = 1 - \frac{1}{1 + 10^{\alpha \geq 3} \cdot \exp\left(-(10^{\beta \geq 1} \cdot (s_{u_{ij}}|(d_i, u_{ij})))\right)},$$

$$(s_{u_{ij}}|(d_i, u_{ij})) = \text{ADP}(s_{u_{i*}}|(d_i, u_{i*})) + \text{ACP}(s_{u_{ij}}|(d_i, u_{ij}))$$

(10)

Here, `ADP` is a document-level penalty due to a drop in accuracy for the best-performance of the model (i.e., the model-generated summary of document $d_i$ ($s_{u_{ij}}$) is closest to the corresponding reader's expected summary $u_{ij}$). `ADP` is formulated as follows:

$$\text{ADP}(s_{u_{i*}}|(d_i, u_{i*})) = \frac{1}{1 + 10^{\gamma \geq 4} \cdot \exp\left(-10 \cdot \frac{\sigma^*(s_{u_{i\bullet}}, u_{i\bullet})|d_i - \mathbf{0}}{(\mathbf{1} - \sigma^*(s_{u_{i\bullet}}, u_{i\bullet})|d_i) + \epsilon}\right)}$$

$$\text{where, } \sigma^*(s_{u_{i\bullet}}, u_{i\bullet})|d_i = \min_{j=1}^{|\mathbf{U}_{d_i}|} \sigma(s_{u_{ij}}, u_{ij})|d_i$$

$$\text{and } \{\epsilon : \text{An infinitesimally small number} \in (0, 1)\}$$

(11)

`ADP` ensures that even if the `DEGRESS` score is acceptable, a penalty due to accuracy drop can still be imposed as a part of EDP. `ADP`, however, fails to address the scenario where the best-case scenario is acceptable (i.e., accuracy is fairly high) but is rather an outlier case – i.e., for most of the other model-generated summary versions, there is a considerable accuracy drop. To address this issue, the second penalty component within EDP called *Accuracy-inconsistency Penalty* (`ACP`) was introduced which evaluates whether a model consistently performs w.r.t accuracy for a specific generated summary compared to its average performance. `ACP`is formulated as:

$$\text{ACP}(s_{u_{ij}}|(d_i, u_{ij})) = \frac{1}{1 + 10^{\gamma \geq 4} \cdot \exp\left(-10 \cdot \frac{\sigma(s_{u_{ij}}, u_{ij})|d_i - \sigma^*(s_{u_{i\bullet}}, u_{i\bullet})|d_i}{(\overline{\sigma}(s_{u_{i\bullet}}, u_{i\bullet})|d_i - \sigma^*(s_{u_{i\bullet}}, u_{i\bullet})|d_i) + \epsilon}\right)}$$

$$\text{where, } \overline{\sigma}(s_{u_{i\bullet}}, u_{i\bullet})|d_i = \frac{1}{|\mathbf{U}_{d_i}|} \sum_{j=1}^{|\mathbf{U}_{d_i}|} \sigma(s_{u_{ij}}, u_{ij})|d_i$$

(12)

The system-level `PerSEval` score is as follows:

$$\text{PerSEval}(M_{\boldsymbol{\theta}, u}) = \frac{\sum_{i=1}^{|\mathbf{D}|} \frac{\sum_{j=1}^{|\mathbf{U}_{d_i}|} \text{PerSEval}(s_{u_{ij}}|(d_i, u_{ij}))}{|\mathbf{U}_{d_i}|}}{|\mathbf{D}|}$$

(13)

The system-level `PerSEval` $\in [0, 1]$ and is bounded by the system-level `DEGRESS` score.

**PerSEval-RG-SU4.** (or PSE-SU4) is the `PerSEval` variant that uses ROUGE-SU4 (Lin, 2004) as a distance metric (i.e., $\sigma$) in the `PerSEval` formula. PSE-SU4 has been reported to have high human-judgment correlation (Pearson's $r$: 0.6; Spearman's $\rho$: 0.6; Kendall's $\tau$: 0.51) Dasgupta et al. (2024). The **ROUGE-SU4** score is based on *skip-bigrams*, which are pairs of words that appear in the same order within a sentence but can have up to four other words between them. The formula is as follows:

For a given generated summary $G$ and reference summary $R$, the ROUGE-SU4 score is calculated as:

**Skip-Bigram Recall ($R_{SU4}$):**

$$R_{\text{SU4}} = \frac{\text{Count of matching skip-bigrams between } G \text{ and } R}{\text{Total skip-bigrams in } R}$$

**Skip-Bigram Precision ($P_{SU4}$):**

$$P_{\text{SU4}} = \frac{\text{Count of matching skip-bigrams between } G \text{ and } R}{\text{Total skip-bigrams in } G}$$

**F1 Score ($F1_{SU4}$):** The F1 score is the harmonic mean of precision and recall:

$$F1_{\text{SU4}} = \frac{2 \times P_{\text{SU4}} \times R_{\text{SU4}}}{P_{\text{SU4}} + R_{\text{SU4}}}$$

Where:

- A **skip-bigram** consists of two words in the correct order but with zero to four words skipped in between.

- Matching skip-bigrams are counted between the generated summary and the reference summary.

The final **ROUGE-SU4** score is typically reported as the F1 measure, balancing precision and recall.

## B    Dataset and Statistics

**PENS**  The PENS dataset Ao et al. (2021) includes 113,762 news articles across 15 topics. Each article contains an ID, title (avg. 10.5 words), body (avg. 549 words), and category, with titles linked to the WikiData entities. The dataset also includes user interaction data, such as impressions and click behaviors, combined with news bodies and headlines from the MIND dataset Wu et al. (2020). For training, 500k user-news impressions were sampled from June 13 to July 3, 2019. Each log records user interaction as [uID, tmp, clkNews, uclkNews, clkedHis], where 'clkNews' and 'uclkNews' represent clicked and unclicked news, and 'clkedHis' refers to the user's prior clicked articles, sorted by click time. To create an offline testbed, 103 English-speaking students reviewed 1,000 headlines in stage-1, and then selected 50 articles, and created preferred headlines (i.e., expected gold-reference summaries) for 200 unseen articles in stage-2 (see Figure 5). Each article was reviewed by four participants. Editors checked for factual accuracy, discarding incorrect headlines. The high-quality remaining headlines serve as personalized gold-standard references in the PENS dataset. The PENS dataset has become the standard benchmark for personalized summarization task Ao et al. (2021); Song et al. (2023); Yang et al. (2023); Cai et al. (2023); Lian et al. (2025).The statistics of PENS dataset are given in Table 8.

**OpenAI (Reddit).**  The OpenAI (Reddit) dataset Völske et al. (2017) comprises 123,169 Reddit posts collected from 29 distinct subreddits. This dataset provides both OpenAI-generated and human-written summaries and is organized into two splits: Comparisons, used for training and validation, and Axis, designated for validation and testing. A curated subset of 1,038 posts was processed by 13 different summarization policies, resulting in the generation of 7,713 summaries. These summaries underwent evaluation by 64 annotators who rated paired summaries based on selection preferences, confidence in their ratings, and dimensions such as accuracy, coherence, coverage, and overall quality (see Table 9 for details). Notably, unlike datasets like PENS, these summaries are not linked to individual annotators or their reading histories, which means they lack elements of personalization and contextual user information. The detailed statistics are given in Table 9.

## C    Baselines

Here, we discuss in details the baseline augmentation strategies, user-encoders and summarization frameworks.

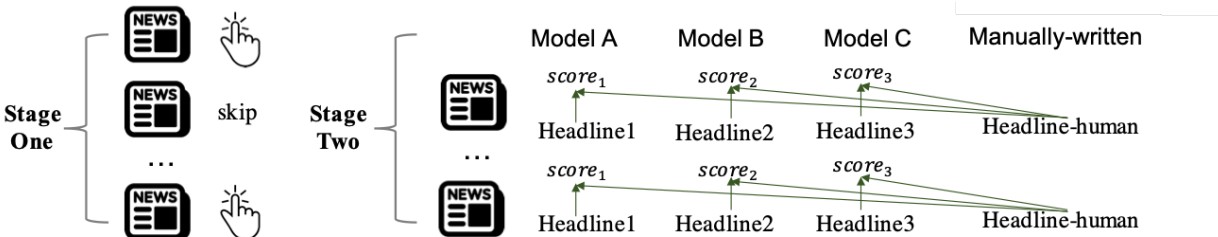

Figure 5: **Two stage PENS test data (original) creation** Ao et al. (2021): Stage 1 - Participants selected 50+ preferred headlines from 1,000 shown titles; Stage 2 - They rewrote headlines for 200 unseen articles using only news bodies, without seeing original titles.

| Characteristic | Dimension | Value |
|---|---|---|
| | **Article Stats** | |
| **General Stats** | # Topics | 15 |
| | # Articles | 113,762 |
| | Avg. Title Length | 10.5 words |
| | Avg. Body Length | 549 words |
| | **Train Dataset Statistics** | |
| **Interaction Data** | # UserNews Impressions (anon.) | 500,000 |
| | # Users (anon.) | 445,000 |
| | Time Period | June 13July 3, 2019 |
| | User Interaction Fields | [uID, tmp, clkNews, uclkNews, clkedHis] |
| | **Test Dataset Statistics** | |
| **Participant Stats** | # Participants | 103 |
| | Participant Category | Englishspeaking college students |
| | # Articles | 3,940 |
| | Browsed Headlines (Click + Skip) | 1,000 per participant |
| | Min. Interested (Click) Headlines | 50 per participant |
| **Gold Reference** | Summarized Article Bodies | 200 per participant |
| **(Participantwritten Headlines)** | Avg. Summaries per Article | 4 |

Table 8: **PENS dataset (original) statistics**: Here the '*clkNews*' and '*uclkNews*' indicate clicked and un-clicked (i.e., skipped) news; '*clkedHis*' refers to the user's prior clicked articles, all sorted by click time; news bodies and headlines sourced from the MIND dataset Wu et al. (2020); Test dataset is created in two stages (see Figure 5).

## C.1 Baseline Augmentations

We compare `PerAugy` with three SOTA algorithmic augmentation methods and three LLM-as-augmentors, and create corresponding UIGs for each of the augmentation methods. The statistical analysis of UIGs are given in Table 10. We describe each method as follows:

**PENS-SH.** We choose PENS-SH Song et al. (2023), a SOTA PENS-based synthetic data generator for personalized summarization, as a comparative baseline w.r.t user-encoder accuracy boost. PENS-SH merges multiple (say, $m$) seed UIG trajectories $\{\tau^{u_{j=1:m}}\}$ from the PENS train dataset ($\mathcal{T}^P_{\text{train}}$) into a single synthetic trajectory $\tau^{\hat{u}_{1:m}}_{\text{P-SH}}$ such that all the common d-nodes on $\tau^{\hat{u}_{1:m}}_{\text{P-SH}}$ are unique to it, thereby forming the pool $\mathcal{T}^{\text{PSH}}_{\text{train}}$. We analyze the diversity of the PENS-SH trajectories in $\mathcal{T}^{\text{PSH}}_{\text{train}}$ when injected with s-nodes from the PENS test dataset, denoted $\mathcal{T}^{\text{syn-PSH}}$ as UIG of PENS-SH, following Section 3.2.

**S3-Aug.** We choose S3 (Segment-Shuffle-Stitch) Grover et al. (2024), a modular neural *intra-trajectory* augmentation mechanism designed for sequential data, as a comparative baseline. S3 restructures user interaction sequences $\tau^u$ by dividing them into $n$ non-overlapping segments, followed by a differentiable

| Characteristic | Dimension | Value |
|---|---|---|
| **Dataset Overview** | | |
| **General Stats** | # Reddit Posts | 123,169 |
| | # Subreddits (Domains) | 29 |
| | Policy-Generated Summaries | 115,579 |
| | Human-Written Summaries | Available |
| **Train + Validation Dataset Statistics** | | |
| **Article Stats** | # Reddit Posts | 21,111 |
| | # Policies | 81 |
| | # Generated Summaries | 107,866 |
| | # Annotators | 76 |
| | # Summary-Pairs Rated | 64,832 |
| **Validation Subset Statistics** | | |
| **Subset Details** | # Reddit Posts | 1,038 |
| | # Policies | 13 |
| | # Generated Summaries | 7,713 |
| | # Annotators | 32 |
| **Test Dataset (RLHF-Tuned Policies) Statistics** | | |
| **Evaluation Stats** | # Evaluated Policies | 4 |
| | # Evaluated Reddit Posts | 57 (out of 1,038) |
| | Evaluation Method | Indirect Benchmarking |
| **Annotation and Feedback** | | |
| **Feedback Collection** | Rating Scale | 17 |
| | Confidence Scale | 19 |
| | Avg. Ratings per Annotator | 1,176 |
| | Annotation Format | Summary-Pairs Selection |

Table 9: **OpenAI TL;DR (Reddit) dataset statistics**: The dataset includes 123,169 Reddit posts across 29 subreddits, with policy-generated and human-written summaries. Evaluation involves summary-pair ratings and RLHF-tuned policy benchmarking.

shuffling and stitching operation on $\mathcal{T}_{\text{train}}^P$ to yield augmented trajectories $\tau_{\text{S3}}^{\hat{u}_{1:m}}$ that preserve local coherence while introducing temporal perturbations. The resulting S3-augmented trajectories $\mathcal{T}_{\text{train}}^{\text{S3}}$ are used to train user-encoders, and we evaluate its diversity after incorporating s-nodes from PENS testbed (denoting the UIG of it as $\mathcal{T}^{\text{syn-S3}}$) and further effect on user encoders as compared to `PerAugy`.

**SDAInter.** We include SDAInter Jiao et al. (2024) as a *cross-trajectory* augmentation baseline that generates pseudo user sequences by identifying interchangeable subsequences in $\mathcal{T}_{\text{train}}^P$ between different user trajectories based on shared anchor items. If the subsequences between two users meet a minimum IoU-based interchangeability confidence $C \geq T_c$, they are swapped to create new synthetic trajectories $\tau_{\text{SDA}}^{\hat{u}_{1:m}}$. The resulting SDAInter-augmented pool $\mathcal{T}_{\text{train}}^{\text{SDA}}$ is evaluated for its effect on user-encoder performance and compared against `PerAugy`, along with evaluation of $\mathcal{T}^{\text{syn-SDA}}$ (synthetic UIG version of SDAInter augmented trajectories by incorporating intermediate s-nodes) for `DegreeD`.

**LLM-as-Augmentor.** We also compare our method against three popular LLMs – Llama-2-13B Touvron et al. (2023), Mistral-v2-Instruct Jiang et al. (2023), and DeepSeek-7b-chat DeepSeek-AI et al. (2025) in two prompt-based settings - *1) Chain-of-Thoughts* Wei et al. (2023) which involves guiding an LLM through a series of logical reasoning steps to solve a task Wei et al. (2023). Using LLaMA-2-13B as the base model, we design a CoT prompt with detailed step-by-step instructions to logically generate a personalized summary based on a users interaction history (see Figure 9), and *2) Prompt-Chaining* Sahoo et al. (2024) which is a technique that involves using a series of prompts to deconstruct a task into sequential steps. Our prompt-chaining setup consists of two sequential tasks. In the first task (step-1), the LLM generates user interactions

in the form of (document, action) pairs, simulating user behavior (e.g., clicks, skips). In step-2, using few-shot prompting, the LLM generates a personalized summary, conditioned on both the input document and the user interactions generated in step-1. Each few-shot prompt contains four in-context examples (see Figure 10). These generated trajectories are then subsequently used to train user-encoder models (as described in Section 5.1.2).

## C.2 Baseline User-Encoders

In this section We discuss SOTA news headline recommendation models which were used as baseline user-encoders to understand the effect of `PerAugy` generated training data.

**NAML.** Neural News Recommendation with Attentive Multi-View Learning (NAML) Wu et al. (2019a) is a neural news recommendation approach that learns informative representations of users and news by exploiting different kinds of news information. The core of this approach is a user-encoder and a news encoder where the news encoder learns unified news representations from titles, bodies, and topic categories by regarding them as different perspectives of the news, through an attentive multi-view learning model, and the user-encoder learns the representations of users based on their browsing history and applies attention mechanism to select informative news for user representation learning.

**NRMS.** Neural News Recommendation with Multi-Head Self-Attention (NRMS) Wu et al. (2019b) is a neural news recommendation where the core of the encoder is a news encoder that uses multi-head self-attention to learn news representations from news titles by modeling the interactions between words and a user-encoder which learns user representations from their browsed news and use multi-head self-attention to capture relatedness between the news. Additive attention to learn more informative news and user representations is used, by selecting important words and news.

**EBNR.** Embedding-based News Recommendation (EBNR) Okura et al. (2017) is an RNN-styled news recommendation approach that incorporates implicit negative user feedback by distinguishing positive and negative news clicks based on the reading dwell time of the news by the user, and learning the user representations from positive and negative samples via a combination of Transformer and additive attention network. It computes a final click score as a combination of positive click scores and negative click scores.

**TrRMIo.** Transfomer-based Recommendation Model Song et al. (2023) utilizes a personalized news recommendation model to represent users' preferences derived from clicked records. The pre-trained transformer models are used for both recommendation and headline generation tasks, where a news encoder and a user-encoder is adopted for content-based recommendation. The textual information from the news encoder is aggregated via Attention Pooling, which is then further integrated by user representation. The user interest is defined on the basis of Click Through Rate (CTR) by examining the frequency of news articles in users' click histories and positive samples. The assumption is that user history consists of both popular news and interested news, where popular news has a higher CTR ranking across the users, while interested news has lower CTR ranks, depicting the personalized choice of that user to generate the user representation.This laid the foundation for Transformer-based Recommendation Model Interest-Only(TrRMIo), where news histories with lower CTR for a particular user is treated as his 'Interest-Only' features.

## C.3 Baseline Personalized Summarizers

To determine whether `PerAugy`-generated training data enhances the regularization of specialized Pretrained Language Models, improving `PerSEval` performance in personalized summarization tasks, we benchmark the PENS personalized summarization framework Ao et al. (2021) and the recent GTP personalized summarization framework Song et al. (2023). We describe each of the frameworks as follows:

**PENS.** The PENS framework employs a transformer-based encoder to process the news body and a pointer networkbased decoder to generate headlines. The pointer mechanism is used to dynamically choose between generating words from a vocabulary and copying words directly from the news text, which helps in handling

| UIG | # u-nodes (trajectories) | # d-nodes per trajectory | # s-nodes per trajectory | Average trajectory length | Maximum trajectory length |
|---|---|---|---|---|---|
| PENS | 360K+ | 83.78 | 1.94 | 85.72 | 2433 |
| PENS-SH† | 197K | 310.56 | 1.01 | 321.58 | 5106 |
| S3† | 360K+ | 79.4 | 2.2 | 83.2 | 2167 |
| SDAInter† | 360K+ | 87.6 | 2.8 | 123.6 | 1742 |
| LLaMA-2-13B† | 2176 | 29 | 2 | 13.6 | 17 |
| Mistral-7B† | 5711 | 31.25 | 2.87 | 34.13 | 58 |
| DeepSeek-R1† | 813 | 37 | 4.23 | 32.7 | 35 |
| OpenAI (Reddit) (OAI) | 126K | 25.19 | 4.82 | 30.02 | 54 |
| PerAugy†* | 360K+ | 123.7 | 5.1 | 129.8 | 200 |
| PerAugy-OAI* | 360K+ | 36.92 | 11.44 | 48.37 | 50 |

Table 10: **User-interaction graph statistics:** Two seed datasets chosen– PENS train dataset (Table 8) and OpenAI (Reddit) train dataset (Table 9); Baseline augmentation methods– (i) PENS-synthetic-base (ours; as *), (ii) PENS-SH, (iii) LLaMA-2-13B, (iv) Mistral-7B, (v) DeepSeek-R1, and (vi) PerAugy-PENS/OAI (ours); †augmentation followed by UIG abstraction on the PENS dataset.

out-of-vocabulary words and maintaining factual consistency. To personalize the headline generator, three distinct injection strategies for incorporating user embeddings (*learned from user behavior data using state-of-the-art news recommendation models as user-encoders*) is proposed: (i) **Decoder Initialization** where the user embedding is used to initialize the decoders hidden state, so the generation process is conditioned from the very start on the user's interests, (ii) **Attention Perturbation** where the user embedding is injected into the attention mechanism. This modulates the attention distribution over the news body words, effectively guiding the model to focus on parts of the text that align with the users preferences, and (iii) **Generation-Copy Switch Adjustment** where the user embedding is also used to perturb the probability (or switch) that determines whether the decoder generates a word from the vocabulary or copies a word from the news body. This helps ensure that the generated headline reflects personalized nuances rather than just summarizing the article content.

**GTP.** General Then Personal (GTP) is a framework that tackles personalized headline generation by decoupling the task into two sequential stages. In stage 1, a Transformerbased encoderdecoder model (e.g., BART) is pre-trained on large-scale news articleheadline pairs to learn robust, content-focused headline generation without any personalization. In stage 2, a separate headline customizer takes the general headline and refines it by incorporating user-specific preferences. These preferences are encoded (control code) by the user-encoder TrRMIo. To bridge the gap between the general generation and personalized refinement, the authors introduce two key mechanisms: (i) **Information Self-Boosting (ISB)** that enhances the customization by reintroducing relevant content details from the news article to ensure that personalization does not lead to information loss, and (ii) **Masked User Modeling (MUM)** that helps the model learn to recognize and utilize the user control code by randomly masking parts of the user embedding during training and then reconstructing them, thereby reducing over-reliance on the general model parameters.

## D   Encoder/Decoder Accuracy Metrics

In this section, we provide a detailed formulation of the user-encoder accuracy metrics in the context of the next d-node prediction task.

**AUC.** The Area Under the Curve (AUC) measures the probability that a randomly chosen positive d-node is ranked higher than a randomly chosen negative d-node in the test dataset $\mathcal{T}_{\text{test}}^{\text{P}}$. The formula is given as:

$$AUC = \frac{1}{|P| \cdot |N|} \sum_{p \in P} \sum_{n \in N} \mathbb{1}(s_p > s_n) \tag{14}$$

where:

- $P$ is the set of positive interactions (set of clicked d-nodes of all users in test data).

- $N$ is the set of negative items (set of skipped d-nodes of all users in test data).

- $s_p$ is the predicted score for a positive d-node.

- $s_n$ is the predicted score for a negative d-node.

- $\mathbb{1}(s_p > s_n)$ is an indicator function that equals 1 if $s_p > s_n$, otherwise 0.

- $|P|$ and $|N|$ are the number of positive and negative d-nodes, respectively.

**MRR.** Mean Reciprocal Rank (MRR) evaluates how early the ground-truth target d-node appears in the ranking. It is defined as:

$$MRR = \frac{1}{|U|} \sum_{u_j \in U} \frac{1}{\text{rank}_{u_j}} \tag{15}$$

where:

- $U$ is the set of users.

- $\text{rank}_u$ is the position of the first relevant d-node for user $u_j$ in the ranked recommendation list.

- $|U|$ is the total number of users.

A higher MRR indicates that the target d-node is ranked closer to the top of the prediction list, improving user experience.

**nDCG@k.** Normalized Discounted Cumulative Gain at rank $k$ (nDCG@k) evaluates the ranking quality by considering both the prediction score and the position of ground-truth target d-node. It is defined as:

$$nDCG@k = \frac{DCG@k}{IDCG@k}; \text{where: } DCG@k = \sum_{i=1}^{k} \frac{s_i}{\log_2(i+1)}; IDCG@k = \sum_{i=1}^{k} \frac{s_i^*}{\log_2(i+1)} \tag{16}$$

Here:

- $s_i$ is the prediction score of the target d-node at rank $i$ in the recommended list.

- $s_i^*$ is the actual score of the target d-node at rank $i$ in the ideal ranking (sorted by prediction score).

- $DCG@k$ is the Discounted Cumulative Gain up to rank $k$.

- $IDCG@k$ is the Ideal Discounted Cumulative Gain, representing the best possible ranking.

A higher nDCG@k indicates that the target d-node is ranked higher, improving prediction effectiveness.

## E  Algorithms

In this section, we discuss the details of the algorithms used in our paper.

**UIG Construction**  We construct the User Interaction Graph (UIG) by parsing interaction logs from two types of seed datasets: (i) PENS-styled, where explicit click and skip behaviors are available, and (ii) OpenAI(Reddit)-styled, where user preferences are inferred from model confidence and summary quality ratings. The algorithm maps user interactions to document (d-node) and summary (s-node) nodes, while assigning appropriate behavioral edges such as `click`, `skip`, `gensum`, and `sumgen` to encode both explicit and inferred preferences.

---

**Algorithm 1 UIG Construction**

---

**Require:** `train data` and `test data`, `dataset_type`
  Initialize $\mathcal{T} \leftarrow \emptyset$
  **for** each user $u$ in train_data **do**
    Initialize $\tau^u \leftarrow \emptyset$
    **for** each interaction in $u$'s data **do**
      **if** dataset_type = `PENS` **then**
        Map interaction to d-node with `click`/`skip` edge
      **else if** dataset_type = `OPENAI` **then**
        **if** any model-generated summary for d-node has confidence score $\geq$ threshold **then**
          Label as `click`
          Select best-rated summary by $u$ as surrogate s-node
          Map to d-node and s-node with `gensum` and `sumgen` edges
        **else**
          Label as `skip`
        **end if**
      **else**
        Map rating to d-node with `click`/`skip` edge
        **if** rating is max **then**
          Map to d-node and s-node with `gensum` and `summGen` edges
        **end if**
      **end if**
      Append d-node to $\tau^u$
    **end for**
    Add $\tau^u$ to $\mathcal{T}$
  **end for**
  **if** dataset_type = `PENS` **then**
    **for** each $\tau^u$ in $\mathcal{T}$ **do**
      **if** d $\in$ `train data` AND d $\in$ `test data` **then**
        Insert (d-s)-nodes from test_data as `genSumm`/`summGen` edges
      **end if**
    **end for**
    **return** $\mathcal{T}^{\text{PENS-D}} \leftarrow \mathcal{T}$
  **else**
    **return** $\mathcal{T}$
  **end if** =0

---

`DegreeD` **Computation of UIGs** In section 7.1, we described a special case where for every d-node in an UIG we have a corresponding s-node in every trajectory $\tau$, which evidently is unrealistic. In reality, many of the d-nodes will not have a corresponding s-node and hence, the calculation of `DePS` cannot be done at every unit time-interval $\Delta_{(t_i, t_{i+1})}$. This requires modification in the computing procedure so as to account for the missing s-nodes. To address this, the first "*surrogate*" s-node $s_j^{(t_1)}$ for the initial d-node $d^{(t_1)}$ at time-step $t_1$ is assumed to be the same as the document's title, as there is no prior preference history of a user at time-step $t_1$ and hence, subjectivity as a function of preference history does not arise yet. Let the first s-node $s_j^{(t_k)}$ occur at time-step $t_k$ (i.e., the first valid interval is $\Delta_{(t_1, t_k)}$). Therefore, `DePS`$^{\Delta_{(t_1, t_k)}}$ for $j$-th user is calculated as:

$$\text{DePS}_j^{\Delta_{(t_1,t_k)}} = \frac{min(\delta[X^{\Delta_{(t_1,t_k)}}]_d, \delta[X^{\Delta_{(t_1,t_k)}}]_{s_j}) + \epsilon}{max(\delta[X^{\Delta_{(t_1,t_k)}}]_d, \delta[X^{\Delta_{(t_1,t_k)}}]_{s_j}) + \epsilon};$$

$$\text{where: } \delta[X^{\Delta_{(t_1,t_k)}}]_d = \frac{1}{k-1} \cdot \sum_{i=1}^{k-1} \sigma(d^{(t_1)}, d^{(t_{1+i})}); \delta[X^{\Delta_{(t_1,t_k)}}]_{s_j} = \sigma(s_j^{(t_1)}, s_j^{(t_k)})$$

(17)

---

**Algorithm 2 Computing DegreeD**

---

1: **Input:** Users $U$, Actions $A$, Summaries $S$, Documents $D$, window size $w$
2: **for** each trajectory $(U, A, D, S)$ **do**
3:     $\mathcal{D}_{\text{total}} = 0$, $D_{\text{dist}} = []$, $D_{\text{MA}} = 0$
4:     **for** $t_1 = 1$ to $|A| - 1$ **do**
5:         Retrieve $D_{t_1}, U_{t_1}$
6:         **if** $A_{t_1}$ is click/skip **then**
7:             $D_{\text{dist}} \leftarrow \sigma(D_{t_1}, D_{\text{prev}})$, update $D_{\text{MA}}$
8:         **else if** $A_{t_1} = \text{gen\_summ}$ **then**
9:             $D_{t_2} = D_{\text{MA}}$, $\delta = \frac{\min(D_{t_1}, D_{t_2}) + \epsilon}{\max(D_{t_1}, D_{t_2}) + \epsilon}$
10:             $P = \frac{\sigma(D_{t_1}, U_{t_1})}{\sigma(D_{t_2}, U_{t_2}) + \epsilon}$
11:             $\mathcal{D}_{\text{total}} + = \delta \cdot P \cdot \sigma(U_{t_2}, U_{t_1})$
12:         **end if**
13:     **end for**
14:     $\mathcal{D} + = \mathcal{D}_{\text{total}}/(|A| - 1)$
15: **end for**
16: $\mathcal{D} \leftarrow \mathcal{D}/|U| = 0$

---

DegreeD is then computed over all valid intervals as per equation 7. In this paper, we represent d-nodes and s-nodes with their embeddings generated from a lightweight S-BERT model Reimers & Gurevych (2019) and use Manhattan Distance as the distance metric $\sigma$. It is important to note that DegreeD is fundamentally defined as a ratio of relative variations in distances between d- and s-nodes within the same interval (Equation 17). This formulation ensures that the metric only depends on how well s-nodes track the semantic drift of d-nodes, rather than on the absolute scale of any embedding space or distance function. In other words, any embedding model (e.g., SBERT, BERT, or domain-specific encoders) and any valid distance metric $\sigma$ (e.g., Manhattan, cosine, Euclidean) merely provide a representation space in which distances are computed, but the ratio-based structure of DegreeD cancels out biases due to embedding geometry or metric scaling. Hence, DegreeD remains a model- and metric-agnostic measure of diversity, relying only on the relative alignment of document w.r.t. summary dynamics. In section 6, we empirically provide strong evidence that higher UIG DegreeD has strong correlation with user-encoder model accuracy when trained on such UIGs (DegreeD computation is in Algorithm 2). We compute the DegreeD of the PENS synthetic base pool $\mathcal{T}_{\text{base}}^{\text{syn-P}}$ and find a very low DegreeD score of 0.009. The OpenAI (Reddit) synthetic base pool $\mathcal{T}_{\text{base}}^{\text{syn-OAI}}$ also shows low DegreeD of 0.0079.

**PerAugy: Double Shuffling – Algorithm Details**    The PerAugy framework enhances user interaction generalization by introducing two complementary augmentation strategies: Double Shuffling (DS) and Stochastic Markovian Perturbation (SMP). In DS (Algorithm 3), target user trajectories are systematically altered by substituting randomly selected segments from other users interaction histories. These substitutions occur at randomized offsets and are spaced by controlled gap lengths to maintain temporal realism and simulate cross-user behavioral blending. This process generates diversified yet structurally plausible trajectories, expanding the training distribution without deviating from feasible user behavior patterns.

**PerAugy: Stochastic Markovian Perturbation – Algorithm Details**    Following DS, the SMP stage (Algorithm 4) further refines the augmented trajectories by focusing on semantic consistency at the summary level. Specifically, newly introduced summaries are evaluated within a local Markovian window of recent document interactions, and replaced with top-ranked candidates based on a relevance score computed via RMSD (Root Mean Square Distance) similarity. These candidates are weighted by an exponential temporal decay factor to prioritize more recent contextual nodes, ensuring that substituted summaries align with short-term user interest profiles. Together, DS and SMP act in synergy to produce coherent, high-quality augmented data that preserves personalization cues while introducing controlled variability – a crucial property for training robust and generalizable user models in recommendation and summarization tasks.

---

**Algorithm 3 Double Shuffling (DS)**

---

**Require:** A UIG trajectory pool $\mathcal{T}_{\text{base}}^{syn}$, sample size $m$ and gap-length $g_l$
**Ensure:** Modified trajectory set $\mathcal{T}_{\text{DS}}^{m}$
1: $\mathcal{T}_{\text{sample}}^{m} \leftarrow \text{SampleWithoutReplacement}(\mathcal{T}_{\text{base}}^{syn}, m)$
2: $\mathcal{T}_{\text{DS}}^{m} \leftarrow \emptyset$
3: **for** each target trajectory $\tau_{\text{target}}^{u_j} \in \mathcal{T}_{\text{sample}}^{m}$ **do**
4:     $O \leftarrow \textbf{RandomOffset}()$
5:     $I_{\text{subs}} \leftarrow O$
6:     **for** each source trajectory $\tau_{\text{source}}^{u_i} \in \mathcal{T}_{\text{sample}}^{m}$, where $i \neq j$ **do**
7:         $\tau_{seg}^{u_i} \leftarrow \textbf{RandomSegment}(\tau_{\text{source}}^{u_i})$ {Select a trajectory segment of random length at random time-steps.}
8:         $\tau_{\text{target}}^{u_j} \leftarrow \textbf{Substitute}(\tau_{\text{target}}^{u_j}, \tau_{seg}^{u_i}, I_{\text{subs}})$
9:         $I_{\text{subs}} \leftarrow O + length(\tau_{seg}^{u_i}) + g_l$ {Determine substitution indices in $\tau_{\text{target}}^{u_j}$, ensuring that two source segments are separated by gap-length $g_l$.}
10:     **end for**
11:     $\tau_{\text{DS}}^{u_j} \leftarrow \tau_{\text{target}}^{u_j}$
12:     $\mathcal{T}_{\text{DS}}^{m} \leftarrow \mathcal{T}_{\text{DS}}^{m} \cup \{\tau_{\text{DS}}^{u_j}\}$
13: **end for**
14: **return** $\mathcal{T}_{\text{DS}}^{m}$ =0

---

**Algorithm 4 Stochastic Markovian Perturbation (SMP)**

---

**Require:** DS trajectories $\mathcal{T}_{\text{DS}}^{m}$, window $k$, decay $\lambda$, top-$p$
**Ensure:** perturbed set $\mathcal{T}_{\text{SMP}}^{m}$
1: $\mathcal{T}_{\text{SMP}} \leftarrow \emptyset$
2: **for** each $\tau \in \mathcal{T}_{\text{DS}}^{m}$ **do**
3:     **for** each step $t$ in $\tau$ **do**
4:         **if** $s^{(t)}$ newly substituted **then**
5:             Retrieve $d^{(t-1)}$, extract $\{st\}$, define window $\{c\}$
6:             **for** each $st \in d^{(t-1)}$ **do**
7:                 $I(st) \leftarrow \sum_c \text{RMSD}(st, c) e^{-\lambda\, pos(c)}$
8:             **end for**
9:             Rank $\{st\}$, pick top-$p$ $\hat{s}^{(t)}$, replace $s^{(t)}$
10:         **end if**
11:     **end for**
12:     $\mathcal{T}_{\text{SMP}} \leftarrow \mathcal{T}_{\text{SMP}} \cup \{\tau\}$
13: **end for**
14: **return** $\mathcal{T}_{\text{SMP}}$ =0

---

# F   Stability of DegreeD Correlation Under Divergence Substitution

## F.1   Setup

Let us assume a dataset $D$ with users $U$. For each user $j$ with trajectory length $L_j$, define from (4)

$$\delta_s^{(j)}(i) = \sigma(s_j(t_i), s_j(t_{i+1})), \quad \delta_d^{(j)}(i) = \sigma(d(t_i), d(t_{i+1})); \quad \Delta_s^{(j)} = \frac{1}{L_j - 1} \sum_{i=1}^{L_j - 1} \delta_s^{(j)}(i).$$

Let the timestep-level ratio be defined as: $r^{(j)}(i) = \delta_s^{(j)}(i)/\delta_d^{(j)}(i)$, with

$$\text{DePS}^{(j)}(i) = \phi(r^{(j)}(i)), \quad \phi(r) = \frac{\min(r, 1) + \epsilon}{\max(r, 1) + \epsilon}.$$

At the same time, the penalty term is

$$p^{(j)}(i) = \frac{\sigma(d(t_i), s_j(t_i)) + \epsilon}{\sigma(d(t_{i+1}), s_j(t_{i+1})) + \epsilon},$$

and the expected penalized DePS

$$\mathbb{E}_j[\text{DePS}^P] = \frac{1}{L_j - 1} \sum_{i=1}^{L_j - 1} \text{DePS}^{(j)}(i) \cdot^{(j)}(i).$$

Finally,

$$\texttt{DegreeD}_\sigma(D) = \alpha \frac{1}{|U|} \sum_{j \in U} \Delta_s^{(j)} \cdot \mathbb{E}_j[\text{DePS}^P].$$

For datasets $\{D_k\}_{k=1}^m$, let $F_k = \text{DegreeD}_\sigma(D_k)$, $G_k = \text{DegreeD}_{\sigma'}(D_k)$, and $A_k$ for accuracy.

**Metric Substitution.** Let $\sigma'$ be the substitute metric of $\sigma$.

Assuming $\sigma'$ is Bi-Lipschitz equivalent to $\sigma$: $0 < \lambda \le \Lambda < \infty$ s.t. $\lambda\sigma(x,y) \le \sigma'(x,y) \le \Lambda\sigma(x,y)$,    (18)

for all pairs used in DegreeD, with $\kappa = \Lambda/\lambda$. For pure scalings $\sigma' = c\sigma$, $\lambda = \Lambda = c$.

## F.2 Post-Substitution Time-stepwise Bounds

**Lemma 1** (Local distortions). *For all $i, j$,*

$$\lambda\Delta_s^{(j)} \le \Delta_s'^{(j)} \le \Lambda\Delta_s^{(j)}, \quad \frac{1}{\kappa}\phi(r) \le \phi(r') \le \kappa\phi(r), \quad \frac{1}{\kappa}p^{(j)}(i) \le p'^{(j)}(i) \le \kappa p^{(j)}(i).$$

**Proposition 1** (User-level Bounding Inequalities). *Let $H_j = \Delta_s^{(j)} \cdot \mathbb{E}_j[\text{DePS}^P]$. Then, using Lemma 1,*

$$\frac{\lambda^3}{\Lambda^2} H_j \;\le\; H_j' \;\le\; \frac{\Lambda^3}{\lambda^2} H_j.$$

**Corollary 1** (Dataset-level Bounding Inequalities). *Aggregating over users, we obtain*

$$K_- F_k \;\le\; G_k \;\le\; K_+ F_k, \qquad K_- = \frac{\lambda^3}{\Lambda^2}, \quad K_+ = \frac{\Lambda^3}{\lambda^2}.$$

**Corollary 2** (Pure scaling). *If $\sigma' = c\sigma$, then $G_k = cF_k$ and all correlations with $A$ are unchanged.*

## F.3 Post-Substitution Rank Stability

Let us define an ambiguity band $\mathcal{B}$ that captures the zone of uncertainty where relative rankings between two dataset DegreeD-scores $F_k$ and $G_k$ may flip under divergence substitution. From Corollary 1 for each dataset $k$ we can conclude that for any dataset pair $k, \ell$,

$$\frac{G_k}{G_\ell} \;\in\; \Big[\frac{K_-}{K_+}\frac{F_k}{F_\ell}, \; \frac{K_+}{K_-}\frac{F_k}{F_\ell}\Big].$$

If $\frac{F_k}{F_\ell}$ lies outside $\left[\frac{K_-}{K_+}, \frac{K_+}{K_-}\right]$, then the order of $F_k$ and $F_\ell$ is preserved for all possible distortions, guaranteeing rank stability. Conversely, if the ratio falls inside this interval, an inversion is possible: one dataset-DegreeD could be stretched to its upper bound while the other shrinks to its lower bound. Substituting the explicit forms of $K_\pm$ gives:

$$\frac{K_-}{K_+} = \Big(\frac{\lambda}{\Lambda}\Big)^5, \qquad \frac{K_+}{K_-} = \Big(\frac{\Lambda}{\lambda}\Big)^5,$$

Let the uncertainty interval be defined as *Ambiguity Band* $\mathcal{B} = [(\lambda/\Lambda)^5, \; (\Lambda/\lambda)^5]$.

**Proposition 2** (Substitution Stability of DegreeD)**.** *If $F_k/F_\ell \notin \mathcal{B}$ for all $k \neq \ell$, then $F$ and $G$ have identical rankings; hence $\rho_s(F, G) = \tau(F, G) = 1$.*

While Proposition 2 ensures exact rank preservation whenever all dataset diversity (w.r.t. `DegreeD`) ratios $F_k/F_\ell$ lie outside the ambiguity band $\mathcal{B}$, in practice some pairs may fall inside $\mathcal{B}$, permitting local inversions. We record here two principled ways to quantify partial rank stability.

**Exact identity.** For any two rankings of $m$ datasets with rank differences $\{d_i\}_{i=1}^{m}$, Spearman's correlation satisfies

$$\rho_s(F, G) \;=\; 1 - \frac{6\sum_{i=1}^{m} d_i^2}{m(m^2 - 1)}. \tag{19}$$

Thus the precise degradation of $\rho_s$ depends on the displacement profile $\{d_i^2\}$, not only on the number of inversions.

**Adjacency-limited inversions.** If all inversions correspond to adjacent swaps, then each inversion contributes at most 2 to $\sum d_i^2$, implying

$$\rho_s(F, G) \;\geq\; 1 - \frac{12K}{m(m^2 - 1)}, \tag{20}$$

where $K$ is the number of inversions. This yields a computable lower bound under localized perturbations.

**General inversions.** Without further assumptions, no nontrivial bound in terms of $K$ alone is possible: moving a single element down by $t$ places creates $K = t$ inversions but contributes $d^2 = t^2 + t$, which may degrade $\rho_s$ substantially more. Hence, robust guarantees require either (a) bounding the displacement profile directly from data, or (b) assuming structural restrictions (e.g. inversions are local).

### F.4 Pearson Correlations

**Proposition 3** (Correlation transfer)**.** *Let $r_{FA} = \mathrm{corr}(F, A)$, $r_{FG} = \mathrm{corr}(F, G)$, $r_{GA} = \mathrm{corr}(G, A)$. Then*

$$r_{GA} \;\geq\; r_{FG} r_{FA} - \sqrt{1 - r_{FG}^2}\,\sqrt{1 - r_{FA}^2}.$$

**Lemma 2** (Lower bound on $r_{FG}$)**.** *From Corollary 1,*

$$r_{FG} \;\geq\; \sqrt{K_-/K_+} \;=\; (\lambda/\Lambda)^{2.5}.$$

### F.5 Summary Results

**Theorem 1** (Correlation stability of DegreeD)**.** *Under the bi-Lipschitz equivalence* (18)*, the following hold:*

1. ***Rank stability (from Proposition 2):*** *If all $F_k/F_\ell$ avoid $\mathcal{B}$, then $\rho_s(F, G) = \tau(F, G) = 1$.*

2. ***Pearson transfer (from Proposition 3 & Lemma 2):*** *For any external accuracy variable $A$,*

$$\mathrm{corr}(G, A) \;\geq\; \kappa_0\,\mathrm{corr}(F, A) - \sqrt{1 - \kappa_0^2}\,\sqrt{1 - \mathrm{corr}(F, A)^2}, \;\; with \;\; \kappa_0 = (\lambda/\Lambda)^{2.5}.$$

3. ***Scaling invariance (from Corollary 2):*** *If $\lambda = \Lambda$, then $G_k = cF_k$ and all correlations are preserved exactly.*

### F.6 Results for $\sigma$ : RMSD

- **Euclidean vs. RMSD:** $\|x - y\|_2 = \sqrt{d}\,\mathrm{RMSD}(x,y)$. This case guarantees pure scaling and hence, correlations remain exactly unchanged (Corollary 2).

- **Cosine distance:** $\cos\_\mathrm{dist}(x,y) = \frac{d}{2}\mathrm{RMSD}^2(x,y)$. Here $\sigma'$ is Lipschitz equivalent to $\sigma$ provided RMSD is bounded away from zero; Theorem 1 applies.

- **Angular distance:** Monotone in cosine similarity, hence Lipschitz related to RMSD in bounded domains. Rank stability (Proposition 2) guarantees identical orderings outside $\mathcal{B}$.

- **Mahalanobis distance:** $\|x - y\|_M$ is bi-Lipschitz to Euclidean with constants $\sqrt{\lambda_{\min}(M)}$ and $\sqrt{\lambda_{\max}(M)}$, so all conclusions of Theorem 1 apply with $\kappa = \sqrt{\lambda_{\max}(M)/\lambda_{\min}(M)}$.

By chaining Lemma 1, Proposition 1, Corollary 1, we established the main squeeze bound for `DegreeD` under divergence substitution. Proposition 2 shows that if all dataset-`DegreeD` ratios $F_k/F_\ell$ avoid the ambiguity band $\mathcal{B}$, then rankings are preserved exactly. In Section F.3 we extend this to partial perturbations where we show the condition for exact identity w.r.t rank displacements in Spearman correlation. This clarifies how DegreeD rankings degrade in controlled ways when some pairs fall inside $\mathcal{B}$.

## G  Implementation Details

**Computing Resources.**  The creation of User Interaction Graphs (UIGs) and computation of `DegreeD` are performed on a standard 4-core CPU with 16GB of RAM. For Stochastic Markovian Perturbation (SMP), we use the SBERT all-MiniLM-L6-v2 model (Reimers & Gurevych, 2019) to generate embeddings, and the SMP process takes approximately 16 hours to complete on an NVIDIA A-100 GPU. LLM-based experiments are conducted using 3 NVIDIA A-100 GPUs.

**PerAugy Settings.**  We generate the embeddings of the d-nodes and s-nodes using the SBERT Reimers & Gurevych (2019) 'all-MiniLM-L6-v2 model', which has 22.7M parameters and an embedding size of 384. Manhattan distance is used to compute embedding divergence during the `DegreeD` calculation, chosen for its linear scalability and efficiency. For SMP, embeddings of sentences and d-nodes within the context window are generated using the same SBERT model, followed by the use of RMSD to compute similarity scores for perturbation.

**Model Settings.**  Three user encoders (NAML (Wu et al., 2019a), EBNR (Okura et al., 2017), NRMS (Wu et al., 2019b)) are trained on `PerAugy` datasets for 2 epochs, with a learning rate of 0.0001 and batch size 128 using the Adam optimizer. The models are finetuned on the $\mathcal{T}^{\mathcal{E}}_{\mathrm{DS/DS+SMP}}$ datasets in the TrRMIo model for one epoch after training from scratch. During training, intermediate s-nodes are modeled as d-nodes to integrate them into the user encoders.

**LLM Settings for Prompts.**  Prompting experiments are conducted using two setups: (1) Chain-of-Thoughts with LLaMa2-13B and (2) Prompt-Chaining with Mistral-Instruct-v2 and DeepSeek-7B-Chat. For LLaMa2-13B, we perform inference using sampling with temperature set to 0.75, top-$p$ to 0.9, and top-$k$ to 50. For Mistral-Instruct-v2 and DeepSeek-7B-Chat, we use a deterministic sampling strategy (temperature = 0.0, top-$p$ = 1.0) for controlled generations. Max_tokens are set to 1024 for both setups.

## H  Correlation Computation

To quantify the relationship between encoder accuracies and diversity metrics, as well as the inter-dependence of diversity metrics themselves, we employ three standard correlation measures: Pearsons correlation coefficient, Spearmans rank correlation coefficient, and Kendalls $\tau$ coefficient. Their formulations are provided below.

**Formulations.** Given paired observations $\{(x_i, y_i)\}_{i=1}^n$, we compute:

1. **Pearson Correlation:**
$$r = \frac{\sum_{i=1}^n (x_i - \bar{x})(y_i - \bar{y})}{\sqrt{\sum_{i=1}^n (x_i - \bar{x})^2}\sqrt{\sum_{i=1}^n (y_i - \bar{y})^2}}, \tag{21}$$
   where $\bar{x}$ and $\bar{y}$ are the sample means.

2. **Spearman Rank Correlation:**
$$\rho = 1 - \frac{6\sum_{i=1}^n d_i^2}{n(n^2 - 1)}, \tag{22}$$
   where $d_i$ is the difference between the ranks of $x_i$ and $y_i$.

3. **Kendalls $\tau$:**
$$\tau = \frac{C - D}{\frac{1}{2}n(n-1)}, \tag{23}$$
   where $C$ and $D$ denote the number of concordant and discordant pairs, respectively.

**Accuracy vs. Diversity Metrics.** Let $\mathcal{A} = \{a_j\}_{j=1}^M$ denote the set of averaged encoder accuracies across all encoders for dataset $j$, and let $\mathcal{D}^k = \{d_j^k\}_{j=1}^M$ denote the set of diversity scores corresponding to the $k$-th diversity metric, where $k \in \{\text{PENS}, \text{PENS-SH}, \text{S3}, \text{SDAInter}, \texttt{PerAugy-DS}, \texttt{PerAugy-DS+SMP}\}$. For each $k$, we compute the correlation with accuracy as

$$\text{Corr}^{(f)}(\mathcal{A}, \mathcal{D}^k) = f\left(\{(a_j, d_j^k)\}_{j=1}^M\right), \tag{24}$$

where $f \in \{r, \rho, \tau\}$ corresponds to Pearson, Spearman, or Kendalls correlation, respectively.

Note that LLM-generated trajectories are excluded from $\mathcal{D}^k$ due to their inability to produce complete trajectory sets with $113K$ news items, resulting in artificially lower accuracies despite exhibiting topical diversity and frequent shifts.

**Inter-Correlation of Diversity Metrics.** Let $\mathcal{M} = \{\text{TP}, \text{RTC}, \texttt{DegreeD}\}$ denote the set of inter-diversity metrics. For each dataset $j$, we obtain the metric values $\mathcal{D}_j = \{d_j^m\}_{m \in \mathcal{M}}$. For any pair $(m_1, m_2) \in \mathcal{M} \times \mathcal{M}$, we compute

$$\text{Corr}^{(f)}(\mathcal{D}^{m_1}, \mathcal{D}^{m_2}) = f\left(\{(d_j^{m_1}, d_j^{m_2})\}_{j=1}^M\right), \tag{25}$$

where $f \in \{r, \rho, \tau\}$.

This formulation allows us to construct a correlation matrix across $\mathcal{M}$, thereby quantifying the degree of alignment or divergence among different internal diversity measures (including those derived from LLMs).

# I Detailed Results

## I.1 Ablation Studies (RQ-1)

We ablate on the mixed training data $\mathcal{T}_{\text{DS}}^{\mathcal{E}-\text{P}}$ to analyze the effect of DS hyper-parameters– gap-length $g_l$ (section 5.1.1) and train history-segment length $\tau_{h_{\text{train}}}$: $\{l/2, 5l/8, 3l/4, 7l/8, l-3\}$ ($l$: trajectory length). For SMP hyper-parameters ($k$: $\{10, 15, 20\}$, $\lambda$: $\{0.3, 0.8, 1\}$, $p_{\text{SMP}}$: $\{0.5, 0.8, 1\}$), we ablate on $\mathcal{T}_{\text{DS+SMP}}^{\mathcal{E}-\text{P}}$. Results are in Figure 6.

**Effect of $\tau_{h_{\text{train}}}$ & $g_l$** We fix $g_l$ to 25 and observe that $\tau_{h_{\text{train}}}$ has a major impact across all user-encoders with the longest $\tau_{h_{\text{train}}}$ ($l - 3$) having the highest mean boost ($0.064 \uparrow$ w.r.t AUC, $0.035 \uparrow$ w.r.t MRR, $0.011 \uparrow$ w.r.t nDCG@5/10) against the least scores, thereby confirming that longer preference history in train is better. $g_l$ ($l$ fixed at 150) also matters particularly w.r.t AUC with best at 40. This shows that synthetic profiles having longer original user segments are better.

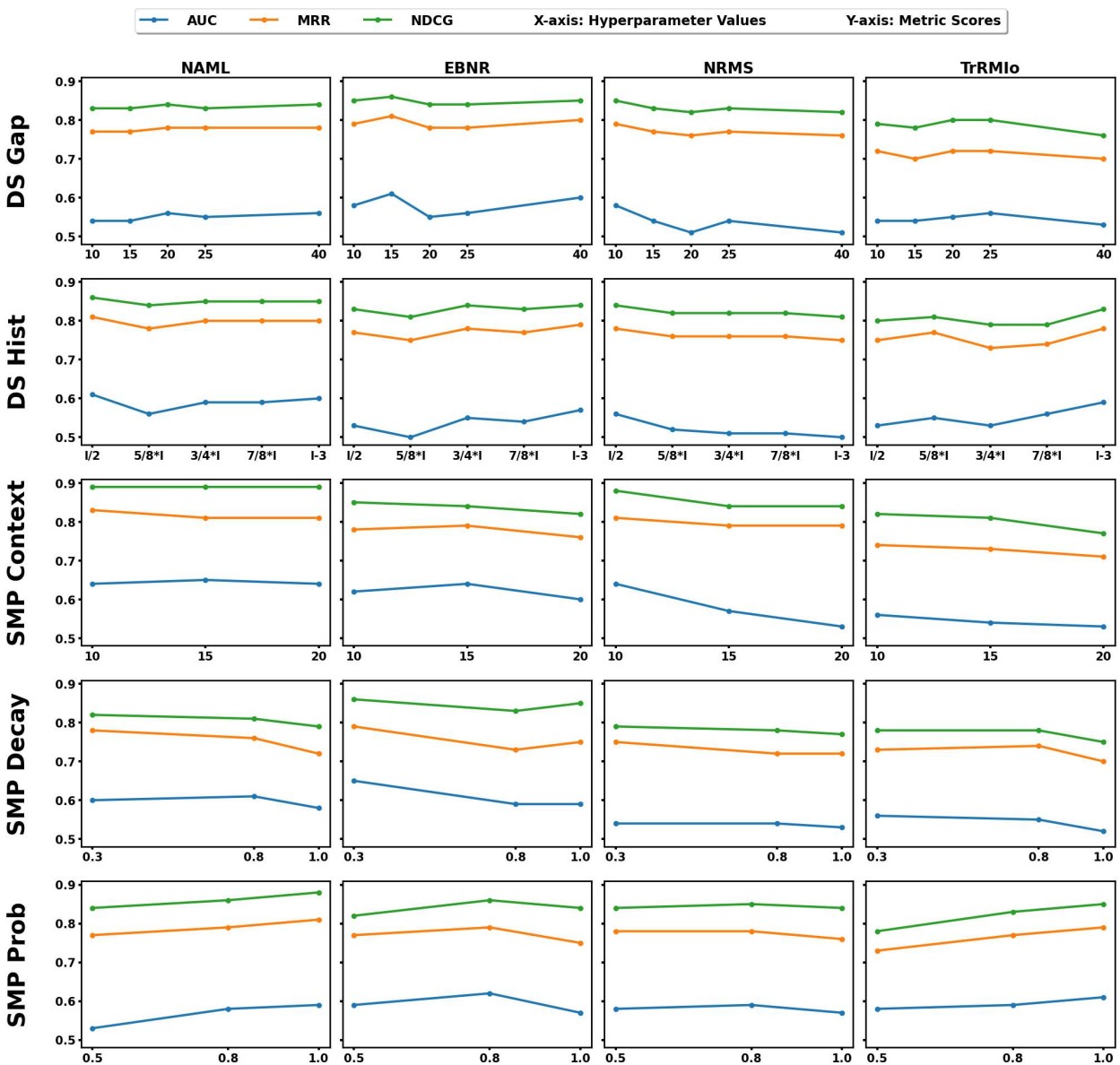

Figure 6: **Effect of `PerAugy` hyper-parameters on User-Encoder Accuracy:** All encoder models are *trained-from-scratch*; results summarized in Table 11. **Observation-1:** *Best hyper-parameter values perform consistently across models;* **Observation-2:** *For DS, $g_l = 40$ and $\tau_{h_{train}} = l - 3$ favor longer profile/history retention;* **Observation-3:** *For SMP, $k = 10$, $p_{SMP} = 0.8$, $\lambda = 0.3$ control abrupt diffusion best, and non-Markovian smoothing is preferred.*

**Effect of $k, \lambda, \& p_{\text{SMP}}$**   We observe that $p_{\text{SMP}}$ has the maximum impact (fixing $k = 10; \lambda = 0.5$), particularly for ranking metrics (MRR, nDCG@5/10). We find that $p_{\text{SMP}} = 0.8/1$ have highest boost (0.04 ↑ w.r.t nDCG@10). This shows that SMP smoothing is mostly required during augmentation. We also observe that the length of the context window ($\tau_{c_k}^{u_{\text{target}}}; \lambda = 0.5; p_{\text{SMP}} = 0.8$) also has a significant effect on the overall AUC (0.05 ↑) and nDCG@5/10 (0.032 ↑) with the best at $k = 10$. With the best $k$ and $p_{\text{SMP}}$ (0.8), we find the best $\lambda$ to be 0.3, particularly for MRR (0.047 ↑) and nDCG@5 (0.032 ↑). This shows that (a) long context window is not useful for SMP smoothing and (b) smoothing cannot be strictly Markovian.

| Hyper-parameter | AUC | | MRR | | nDCG@5 | | nDCG@10 | |
|---|---|---|---|---|---|---|---|---|
| | RWM | AWM | RWM | AWM | RWM | AWM | RWM | AWM |
| $g_l$ | 0.018 | 0.043 | 0.005 | 0.021 | 0.007 | 0.023 | 0.007 | 0.018 |
| $\tau_{h_{train}}$ | 0.024 | **0.064** | 0.013 | 0.035 | 0.011 | 0.026 | 0.011 | 0.028 |
| $k$ | **0.028** | 0.049 | 0.016 | 0.027 | 0.016 | 0.032 | 0.016 | 0.032 |
| $\lambda$ | 0.019 | 0.034 | **0.023** | **0.047** | 0.01 | 0.029 | 0.01 | 0.029 |
| $p_{\mathrm{SMP}}$ | 0.019 | 0.04 | 0.018 | 0.039 | **0.023** | **0.035** | **0.018** | **0.04** |

Table 11: **Comparative impact of hyper-parameters.** Metrics shown are Relative Win Margin (RWM) and Absolute Win Margin (AWM). **Observation:** *Shorter gap-length leads to consistent wins across encoders w.r.t AUC, but for prediction ranking, higher perturbation probability and context-window length matter more.*

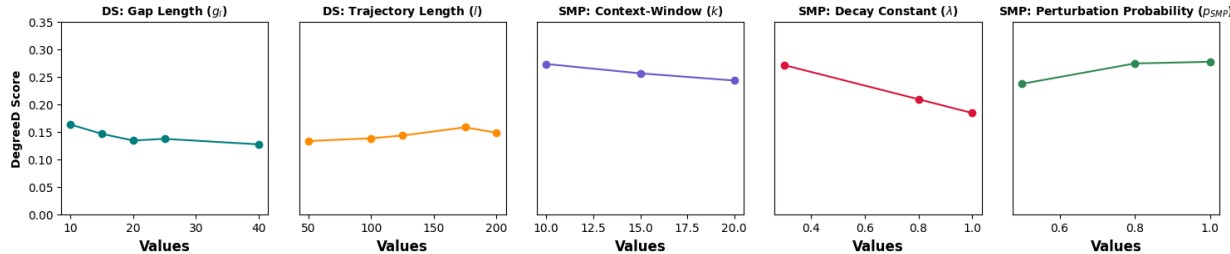

Figure 7: **Ablation effect of hyper-parameters on DegreeD:** Diversity analysis for all hyper-parameters of PerAugy; **Observation-1**: *lower gap length increases diversity due to diffusion into different topics across a trajectory*; **Observation-2**: *Longer context window may not lead to sufficient perturbation*; **Observation-3**: *stricter Markovian does not yield higher diversity*; & **Observation 4**: *frequent SMP on s-nodes lead to higher diversity.*

### I.2 DegreeD Ablations

We ablate various hyperparameters of PerAugy, including gap length $g_l$ and trajectory length $l$ for $\mathcal{T}_{\mathrm{DS}}^{\mathcal{E}-\mathrm{P}}$, as well as context length $k$, decay constant $\lambda$, and perturbation probability $p_{SMP}$ for $\mathcal{T}_{\mathrm{DS+SMP}}^{\mathcal{E}-\mathrm{P}}$. A summary of our findings is given in Figure 7.

**Gap Length $g_l$ and Trajectory Length $l$.** Smaller values of $g_l$ generally lead to higher DegreeD, with $g_l = 10$ yielding the best results (DegreeD of 0.163). This suggests that *frequent substitutions in 'source' segments boost thematic divergence.* Similarly, increasing $l$ results in higher DegreeD, with the highest score (0.158) observed at $l = 175$. This indicates that *the length of 'source' segments plays a crucial role in promoting diversity.*

**SMP Parameters.** We observe that smaller values of the context window $k$ and the decay constant $\lambda$ lead to higher DegreeD, while higher $p_{SMP}$ improves DegreeD, with the optimal setup being $k = 10$, $\lambda = 0.3$, and $p_{SMP} = 1$ (with a score of 0.278). This suggests that, while there is a *Markovian effect* (as a lower $k$ results in higher diversity), the role of *higher-order influence* should not be overlooked. In other words, user-generated subjective summaries are not solely governed by a Markovian process (as evident from the fact that lower $\lambda$ corresponds to higher DegreeD), and might have long-term dependencies. We conclude that user behavior exhibits a tendency toward *diffusion* (random or exploratory variation in a users reading behavior); however, an abrupt diffusion does not necessarily lead to higher diversity. This underscores the importance of *SMP as a smoothing mechanism to regulate diffusion.* The comparative analysis of the ablations on different hyperparameters w.r.t DegreeD are in Table 11.

**Comparative Impact of Hyperparameters.** We conduct a detailed ablation to understand the influence of each hyperparameter of PerAugy on user-encoder model performance across AUC, MRR, nDCG@5 & 10 using *Relative Win Margin* (the difference between the best and second-best performance for a hy-

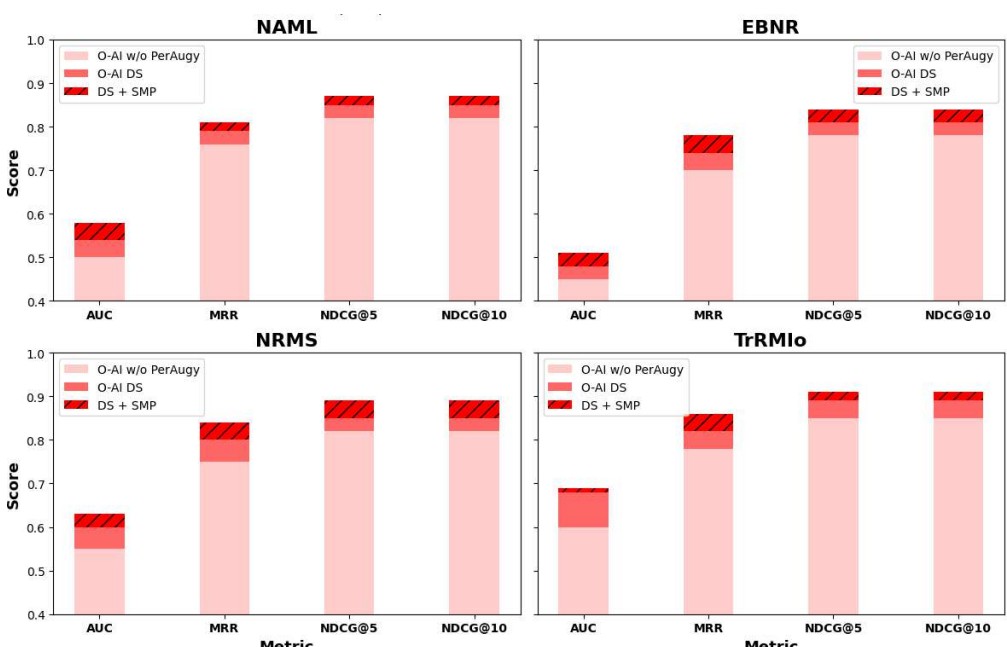

Figure 8: **User-encoder performance (OpenAI (Reddit)):** Impact of Double Shuffling (DS) and DS+SMP on OpenAI seed base $\mathcal{T}_{\text{base}}^{\text{OAI}}$ (SMP hyper-parameters: $k$=10, $\lambda$=0.3, $p_{\text{SMP}}$=0.8). TrRMIo is fine-tuned; others are trained from scratch. **Observation-1:** *Vanilla OpenAI lags behind DS and DS+SMP, indicating that random augmentation is less effective than* PerAugy*;* **Observation-2:** *DS achieves performance boosts as in* PerAugy-*PENS;* **Observation-3:** *DS+SMP further boosts performance, demonstrating the cross-domain strength of* PerAugy*.*

perparameter) and *Absolute Win Margin* (difference between best and worst). History Length exhibits the strongest effect on AUC with an Absolute Win Margin of 0.064 and a Relative Win Margin of 0.024, indicating that *longer historical context is crucial for general user modeling.* Context-Length $k$ provides the highest Relative Win Margin of 0.028 for AUC and consistent gains across all ranking metrics, showing the *importance of larger context windows.* Decay Constant $\lambda$ achieves the highest MRR Absolute Win Margin of 0.047 and a strong Relative Win Margin of 0.023, highlighting the *impact of temporal recency weighting.* Perturbation Probability $p_{\text{SMP}}$ leads in ranking metrics, with nDCG@5 Relative=0.023, Absolute=0.035 and nDCG@10 Absolute=0.04, suggesting *higher perturbation improves top-k relevance.* Finally, Gap Length contributes stable gains to AUC (Abs.=0.043), indicating that *shorter gaps between actions lead to consistent encoder performance.* Overall, each hyperparameter uniquely benefits different objectives, and *careful tuning is vital for optimal performance.* The detailed results are in Table 11.

## J   Comparative Study

In this section, we provide a comparative analysis of SOTA relatable data augmentation techniques based on the operations they perform on trajectory-like datasets. In terms of operations, the methods are divided into two main categories: Intra-Trajectory Augmentation and Cross-Trajectory Augmentation, serving the purpose of sequential recommendation tasks.

### J.1   Intra-Trajectory Augmentation

**S3 Grover et al. (2024):**  Segment-Shuffle-and-Stitch is an intra-trajectory augmentation where non-overlapping segments within same trajectory sequences are segmented, shuffled and finally concatenated (stitched) to form a new optimal trajectory. Since our goal is to generate diverse synthetic trajectories,

shuffling among the segments within same sequence does not guarantee a smooth thematic transition of s-nodes w.r.t the historical interactions. Also, in our case, Shuffling does not enhance diversity w.r.t. `DegreeD`.

**MBASR Xiao et al. (2024a):** Multi-behavior Augmentation for Sequential Recommendation method employs an intra-trajectory augmentation technique by performing pairwise swapping of segments to generate diversity. However, in our case, the base historical dataset has highly monotonous trajectories. Therefore swapping nearby subsequences fails to produce sufficient diversity, and additional operations like order perturbation or redundancy reduction do not effectively smoothen the s-node content in line with historical preferences and might inject (or remove) unrealistic time-step information thereby disrupting the flow of the trajectory. For these reasons, we do not adopt MBASR.

**STEAM Lin et al. (2023):** STEAM operates in an intra-trajectory manner by deciding whether to drop or insert nodes within a trajectory to create augmented data. However, the method is not scalable to longer trajectories, and the insertion or deletion of nodes can disrupt the historical sequence, ultimately undermining the realistic flow of the synthetic user profiles. Hence, we do not use STEAM.

**L2Aug Wang et al. (2022):** Learning-to-Augment is an intra-trajectory augmentation method where a node is deleted from the sequence of core users to generate sequence of synthetic casual users through a reinforcement learning-based policy mechanism. Node deletion is irrelevant in our case as it can disrupt the sequential flow of the trajectories.

**BTBR Li et al. (2023):** Bi-directional Transformer Basket Recommendation model incorporates masking strategies and swapping operation to train the model for 'Next Novel Basket Recommendation'. Despite some similarity in the purpose at broader level, our goal is not to create a model/encoder that encodes the input sequence but to generate a diverse input sequence to make the existing encoders learn the representations.

## J.2 Cross-Trajectory Augmentation

**SDAinter Jiao et al. (2024):** SDAinter is a cross-trajectory technique that matches anchor items (e.g., identical start and end d-nodes or s-nodes) across trajectories to facilitate segment exchange. However, the reliance on anchor-based matching does not effectively capture the subjective nuances of individual user interests, limiting its applicability in personalized summarization. For this reason, we do not consider SDAinter suitable for historical interaction sequence-based tasks.

**DR4SR Yin et al. (2024):** Data-Regeneration-for-Sequential-Recommendation is a transformer-styled cross-trajectory sequence regeneration model where the pertaining task is constructed for to extract patterns from given set of sequences and feed the patterns to the model to regenerate other set of possible sequences. However, patterns in our case would mean reading and summarizing habits of two different users, where the s-nodes are subjective. Therefore, this technique might incorporate redundant s-nodes, defeating the goal of personalized summarization.

**TiCoSeRec Dang et al. (2024):** Time Interval Aware Augmentation technique ensures uniform time-interval distribution in the sequence based on the time-aware traditional operations like Crop, Mask, Insert, Reorder and Substitute. However, our trajectories are primarily assumed to be uniform in terms of time-steps (unit time between two successive interactions).

**FDA Chen et al. (2023a):** Fairness-oriented Data Augmentation is used to generate synthetic user profiles from the realistic profiles to balance between realistic data and pseudo pseudo data. However, modeling historical preference trajectories by generating fake interaction sequences will not lead to diversified trajectories as the 'complemented' sequence will also remain monotonous. Also, ideal datasets for personalized summarization tasks must have intermediate summary nodes for supervised learning setup, which makes the generation of fake interactions challenging.

---

## Chain-of-Thoughts (COT) Prompt

**You are an AI model generating synthetic user interaction trajectories with news articles.**

### Task Definition
Each user follows a sequence of interactions with news articles. The dataset consists of:
- "UserID" : Unique identifier for the user.
- "Sequence of Docs" : Ordered list of news article IDs the user interacts with.
- "Sequence of Actions" : Ordered list of actions taken ( click, skip, gensum, sumgen ).
- "Number of Summary Nodes" : The count of summary nodes (e.g., S-1, S-2 ) generated during the trajectory.

### Rules for Interaction Generation
1. Each User's Trajectory is 100-200 Interactions Long
   - The user interacts with a sequence of news articles.
   - The sequence follows logical decision-making based on relevance and interest.

2. Action Types
   - "click" → User reads the article.
   - "skip" → User ignores the article.
   - "gensum" → User generates a rewritten headline for that document.
   - "sumgen" → User written personalized headline.

3. Summary Node Constraints
   - "sumgen" must immediately follow "gensum" .
   - Each "sumgen" introduces a new summary node ( S-{id} )in the document sequence.
   - Each user must have 3 to 50 summary nodesin their trajectory.

### Step-by-Step Thought Process
1. Assign a UserID
   - Generate a unique identifier for the user.

2. Generate a Long Sequence of Interactions
   - Select 100-200 news articlesfrom various categories.
   - Apply logical reasoning to assign "click" , "skip" , "gensum" , or "sumgen" actions.

3. Ensure Summary Nodes are Introduced Properly
   - When "gensum" occurs, assign it a new summary node ( S-{id} ).
   - The next "sumgen" action must refer to a previously generatedsummary node.
   - Ensure there are at least 3 summary nodes per user.

4. Output the Structured Dataset
   - "UserID"
   - "Sequence of Docs" : Ordered list of article IDs and summary nodes.
   - "Sequence of Actions" : Corresponding user actions.
   - "Number of Summary Nodes"

### Expected Output Format (JSON)
```
{
 "UserID": "U001",
 "Docs": ["N101", "N102", "N103", "S-1", "N104", "N105", "S-2", "S-1", "N106"],
 "Actions": ["click", "skip", "gensum", "sumgen", "click", "gensum", "sumgen", "sumgen", "click"],
 "Num_Summary_Nodes": 3
}
```

Figure 9: Chain-of-Thoughts (CoT) prompt template used in LLM-based experiments.

**divSPA-styled methods Liu et al. (2023):** These methods use a cross-trajectory augmentation strategy by exchanging segments between trajectories based on similarity metrics. Despite this, the exchanged segments often lack sufficient variation with respect to the overall degree (`DegreeD`), resulting in minimal diversity gains. This limitation makes the approach less effective for our needs.

## K  Prompt Details

**Chain-of-Thoughts.**  Chainofthought (CoT) prompting is a powerful technique that guides large language models to decompose complex problems into a series of intermediate reasoning steps before emitting their

final answer. CoT prompting was shown to significantly improve multistep arithmetic, commonsense, and symbolic reasoning tasks by eliciting explicit rationale chains that mirror human logic Ye et al. (2024). Subsequent work in ACL demonstrated that CoT can be extended to address hallucination and faithfulness issues by injecting structured knowledge during rationale generation (Wang et al., 2024). EMNLP findings further confirmed that structured CoT variations, such as statebased prompting, yield substantial gains in contentgrounded dialogue systems by promoting intermediate subtask decomposition Sultan et al. (2024). Another EMNLP study introduced prompt tuning of masked language models to generate both intermediate and final reasoning steps jointly, striking a balance between interpretability and performance without full finetuning (Kunnath et al., 2023). Across these efforts, a consistent insight is that CoT acts as a bridge, enabling LLMs to expose latent reasoning processes. The method is particularly effective for tasks demanding logical coherence and multihop inference, such as math word problems and question answering. Although CoT relies on large model scale to be effective, research shows that even generated exemplars (e.g. Lets think step by step) can approximate fewshot behavior. In the context of our work, CoT prompting with LLaMA213B is used to craft personalized user summaries by breaking down interactions step by step, so as to enhance transparency and accuracy, making LLM reasoning more interpretable and reliable.

**Prompt Chaining.** Prompt chaining is an effective prompting strategy where a complex task is decomposed into a sequence of smaller, well-defined prompts, with the output of one prompt becoming the input to the next, thereby guiding the model through a structured reasoning pipeline. It improves performance on multi-step tasks by reducing cognitive load on the model and increasing transparency at each stagedevelopers can verify and debug intermediate outputs, enhancing controllability and reliability. Academic research, such as , shows prompt chaining excels in iterative summarization by orchestrating drafting, critiquing, and refining phases via discrete prompts, outperforming one-shot or stepwise alternatives (Sun et al., 2024).Across these efforts, the core insight is that chaining leverages the models strengths at each subtask rather than relying on single-shot reasoning, leading to superior performance, especially when task complexity or input length is high. In our context, prompt chaining is implemented via two stepsuser behavior simulation followed by summary generationmirroring the validated pattern of decomposition for enhanced LLM task performance.

---

## Prompt Chaining (Task 1)

### Task
Generate a sequence of interactions for User {user_id}.
Document IDs: {doc_sequence}
Actions: [click, skip]

### Rules:
   1)    Each action corresponds to a document in the sequence.
   2)    The sequence should only contain "click" or "skip" actions at this stage.

### Expected Output Format (JSON)
{
   document_1 : action_1, document_2 : action_2, document_n, action_n
}.

---

## Prompt Chaining (Task 2)

### Examples
Below are examples of personalized headlines generated for different users based on their document content:

{fewshot_examples}

### Task:
Generate a personalized headline based on the document content. Ensure that the headline aligns with the user's preferences and effectively captures the essence of the document.

User: {user_id}
Document: {doc_id}
Document Content: {doc_content}

### Expected Output Format (JSON)
Strictly return a JSON object in the following format:
{
   "headline": "your generated headline"
}

---

Figure 10: Prompt-Chaining template used in LLM-based experiments.

