# OpenReview forum: "Diversity Augmentation of Dynamic User Preference Data for Boosting Personalized Text Summarizers"
_TMLR — Accepted by TMLR_

### Review · Reviewer_qSYo · 2025-08-09

**Summary Of Contributions:**

* The paper proposes DegreeD, a novel metric intended to capture the diversity of user-preference trajectories for training personalized text summarization models. Lower DegreeD implies that the training data contain limited user interests, which may lead to overfitting and weaker generalization.
* The paper also proposes PerAugy, a novel user trajectory augmentation pipeline for the same application. Specifically, PerAugy combines Double Shuffling (DS) and Stochastic Markov Perturbation (SMP), which effectively generates realistic synthetic user trajectories.
* Experimental results show that PerAugy improves the personalized text summarization performance across multiple models and datasets, especially for two distinct dataset types (PENS vs. OpenAI-Reddit).

**Additional Comments:**

* The reference style makes the paper difficult to read. Please use proper parentheses.
* The figures and corresponding explanations are sometimes difficult to follow. Before and after do not look very different. Please consider adding realistic examples or consider an alternative visualization to illustrate the idea.
* I am somewhat confused about the “genSum” and “summGen”. So, the “genSum” action connects a document to its summary (which is a document node, not a summary node…). Then, “summGen” action again connects the summary to the new summary node?
* Is the test set size of 103 user trajectories sufficient? Can you compare the size of the test datasets commonly used in this domain?
* Can you explain more about ‘why’ the “unique topics per trajectory” or “Rate of change of topics” are insufficient, especially compared to DegreeD? They are interpretable and easy to compute.

**Audience:**

No

**Audience Explanation:**

I acknowledge that this topic is important yet underexplored. However, to broaden appeal and accessibility, especially for readers new to personalized summarization, please expand the related-work context and clarify the definitions, figures, and notations.

**Broader Impact Concerns:**

Although the authors do not explicitly discuss the broader impact, I think this paper does not have ethical or political concerns. On the other hand, the paper includes a limitation section.

**Claims And Evidence:**

No

**Claims Explanation:**

* The main contribution, the DegreeD metric, does not seem sufficiently supported by experimental results. The reader would expect that a higher DegreeD will improve the end performance; however, Tables 2–3 alone do not establish a monotonic relationship between DegreeD and downstream performance across baselines. Although the authors present correlation results in Table 4; it would strengthen the claim if the paper reports the correlations computed ‘without’ (i.e., exclude) PerAugy.
* It needs to be clearly explained that PerAugy is targeted to increase the DegreeD score. In other words, they both start with the same intuition that diverse trajectories would lead to better results. Because PerAugy is designed to increase DegreeD, there is a risk of circularity.
* DegreeD relies on the embedding space from a specific model and specific distance (e.g., Manhattan distance). It is unclear if the proposed metric is dependent on a certain embedding model or distance type.
* The notation can be improved further. For example, the divergence ($delta$) operator is dependent on the timestep $t$, but the notation itself does not include $t$. Overall, the notations are too complicated, and I needed to revisit the notations repeatedly – yet I could not sure I have understood the process correctly.
* The definition of DePS is somewhat difficult to follow, particularly in equations (1)-(5). Maybe a concrete example would help readers understand the concept of DePS. In particular, please justify why the min/max ratio is the right choice for capturing proportional shifts between document and summary divergences.

**Requested Changes:**

Please see the other sections.

---

> ### Author Response · Authors · 2025-09-08
> **Rebuttal to the review by reviewer qSYo - PART-I**
>
> We thank the reviewer for the thoughtful and detailed feedback. We carefully considered each concern and provide our responses below.
>
> $\textbf{Concerns regarding Stronger Support for DegreeD-Accuracy Correlation:}$ We thank the reviewer for pointing out this valid concern. As per the suggestion, we strengthened the evidence on two fronts in Sec 7.3: (i) stability w.r.t PerAugy as a strong outlier (Tables 6) and (ii) stability w.r.t strong user-encoder outlier (Table 7) across all datasets (both original and augmented). We find that \textbf{DegreeD (along with one of the other diversity metrics, TP) tracks model accuracy robustly even when excluding PerAugy-augmented datasets}. Also, we observe that Degreed and TP are stable w.r.t. strong positive user-encoder outlier, and hence, the aggregate mean correlation reported in Table 6 is reliable. We also give a detailed mathematical analysis on the stability of DegreeD's correlation results upon substitution of $\sigma$ with other alternative $\sigma$ (App. F, p.37, L36-49; p. 38, L17).
>
> $\textbf{Circularity of PerAugy and DegreeD:}$ We completely acknowledge the misunderstanding that has been raised across the reviewers. We admit that the paper's narrative structure led to that, and hence, \textbf{we have restructured the revised manuscript by keeping DegreeD and other diversity metrics in a separate discussion (Sec 7) on quantification of the induced diversity of PerAugy (and other augmentation methods)}. We would like to clarify that PerAugy is designed to increase \emph{trajectory diversity} in UIGs, not to optimize DegreeD directly (i.e., as an optimization objective). The DS operation, which is key to diversity induction, does not explicitly model the three types of penalties (unfaithfulness, disproportionate divergence, and lack of divergence) baked into DegreeD. In other words, PerAugy is agnostic to the ratio-based alignment of document vs.\ summary shifts that DegreeD measures. To emphasize, DegreeD is a model/performance-metric-agnostic $\textbf{post-hoc diagnostic dataset diversity metric}$ to $\textit{quantify (and not prove) the role of dataset diversity in performance gains}$ (Sec. 7.1). This makes $\textbf{PerAugy the key contribution of the paper, and not DegreeD}$ (which is merely a diversity metric like TP and RTC). This addresses circularity by decoupling PerAugy's mechanism from the evaluator.
>
> $\textbf{Concern regarding DegreeD's Sensitivity to Choice of Embedding and Divergence Metric:}$  We understand the concern and acknowledge that this has not been clearly treated in the original manuscript. We clarify that DegreeD computation depends on $\textbf{relative}$ ratio rather than absolute scales (App. E, p. 33-34, L36-42: "$\textit{any embedding model ... and any valid distance metric $\sigma$ (e.g., Manhattan, cosine, Euclidean) merely provide a representation space}$ ... $\textit{the ratio-based structure of DegreeD cancels out biases due to embedding geometry or metric scaling. Hence, DegreeD remains a model- and}$ $\textit{metric-agnostic measure ...}$). We also clarify the default setting (App. E, p. 33, L47-50): "$\textit{we represent d/s-nodes with S-BERT embeddings ... and use Manhattan Distance as}$ $\sigma$". At the same time, we provide a theoretical analysis that establishes the stability of DegreeD under divergence metric substitution (App. F, p. 37, L36-49; p. 38, L17): {"$\textit{Theorem 1 (Correlation stability of DegreeD) ... Scaling invariance ... preserved exactly.}$"}). We, however, would like to emphasize that DegreeD (and any other arbitrary dataset diversity metric) is only important to the extent of strengthening the \textit{interpretability} of what the effect of DS and SMP operations is w.r.t diversity. In fact, even the simpler TP metric shows the same trend, further confirming that $\textbf{our findings do not hinge on one specific metric design - it is more of a question of which is a better}$ $\textbf{diagnostic metric rather than a conclusion on the performance of PerAugy itself.}$
>
> $\textbf{Concern for Clarification of DegreeD Notations:}$ We simplified and standardized notation and consolidated all symbols in Table 1 (p. 15). Clarifications include explicit superscripts/subscripts for $\delta$ across document vs.\ summary channels and introduction of the regulator $\alpha$ (p. 15, L19): "$\textit{$\alpha$ Regulator that controls the influence of Penalized DePS..}$. We also tightened the surrounding prose in Sec. 7.1.
>
> $\textbf{Concern regarding Example-based Clarification of DePS and Clarification of min-max ratio:}$ We added worked examples and intuition in Sec. 7.1. Regarding min-max, the min/max design yields bounded, scale-invariant proportionality, aligning with the ratio-based robustness emphasized in App. F (bi-Lipschitz stability) and Sec. 7.1.
>
> $\textbf{Concern related to  Elaboration of Related Work}: $ We expanded Sec. 8 (Related Work).

---

> ### Author Response · Authors · 2025-09-08
> **Rebuttal to the review by reviewer qSYo- PARTII**
>
> $\textbf{Confusion between genSumm and summGen: }$ We clarified the UIG actions explicitly in Sec. 3 (p. 4, L48 \& L52). Inserted text: summarize (also called genSumm) explicitly captures the interest to read a summarized version of the d-node and "$\textit{the follow-up edge of summarize denoting summarized version of $d_{t_{q-1}}$, acting on $s_{t_q}$ (also termed as summGen)}$". (Sec. 3, p. 4, L48-52; Fig. 3)
>
> $\textbf{Regarding Sufficiency of Test Data: }$ We describe the construction and scale of the PENS test setup used for next-click and summarization evaluation. The test pool is substantial: "$\textit{The final test set contains 103 trajectories with $\approx$20K candidate pool ... of d-nodes with 10K target d-nodes.}$" (Sec. 5.1.1, p. 9, L52-54) This PENS test settings is largely adopted in literature "{PENS}: A Dataset and Generic Framework for Personalized News Headline Generation, Ao et. al.", "General then Personal: Decoupling and Pre-training for Personalized Headline Generation, Song et.al."(Table 8).
>
> $\textbf{Usage of Simpler TP/RTC Metrics: }$ We do a head-to-head comparison of TP, RTC, and DegreeD, both theoretically and empirically. Table 4 shows that TP/RTC correlate with diversity but fail to penalize mis-aligned summary shifts. Caption (p. 16, L17): "$\textit{While TP and RTC relate to diversity, they fall short in capturing preference ...}$". DegreeD explicitly measures proportional alignment between document and summary drift, which best tracks personalization accuracy (Sec. 7.3).
>
> $\textbf{Regarding Reference Style: }$ We have corrected it throughout (there was a newer TMLR package compatibility issue).

---

### Review · Reviewer_hXxr · 2025-08-18

**Summary Of Contributions:**

This paper introduces DegreeD, a metric for measuring diversity in user preference trajectories, and PerAugy, a data augmentation method that combines cross-trajectory shuffling with summary perturbation to generate more diverse and realistic interaction histories, thereby achieving higher DegreeD scores. Experiments show that PerAugy  improves user-encoder performance and boosts personalization for downstream SOTA summarizers.

### Strengths

- Tackles an important challenge in personalized summarization: lack of diverse training data.

- Strong experimental evaluation across multiple baselines, including heuristic and LLM-based augmentation methods.

- Demonstrates both theoretical (diversity–performance correlation) and practical (improved summarization quality) contributions.

### Weaknesses

- The method introduces multiple hyperparameters which raises concerns about sensitivity, reproducibility, and potential overfitting to specific datasets.

- Current evaluation focuses heavily on PENS; broader coverage on other domains or languages would strengthen claims.

- The interpretability of DegreeD is less intuitive than simpler diversity metrics, and the connection to personalization performance does not feel very direct to me.

**Audience:**

Yes

**Audience Explanation:**

This work addresses the data scarcity problem in personalized NLP. The proposed augmentation strategy has potential applications beyond summarization, including other preference-driven tasks such as recommendation and dialogue personalization, which broadens its relevance to TMLR’s readership in both LLM4Rec and NLP. I also found Section 3 on user preference modeling to be an interesting and timely direction that is likely to attract attention from the community.

**Claims And Evidence:**

No

**Claims Explanation:**

While I appreciate that the paper addresses an overlooked problem and introduces both a new evaluation metric and an augmentation methodology, I have key concerns. The tight coupling of the proposed metric (DegreeD) and the method makes it difficult to independently justify the methodology and weakens the overall evidence. In Section 6.1.4 (RQ-2), the authors present a small-scale study suggesting a positive correlation between DegreeD and user-encoder performance, but this feels insufficient. Furthermore, I believe the new metric should be validated more rigorously, for example through human assessments or by demonstrating consistent alignment with existing other evaluation metrics.

**Requested Changes:**

### Major

- Provide more justification and intuition behind DegreeD, since the entire paper relies on it. This could include comparisons with simpler existing metrics or validation against human assessments.

- Clarify the sensitivity of PerAugy to hyperparameters and offer practical guidelines to improve reproducibility.

- Please correct the formatting of references, as the current style hinders readability badly.

### Optional

- Provide some qualitative examples or cases of augmented trajectories.

- Discuss the limitations when ground-truth preference histories are unavailable (i.e., anyway to cold start?).

---

> ### Author Response · Authors · 2025-09-08
> **Rebuttal to the review by reviewer hXxr - PART-I**
>
> We are grateful to the reviewer for the insightful comments and suggestions, which we found very helpful in refining both our analyses and presentation. We address the concerns one by one below.
>
> $\textbf{Regarding Sensitivity to Hyperparameters and Overfitting Risk: }$  We added a comprehensive ablation suite (App. I.1-I.2). Sec. I.1 details RQ-1 ablations over DS gap‐length $g_\ell$, train history length, and SMP parameters $k$, $\lambda$, $p_{SMP}$ (p. 39, L51-55). Sec. I.2 summarizes DegreeD ablations with concrete settings (p. 41, L27-29) -- "$\textit{optimal setup being $k=10$, $\lambda=0.3$, and $p_{SMP}=1$ (with a score of 0.278)}$". These results show robustness windows rather than knife-edge tuning. Also, models trained in PerAugy-augmented OpenAI-Reddit dataset (augmented under the same configurations as PENS augmentation) have been tested in the PENS test dataset, showing promising results. This also indicates that underfitting/overfitting cannot be attributed to the hyper-parameters. Detailed results are given in Sec 6.1 (Cross-domain study) and corresponding Figure 8 (Appendix I; pg: 42).
>
> $\textbf{Regarding Comparative Study with Simpler Interpretable Metrics: }$ We thank the reviewer for pointing out this very important discussion that was omitted in our last submission. In the revised version, we do a head-to-head comparison of TP, RTC, and DegreeD, both theoretically and empirically. Table 4 shows that TP/RTC correlates with DegreeD. Also, we find that the PerAugy-augmented dataset diversity across all three metrics (DegreeD, TP, RTC) shows an increase when compared to the corresponding original dataset. $\textbf{This helps us to consistently quantify the induced diversity by PerAugy}$. We also discuss at length how TP and RTC can fail to penalize misaligned summary shifts but at the same time are more interpretable. Caption (p. 16, L17): "$\textit{While TP and RTC relate to diversity, they fall short in capturing preference ...}$". DegreeD explicitly measures proportional alignment between document and summary drift, which best tracks personalization accuracy (Sec. 7.3). We would finally like to stress that $\textit{our findings on the performance of PerAugy do not hinge on one specific metric design}$ - it is more of a question of $\textbf{which is a better post-hoc diagnostic metric of the induced diversity}$ $\textbf{(not just by PerAugy but by any other augmentation method), rather than a conclusion on the performance of PerAugy itself.}$
>
> $\textbf{Concern regarding Stronger Support for DegreeD-Accuracy Correlation; and Degreed-PerAugy Tight Coupling}$ We admit that the paper's narrative structure led to the misunderstanding of DegreeD being tightly coupled to PerAugy. Hence, $\textbf{we have restructured the revised manuscript by keeping DegreeD and other diversity metrics}$ $\textbf{in a separate discussion (Sec 7) on quantification of the induced diversity of PerAugy (and other augmentation methods)}$. We would like to clarify that PerAugy is designed to increase $\textit{trajectory diversity}$ in UIGs, not to optimize DegreeD directly (i.e., as an optimization objective). The DS operation, which is key to diversity induction, does not explicitly model the three types of penalties (unfaithfulness, disproportionate divergence, and lack of divergence) baked into DegreeD. In other words, PerAugy is agnostic to the ratio-based alignment of document vs.\ summary shifts that DegreeD measures. To emphasize, DegreeD is a model/performance-metric-agnostic $\textbf{ post-hoc}$  diagnostic dataset diversity metric to \textit{ quantify} (and not prove) the role of dataset diversity in performance gains (Sec. 7.1). This makes $\textbf{PerAugy the key contribution of the paper, and not DegreeD}$ (which is merely a diversity metric like TP and RTC). As for the suggestion regarding strengthening the correlation results, we did that Sec 7.3: (i) stability w.r.t PerAugy as a strong outlier (Tables 6) and (ii) stability w.r.t strong user-encoder outlier (Table 7) across all datasets (both original and augmented). We find that $\textbf{DegreeD (along with one of the other diversity metrics, TP) tracks model accuracy robustly even when excluding PerAugy-augmented datasets}$. Also, we observe that Degreed and TP are stable w.r.t strong positive user-encoder outlier, and hence, the aggregate mean correlation reported in Table 6 is reliable. We also give a detailed mathematical analysis on the stability of DegreeD's correlation results upon substitution of $\sigma$ as RMSD with other alternative $\sigma$ (App. F, p. 37, L36-49; p. 38, L17).

---

> > ### Author Response · Authors · 2025-09-08
> > **Rebuttal to reviewer hxXr - PART-II**
> >
> > $\textbf{Regarding Coverage beyond PENS dataset: }$ We agree that this is a very valid concern. However, we would humbly like to point out that we have done cross-domain analysis leveraging OpenAI (Reddit) data. This dataset is a non-news set of Reddit threads covering 29 different domains, as compared to the PENS dataset (Sec 3.2, Sec 5.1.1: Training Data). OpenAI-Reddit is also structurally different than Reddit. We find diversity gains and accuracy trends persist (Sec 6: Cross-domain-study and App. I: Fig. 8).
> >
> > $\textbf{Regarding Reference Formatting: }$ We have corrected it throughout (there was a newer TMLR package compatibility issue).
> >
> > $\textbf{Regarding Qualitative Examples of Augmented Trajectories: }$ We added concrete DS/SMP examples and case studies—Fig. 3(b,c) and Sec. 4.2 include explicit trajectory snippets (p. 7, L20-21).
> >
> > $\textbf{Limitations without ground-truth histories: }$ We have discussed this issue in Sec. 9 (p. 20, L18): \textcolor{blue}{"\textit{model-generated summaries can serve as effective proxies, especially during cold-start scenarios \dots}"}. We also show how OpenAI-Reddit styled datasets can be appropriated to facilitate bootstrapping (Sec. 3) -- "$\textit{Additionally, it also addresses cold-start problem as $\mathcal{T}_{\text{base}}^{\text{syn-OAI}}$ itself is synthetically designed as a random sequence}$".

---

### Review · Reviewer_si1k · 2025-08-28

**Summary Of Contributions:**

The paper addresses the lack of diversity in user interaction datasets for personalized summarization. It argues that current datasets (e.g., PENS, OpenAI Reddit) are too small and fail to capture realistic shifts in user preferences, which limits the generalizability of user-encoder models. To tackle this, the authors propose DegreeD, a metric to quantify preference-shift diversity, and PerAugy, a two-step augmentation pipeline combining Double Shuffling (DS) to mix trajectory segments across users and Stochastic Markovian Perturbation (SMP) to smooth coherence issues in summaries. DegreeD is computed using SBERT embeddings and Manhattan distance to capture divergences between documents (d-nodes) and summaries (s-nodes). At the same time, PerAugy generates synthetic but realistic user histories that preserve temporal coherence.

The experiments evaluate user encoders (NAML, NRMS, EBNR) and personalized summarization frameworks (PENS for extractive, GTP/TrRMIo for generative) across multiple datasets, including PENS, OpenAI Reddit, and LLM-generated UIGs. Results show that the base datasets have a low DegreeD (<0.01), while PerAugy boosts it substantially (up to approximately 0.28). Higher DegreeD correlates strongly with encoder accuracy (Pearson r = 0.68–0.69 across AUC, MRR, nDCG). Ablation studies confirm DS increases diversity, with DS+SMP producing the most coherent trajectories. Finally, integrating improved encoders into summarization frameworks reveals that GTP models benefit most (up to 4 times gains in AUC/nDCG), while PENS shows only modest improvements.

**Audience:**

Yes

**Audience Explanation:**

Researchers in personalized summarization, recommender systems, and data augmentation introduce a novel diversity metric (DegreeD) and augmentation pipeline (PerAugy), with clear implications for personalization and dataset quality. Additionally, the proposed method may be adaptable to other types of time series datasets.

**Claims And Evidence:**

No

**Claims Explanation:**

Claim 1: DegreeD is a valid metric for dataset diversity, and Current datasets lack diversity.

The paper reports very low DegreeD scores for baseline datasets (PENS: 0.009 and OpenAI (Reddit): 0.008). In addition, other augmentation methods did not improve DegreeD as much (Table 2). Correlation study in Meta-evaluation (Table 4) shows that DegreeD positively correlates with encoder performance metrics.

While it is plausible that high diversity leads to higher performance, the correlation between DegreeD and performance does not establish a direct trend with diversity. Alternatively, albeit less likely, it might be that the other augmentation method is too diverse and the models fail to learn it (i.e., underfitting rather than overfitting).

Claim 2: Double Shuffling (DS) and Stochastic Markovian Perturbation (SMP) are both necessary

Understandably, the DS is the core of the PerAugy method; however, it is unclear whether SMP is necessary for its effectiveness. There are two main flaws in the supporting evidence. First, there is no experiment without the SMP. Second, the ablation experiment in the Appendix was on the DegreeD. The conclusion is then a two-step reasoning, DS/SMP -> DegreeD -> Model Performance.

Claim 3: PerAugy improves downstream personalization

User encoders (NAML, NRMS, EBNR) trained on PerAugy data outperform those trained on baseline data across AUC, MRR, nDCG. Both summarization frameworks, PENS and GTP, benefit from the proposed method. Significant gains (up to 4 times improvement) are observed in GTP.

**Requested Changes:**

From the comments above:

1. Provide stronger evidence that higher DegreeD directly improves model generalization, rather than only reporting correlations. For example, the paper may present the training performance along with the test performance to emphasize the generalization ability of using the PerAugy datasets.
2. Include experiments with DS-only (without SMP) to test whether SMP contributes to performance improvements directly.
3. Complement ablation evidence with downstream model performance comparisons. This would strengthen the causal link between SMP and improved personalization.

For the paper's clarity

1. Provide simpler examples of the DS. Rather than a figure, it would be much simpler to see a user trajectory (A) is augmented with another trajectory (B) like this [A1, A2, A3, B4, B5, A6, B7, B8, A9] (O=3, G=1). The current description leaves ambiguity (e.g., is it substitution or insertion? What exactly is being replaced in the target trajectory?). A more explicit worked example of multiple users’ trajectories under different gap lengths would help readers follow.
2. Differentiate and clarify $\sigma$. Currently, it uses SBERT embeddings and Manhattan distance. Authors should explain why RMSD/Manhattan was chosen over alternatives like cosine similarity, and how sensitive DegreeD is to this choice.
3. Clarify the notation $\delta$. In Eq. 1, it has no superscript, but in Eq. 5, it does. What is the difference? (Why add $X$ in the notation?)
4. Table 1 misses $a$ and typo in $E_j$ (should have been $\mathbb{E}_j$).

---

> ### Author Response · Authors · 2025-09-08
> **Rebuttal to the reviewer si1k- PART I**
>
> We appreciate the reviewer’s time and effort in carefully examining our work and raising important points that guided several improvements in the revision. We intend to respond to the primary concerns as below.
>
> $\textbf{Regarding Robustness of Correlation Relation and Generalizability Across Alternative Augmentation Methods: }$ We agree that (i) correlations alone do not prove causation and (ii) extremely noisy or misaligned augmentation could in principle harm learning. Two sets of results in our paper address these concerns directly:
>
> $\textbf{(A) Baseline augmentations also induce diversity and (often) improve accuracy (not just PerAugy)}$.
> We evaluate three baseline algorithmic augmentors (PENS-SH, S3-Aug, SDAInter) and three LLM-based augmentors (LLaMA-2, Mistral, DeepSeek). As summarized in Table 2, $\textbf{several baselines improve encoder accuracy relative to the original PENS training set}$, $\textbf{e.g. SDAInter yields consistent gains across NAML, EBNR, NRMS; PENS-SH}. This shows that increased (and well-aligned) diversity is beneficial even when PerAugy is not used. In contrast, others are neutral or harmful, depending on how their induced diversity aligns with trajectory semantics. This, conversely, shows that misaligned or low-quality diversity can hurt, as pointed out by the reviewer -- i.e., “too diverse” (or noisy) data degrading learnability. To make this explicit in the paper, we already enumerate the baseline methods and the way each induces diversity in UIGs (inter-trajectory merges in PENS-SH, intra-trajectory perturbations in S3-Aug, and cross-user subsequence swaps in SDAInter) before measuring their effects on encoders. All the augmentation methods increase dataset diversity w.r.t the three diversity metrics (TP, RTC, and DegreeD) -- see Table 4.
>
> $\textbf{(B) Correlation between diversity and accuracy holds across all datasets, and remains strong even when PerAugy is excluded}$. The meta-evaluation in Sec 7 computes correlations between diversity metrics (TP, DegreeD) and encoder accuracy across the entire set of original and augmented datasets (including baselines). Table 6 shows that DegreeD and TP both correlate positively with accuracy; critically, Observation-3 reports that these correlations remain strong even after excluding PerAugy, so the trend is not because of PerAugy dominating the pool. This directly counters the concern that “only PerAugy goes up and drags correlation” - the relationship between (well-measured) diversity and accuracy persists for other augmentation families too. We also run stability checks (Section 7.3.3): removing PerAugy (“outlier-method analysis”) leaves the correlation band essentially unchanged (reported mean deltas and low variance), and analyzing model-specific correlations shows low inter-model variance, with DegreeD generally more stable than TP. Together, these indicate that the diversity–accuracy trend is robust and not driven by a single method or a single encoder.
>
> To conclude, baseline augmentations do induce diversity and often improve accuracy, and the positive diversity–accuracy association holds broadly, not only with PerAugy. Where some baselines underperform, our results support a $\textbf{$\textit{ quality}$}$-of-diversity explanation (semantic misalignment / noisy diffusion) rather than a blanket "$\textit{too much diversity causes underfitting.}$" PerAugy's DS+SMP is designed precisely to regulate diffusion (DS) and enforce faithfulness/temporal coherence (SMP), which explains its stronger and more consistent gains. (We already highlight that DS-only is weaker than DS+SMP across encoders.)
>
> $\textbf{Regarding the Necessity of SMP Operations: }$ We thank the reviewer for pointing this out. We saw that we omitted the result earlier and have included both DS-only vs.\ DS+SMP comparisons on both encoder and summarizer tasks (Tables 2-3) in the revised manuscript. As noted in Table 3 caption (p. 13, L11-12): "$\textit{SMP consistently improves over DS, indicating its necessity for maximizing gains.}$". Downstream ablations are given in App. I.1; Fig 6, while results are discussed in Sec 6.1.
>
> $\textbf{Regarding Example-based Clarification of DS Operation: }$ We inserted a concrete stitched example in Sec. 4.2 with explicit operations (p. 7). For instance (p. 7, L20-21): "$\textit{Alice’s first three interactions involve reading "Meditation tips''... and ``Yoga retreat guides'' ...}$".

---

> > ### Author Response · Authors · 2025-09-08
> > **PART II**
> >
> > $\textbf{DegreeD's Sensitivity to Choice of Embedding and Divergence Metric: }$ We understand the concern and acknowledge that this has not been clearly treated in the original manuscript. We clarify that DegreeD computation depends on $\textbf{$\textit{relative}$}$ ratio rather than absolute scales (App. E, p. 33-34, L36-42: "$\textit{any embedding model $\dots$ and any valid distance metric $\sigma$ (e.g., Manhattan, cosine, Euclidean)}$ $\textit{merely provide a representation space $\dots$ the ratio-based structure of DegreeD cancels out biases due to embedding geometry or metric scaling.}$ $\textit{Hence, DegreeD remains a model- and metric-agnostic measure $\dots$}$"). We also clarify the default setting (App. E, p. 33, L47-50): \textcolor{blue}{"\textit{we represent d/s-nodes with S-BERT embeddings \dots and use Manhattan Distance as $\sigma$}"}. At the same time, we provide a theoretical analysis that establishes the stability of DegreeD under divergence metric substitution (App. F, p. 37, L36-49; p. 38, L17): {"$\textit{Theorem 1 (Correlation stability of DegreeD) $\dots$ Scaling invariance $\dots$ preserved exactly.}$"). We, however, would like to emphasize that DegreeD (and any other arbitrary dataset diversity metric) is only important to the extent of strengthening the $\textit{interpretability}$ of what the effect of DS and SMP operations is w.r.t diversity. In fact, even the simpler TP metric shows the same trend, further confirming that $\textbf{our findings do not hinge on one specific metric design -- it is more of a question}$ $\textbf{of which is a better diagnostic metric rather than a conclusion on the performance of PerAugy itself}$.
> >
> > $\textbf{Notation Clarification of $\delta$ notation; Concern-5b: Missing Notation Brief in Symbol Table: }$ We fixed notation in Sec. 7.1 and updated Fig 2b Table (p. 15), including the "$\textit{$\alpha$ Regulator that controls the influence of Penalized DePS.}$" Symbols for $\delta$ across timesteps are now explicit.

---

> > > ### Comment · Reviewer_si1k · 2025-09-23
> > > **Good revision**
> > >
> > > I appreciate the authors’ thorough and constructive revision. In my original review, I raised three main concerns: (1) the robustness of the correlation between DegreeD and model performance, and whether this relationship generalizes beyond PerAugy; (2) the necessity of the SMP component, since DS appeared to be the core augmentation; and (3) the need for broader evidence across alternative augmentation methods.
> > >
> > > The revised submission addresses all of these points convincingly
> > >
> > > ### 1. Correlation and Generalization
> > >
> > > The authors provide new analyses (Sec. 7.3, Tables 6–7, App. F) that show DegreeD correlates robustly with encoder accuracy across multiple datasets and augmentation methods, even when PerAugy is excluded.
> > >
> > > They also add stability checks (outlier removal, model-specific variance) and a theoretical justification of metric invariance. These additions strengthen the claim that DegreeD is a reliable diagnostic of diversity, while clarifying that it is not an optimization objective for PerAugy.
> > >
> > > ### 2. Necessity of SMP
> > >
> > > The revision now includes explicit DS-only vs. DS+SMP comparisons (Tables 2–3, App. I.1), showing consistent downstream improvements with SMP. This directly addresses my concern and supports the necessity of SMP for maximizing gains.
> > >
> > > ### 3. Generality Beyond PerAugy
> > >
> > > The authors expand evaluation to include several baseline augmentation methods, both algorithmic (PENS-SH, S3-Aug, SDAInter) and LLM-based (LLaMA-2, Mistral, DeepSeek). Results in Table 2 show that some baselines (e.g., SDAInter across NAML/EBNR/NRMS; PENS-SH with TrRMIo) improve encoder accuracy relative to the original datasets, while others are neutral or harmful depending on how their induced diversity aligns with trajectory semantics.
> > >
> > > Table 4 further shows that all augmentation methods increase dataset diversity across TP, RTC, and DegreeD metrics, though only well-aligned diversity consistently improves downstream performance.

---

> > > > ### Author Response · Authors · 2025-09-28
> > > > **Thank you for your feedback**
> > > >
> > > > Dear Reviewer si1k,
> > > > We thank you for the time and effort that you have put into going through the details of our paper. We strongly believe that the feedback will go a long way for us in contuing this research thread.
> > > >
> > > > Best

---

### Decision · Action_Editor_dZps · 2025-09-29

**Recommendation:** Accept as is

**Additional Comments:**

This paper proposes a new method for personalized text summarization conditioned on user histories. The key idea is to generate synthetic data using two perturbations:

* User trajectory perturbation

* Corresponding perturbation in the ground-truth text summary

The result is a dataset that can be used for training methods for downstream tasks. The proposed approach improves dataset diversity, which correlates with improved performance of the methods trained from more diverse datasets. The authors comprehensively evaluate the approach and also propose a new metric for measuring dataset diversity called DegreeD.

The reviewers liked the topic of the paper but had several concerns:

* **Diversity metric DegreeD:** The metric is neither clearly presented nor properly justified. One confusion is that DegreeD is presented together with the algorithm.

* **Notation:** Clarifications are needed.

* **PerAugy:** Sensitivity to hyper-parameters. Ablate both perturbations mechanisms and show that they are needed.

The authors addressed the concerns of the reviewers, which in turn recommend acceptance.

**Audience:**

Yes

**Audience Explanation:**

Yes. This paper is on the intersection of personalization, recommender systems, and synthetic data generation. All of these are large communities.

**Claims And Evidence:**

Yes

**Claims Explanation:**

Yes. The paper clearly shows that the proposed synthetic-data generation approach PerAugy improves dataset diversity, which correlates with improved performance of the methods trained from more diverse PerAugy datasets.